# SpaceVista: All-Scale Visual Spatial Reasoning from $mm$ to $km$

Peiwen Sun [* 1 2]   Shiqiang Lang [* 3]   Dongming Wu [1]   Yi Ding [2]   Kaituo Feng [1]   Huadai Liu [4]   Zhen Ye [4]   Rui Liu [1]
Yun-Hui Liu [1]   Jianan Wang [† 2]   Xiangyu Yue [✉ 1]

## Abstract

With the current surge in spatial reasoning, researchers have made significant progress in understanding indoor scenes, but still struggle with more diverse applications. This paper aims to advance all-scale spatial reasoning by tackling two key challenges: 1) the heavy reliance on indoor 3D scans and labor-intensive annotations for dataset curation; 2) the absence of all-scale modeling, which often leads to overfitting to single scenes. In this paper, we introduce a holistic solution that integrates a structured spatial reasoning knowledge system, scale-aware modeling, and a progressive training paradigm, as the **first attempt** to broaden the scope of all-scale spatial intelligence. Using a task-specific, specialist-driven automated pipeline, we curate over 38K video scenes across 5 spatial scales to create **SpaceVista-1M**, a dataset comprising 1M spatial QAs spanning 19 diverse tasks. While specialist models offer valuable domain knowledge, they are often unreliable evaluators. Therefore, we build an all-scale benchmark with precise annotations by manually recording and retrieving videos. Nevertheless, naive training with SpaceVista-1M often yields suboptimal results due to the potential knowledge conflict. Accordingly, we introduce **SpaceVista-7B**, a spatial reasoning model that accepts inputs beyond semantics and uses scale as an anchor for scale-aware experts and progressive rewards. Finally, extensive evaluations across 5 benchmarks, including our **SpaceVista-Bench**, demonstrate competitive performance, showcasing generalization across all scales and scenarios. Our demo page is posted on ◉website.

*Equal contribution †Indicates project leader. [1]Multimedia Laboratory, The Chinese University of Hong Kong [2]Astribot [3]Beijing University of Posts and Telecommunications [4]Hong Kong University of Science and Technology. Correspondence to: Xiangyu Yue <xyyue@ie.cuhk.edu.hk>.

*Proceedings of the 43rd International Conference on Machine Learning*, Seoul, South Korea. PMLR 306, 2026. Copyright 2026 by the author(s).

## 1. Introduction

Spatial reasoning, the ability to sense, interpret, and interact with environments across scales from understanding tiny objects to remote drone sensing, is crucial for next-generation intelligent systems. It significantly enhances 3D and even 4D scene understanding, enabling agents to interpret complex environments from easily obtainable videos. **All-scale reasoning** capability supports diverse applications: $mm$ for advanced manufacturing (Song et al., 2024), $cm$ and $m$ for embodied intelligence (Pan et al., 2025), $10m$ for autonomous driving (Liu et al., 2022), and $100m$ for drone-based sensing (Xiao et al., 2023). Recent research (Yang et al., 2025a), especially on how Multimodal Large Language Models (MLLMs) perceive and recall space, is narrowing the gap in visual spatial reasoning.

The current works on spatial reasoning primarily focus on improvements from two perspectives: data and model. From the data perspective, pioneer works (Ouyang et al., 2025; Zhang et al., 2025c; Deng et al., 2025b) utilize more scanning-based data, or image-based data employing fully automated pipelines to acquire additional information for Supervised Fine-Tuning (SFT) and Reinforcement Learning (RL). When modeling indoor spatial scenes, Wu et al. (2025c); Zheng et al. (2025) leverage latent features from VGGT (Wang et al., 2025a) by incorporating geometric information to enhance spatial understanding. Concurrently, a series of outstanding works (Ouyang et al., 2025; Zhang et al., 2025c) have improved the performance of existing models by refining the training and thinking approaches. Moreover, Wu et al. (2025d) employs multi-turn dialogues to enhance self-correction capabilities.

Despite these works' advancements, their spatial perception capabilities are primarily limited to indoor settings, specific objects, and constrained scales, as shown in the bar chart Fig.1. Moreover, current methodologies lack dedicated training frameworks for holistic all-scale scene understanding. To bridge this gap, we introduce the **first comprehensive solution** to address data, model, and evaluation dimensions for all-scale scenarios.

Previous datasets (Yang et al., 2025a;b; Ouyang et al., 2025; Zhang et al., 2025c) for spatial reasoning have primarily been constructed based on indoor scanning video data (Dai

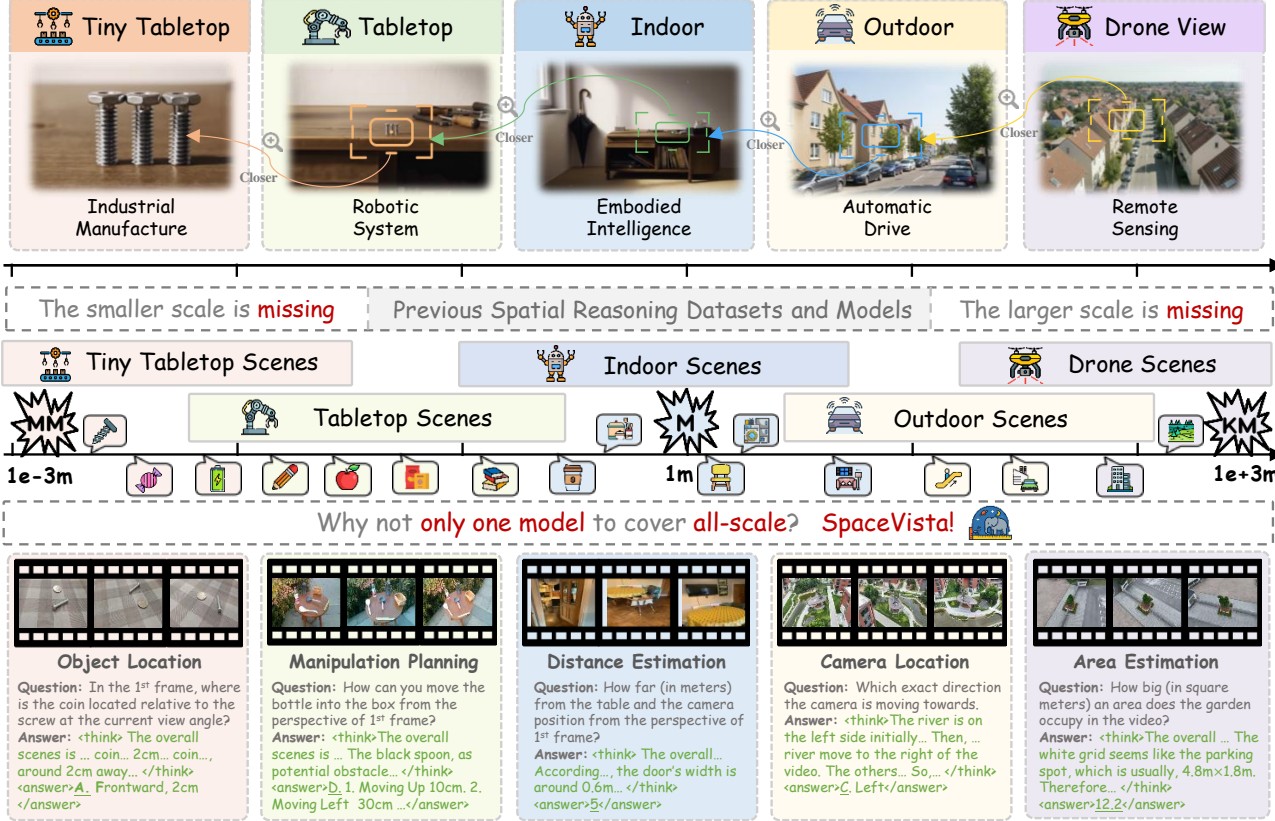

**Figure 1.** Prior works of spatial reasoning have largely focused on indoor (1-30 m) scenes, while our **SpaceVista model and dataset span scales from mm (1e-3 m) to km (1e+3 m)**. Dotted lines represent our contribution in filling the gap. This six-order-of-magnitude range introduces not only scale variation but also rich semantics and diverse tasks. SpaceVista enables all-scale spatial reasoning by integrating cues from micro-objects to macro-scenes.

et al., 2017; Yeshwanth et al., 2023) as shown in Fig. 2(b). These indoor datasets often feature relatively simple scenes and depend on manual 3D annotations. Scaling up to build large-scale, wild datasets encompassing video scenes ranging from $mm$ to $km$ presents two major challenges: 1) the **high cost** of large-scale annotation from complex and wild scenes; 2) the difficulty in obtaining **precise evaluations** that align with the physical world. To address these challenges, we use an automated pipeline leveraging popular specialized models to generate structured training data across 5 different scales. Since different scales have distinct characteristics and applications, we define several scale-specific tasks for better application, i.e., manipulation planning and area estimation. Overall, we provide over 1 million QA pairs across 19 diverse tasks from around 38K wild video scenes. To adapt to different stages of training, we provide both answers with rationale for SFT and regression/multiple-choice answers for RL. To facilitate accurate evaluation, we collect a highly accurate SpaceVista-Bench through manually recording or retrieving authoritative sources, supplemented with human annotations.

Most popular reasoning models are optimized for indoor settings, which leads to clear limitations: their responses

**often deviate** significantly, in tabletop and other diverse real-world scenes illustrated in Fig. 2(a). We address this by first injecting SpaceVista-1M knowledge to fine-tune existing models with the self-supervised visual encoder to make compensation for the classic semantic visual tokenizer, enabling extra geometry-based and depth-based spatial understanding. However, naive fine-tuning rarely yields optimal results, largely due to **cross-scale conflicts** between scenes and objects based on our observation. To address this, we introduce LoRA-like scale experts that cooperates with a scale router during fine-tuning. Moreover, to strengthen the model's ability to learn scale-centric spatial reasoning processes, we design a training strategy that uses scale as an anchor for progressive rewards. During evaluation, SpaceVista-7B shows superior understanding of spatial layout, size, and comparison, delivering a clear improvement on popular benchmarks and SpaceVista-Bench.

Our key contributions with this comprehensive solution are:

- Developing an automated pipeline to create a diverse, real-world, all-scale reasoning dataset, **SpaceVista-1M**, with 1M QA pairs across 5 scales and 19 tasks (including specific-scale tasks), and supporting both SFT with rationale and high-quality RL.

- Introducing **SpaceVista-7B**, a spatial reasoning model that integrates rich spatial information and employs scale experts with a customized training strategy to alleviate potential cross-scale conflicts during all-scale finetuning.

- Hand-crafting **SpaceVista-Bench**, an accurate video benchmark spanning all scales, by measuring and recording real-world objects, retrieving authoritative sources, and performing human annotation.

## 2. Related Works

**Visual Reasoning.** Currently, vision-based general reasoning has seen diverse developments (Tan et al., 2025; Wang et al., 2025b; Qiao et al., 2025; Wu et al., 2025a). General MLLMs (Wang et al., 2025c; Bai et al., 2025) first provided the basic understanding ability towards video to the community. Pioneering works (Feng et al., 2025; Liao et al., 2025) started to provide reasonable rewards during model training using Group Relative Policy Optimization (GRPO) for the reasonable Chain of Thought (CoT). Then, visual reasoning (Li et al., 2025c; Chen et al., 2025; Liu et al., 2025c) was considered from broader perspectives, ranging from data to training structure. In general video reasoning, spatial claims are generally divided into two categories: 2D plane-based spatial reasoning (Han et al., 2025; Zhou et al., 2025), and 3D space-based spatial reasoning (Wu et al., 2025c; Zheng et al., 2025). This paper primarily focuses on the latter. Although these general models have achieved a certain degree of spatial ability, spatial MLLM is still in its early stages.

**Spatial Reasoning.** Mainstream spatial reasoning models can be categorized based on input modalities into image (Ma et al., 2025; Liu et al., 2025b; Chen et al., 2024a), multi-image (Xu et al., 2025), multi-view (Li et al., 2025b; Wu et al., 2025b), video (Wu et al., 2025c; Zheng et al., 2025; Ouyang et al., 2025; Zhang et al., 2025a; Ghazanfari et al., 2025), and simulation (Li et al., 2025a; Tang et al., 2025; Zhang et al., 2025b; Wang et al., 2025d; Zhang et al., 2025d). Among these categories, video stands out as a challenging task due to the difficulty of data acquisition and modeling. As the first work in spatial reasoning, VSI-Bench (Yang et al., 2025a) introduced a video-based benchmark that removes linguistic shortcuts and evaluated MLLMs on spatial tasks such as counting, direction, and planning, highlighting substantial performance gaps compared to humans. InternSpatial (Deng et al., 2025b), SPAR (Zhang et al., 2025c), and SpaceR (Ouyang et al., 2025) enriched spatial supervision through extensive QA pairs spanning indoor and other limited settings. Qi et al. (2025) used the bird-view map to aid overall understanding. Then, Spatial-MLLM (Wu et al., 2025c), VG-LLM (Zheng et al., 2025), and VLM-3R (Fan et al., 2025) adopted geometry-aware dual encoders to capture geometry cues and inferred occluded structures from monocular inputs. Additionally, spatial reasoning on

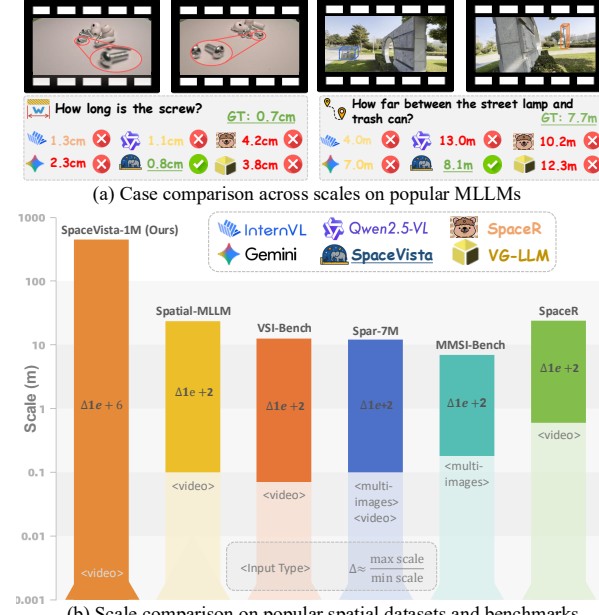

(a) Case comparison across scales on popular MLLMs

(b) Scale comparison on popular spatial datasets and benchmarks

*Figure 2.* **Case performance and dataset distribution across scales.** Current models and datasets necessitate all-scale spatial reasoning.

long (Zhang et al., 2025a), omni (Dongfang et al., 2025), ego-centric (Wu et al., 2025e) and aerial video (Zhang et al., 2025a; Sun et al., 2026) were also explored separately. However, the systematic data and designed model with all-scale video scenes remain unexplored.

**All-Scale Exploration.** The challenge of multi-scale in early years lay in information loss within low-resolution image patches (Zhao, 2025; Nikouei et al., 2025), which has almost no effect on spatial reasoning. In this paper, "all-scale" primarily concerns the real scales of the physical world, including distances, semantics, and object states across different scales. Deng et al. (2025a) pushed the limits of 3D perception and reconstruction from meters to kilometers; Wen et al. (2025) extended metric depth estimation from close range to infinity; and Liu et al. (2025a) curated uncommon objects, ranging from screws to airplanes, with object-centric annotations. Together, these developments underscore the need for AI to move beyond simple single-scale memorization toward robust, multiscale, and reasonable visual understanding.

## 3. Dataset

Due to high labeling cost, Tab.1 and Fig.2 show the clear drawback of the previous datasets. The limited data and performance constraints in existing models necessitate the creation of a dataset with all-scale spatial context. We propose **SpaceVista-1M**, a diverse, real-world, all-scale reasoning dataset, as the **first** to the best of our knowledge. SpaceVista-1M primarily comprises diverse spatial reasoning question–answer pairs, with rich semantic (category,

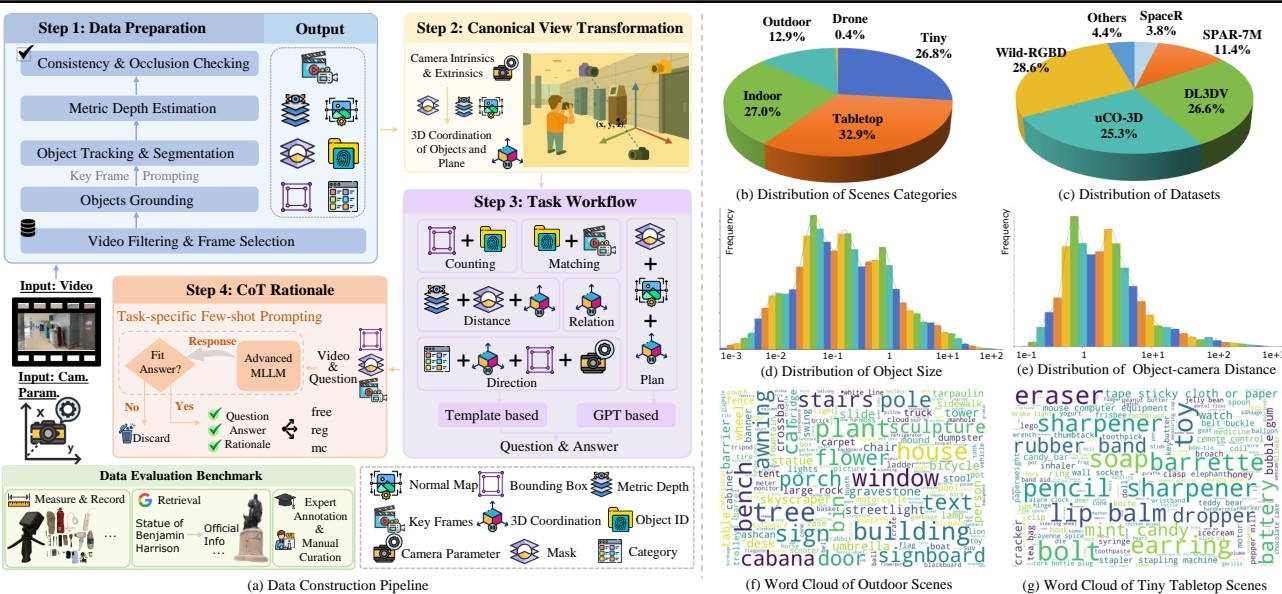

*Figure 3.* Fig.(a) shows our **automated data construction pipeline**. The pie charts (b-c) depict the **composition of scenes and sources**. The bar charts (d–e) show object sizes ranging $mm$-$100m$, while object-to-camera distances typically span $10$-$600m$. Accordingly, we claim SpaceVista-1M basically covers the $mm$-$km$ scale. The word clouds (f-g) provide a glimpse of the scene diversity.

rationale), 2D (mask, box, point), and 3D (depth, camera parameters, point cloud) annotations, obtained either natively or through processing. The construction pipeline in Fig. 3 follows the step-by-step procedure of preparing, transforming, and generating to obtain an all-scale dataset by integrating specialized models.

**Data Preparation.** We begin by selecting widely used video datasets that provide 3D scene modeling (Ling et al., 2024; Xia et al., 2024; Park et al., 2020; Liu et al., 2025a; Dai et al., 2017; Yeshwanth et al., 2023) along with camera intrinsic and extrinsic parameters. Most of these sources are videos of static scenes without moving objects. Leveraging the known camera parameters, we estimate depth maps and normal maps using specialized metric depth models (Hu et al., 2024; Piccinelli et al., 2025) and video depth models (Chen et al., 2025). For semantic understanding, we extract per-frame semantics and bounding boxes using proprietary grounding specialists (Ren et al., 2024; Liu et al., 2023). To establish cross-frame object consistency, by further integrating SAM 2 (Ravi et al., 2024) with the previously mentioned grounding experts, we enable robust object ID association and mask generation. This pipeline ensures both semantic and spatial consistency across frames.

**Task Construction.** With the help of official camera parameters and the preparations mentioned above, we can obtain the positions and dimensions of target objects. As a common practice (Deng et al., 2025b), we adopt a canonical view space of the reference frame, defined as a 3D Cartesian coordinate system centered at the camera's optical center. We then design 19 tasks and their corresponding workflows, even including scale-specific tasks such as tabletop object manipulation and drone-view area estimation. Taking object

counting as an example, the workflow follows: detect objects, propagate masks across frames, track identities over time, filter out scenes with camera parameters and ambiguous objects, and derive temporally consistent counts. For each task, we obtain the data by using similar, carefully designed computational workflows.

**QA Construction.** The pipeline for constructing the QA data is shown in Fig. 3. At the construction level of QA, we employ two strategies: GPT-based and template-based. For relatively fixed questions such as counting and object size, we adopt a template-based approach to obtain reasonable QA pairs. To ensure the diversity of the questions, we man-

*Table 1.* **Comparison of popular spatial reasoning datasets.** Only spatial reasoning QA is included. Lower QA/Scene Ratio usually means more diverse language and visual scenes. "free","reg", and "mc" mean free-form, regression, and multiple-choice, respectively. SpaceVista-1M does not differentiate QA pairs by the type; i.e., the semantically similar questions with reg/mc/free answers are counted only once. The citations here are listed in Appendix §B.5 for conciseness.

| Usage | Dataset | Type | QA Pairs↑ | Video Scenes↑ | QA/Scene Ratio↓ |
|---|---|---|---|---|---|
| Train | SpaceR | reg/mc | 191K | 1.2K | 159 |
| | SPAR-7M | reg/mc/free | 7M | 4.5K | 1,556 |
| | Spatial-MLLM | reg/mc/free | 120K | 1.5K | 83 |
| | InternSpatial | free | 2.5M | 5.5K | 455 |
| | SpaceVista-1M (Ours) | free/reg/mc | 1M | 38K | 25 |
| Benchmark | TempCompass | mc | 7.5K | 0.4K | 18 |
| | VideoMME | mc | 2.7K | 0.9K | 3 |
| | All-Angles | mc | 2.1K | 90 | 23 |
| | VSI-Bench | reg/mc | 5.0K | 0.3K | 17 |
| | MMSI-Bench | mc | 1.0K | - | - |
| | SPAR-Bench | reg/mc | 7.2K | - | - |
| | STI-Bench | mc | 2.0K | 0.3K | 7 |
| | SpaceVista-Bench (Ours) | reg/mc | 1.6K | 0.3K | 5 |

ually curate over 3,000 templates. However, for more flexible questions like planning, we use a GPT-based (OpenAI, 2025) method to generate reasonable answers in natural language. Additionally, through appropriate randomizing and prompting, we obtain multiple options to serve as rewards for RL.

**CoT Annotation.** To facilitate an efficient cold start, we follow Feng et al. (2025) to leverage cognition-inspired few-shot prompting strategy with Qwen2.5-VL-72B-Instruct (Bai et al., 2025) to generate CoT rationales. After employing the filtering policy for low-quality or inconsistent rationale outputs, we obtain the CoT for SpaceVista-1M, with high-quality rationale for fundamental knowledge injection for SFT.

**Input Extension.** Usually, people refer to objects in videos using more than just language. To support this, we extend video-based QA with extra annotations from the video's key frames. Besides plain visual input, we allow three extra inputs: point, bounding box, and mask, which may support future interactive usage. Each input type is designed to fit its own template and CoT rationales.

**Quality Control & Evaluation.** 1) SpaceVista-1M (training): To ensure data quality, we manually verify a small portion of the **training set** for quality control, achieving an average global accuracy of approximately 83%. 2) SpaceVista-Bench (evaluation): We choose the reliable pathway for **benchmark** based on measuring and recording real-world data, retrieving authoritative sources, and performing human annotation for both distance and non-distance problems, shown in the green block Fig.4(a). For tiny and tabletop scenes, we capture and annotate videos of over 50 objects of different sizes. For some indoor and outdoor scenes, we search for the landmarks and retrieve statistics from authoritative sources like Wikipedia. As for other tasks like camera moving, the experts are hired for checking and annotating. By aligning the answer with the physical world, SpaceVista-Bench comprises about 1,600 QA pairs across approximately 300 unique video scenes, with quality ensured through manual review and source verification. Due to the scarcity of public data, the exploration in drone scenarios remains preliminary.

**More Information on Dataset and Benchmark**: We encourage readers to consult the appendix for more details.

- Source investigations in Sec. § B.2.
- Benchmark collection in Sec. §B.2.7.
- Data preparation in Sec. §B.3.1.
- Tasks and workflows in Sec. §B.3.
- In-depth distribution analysis in Sec. §B.4.
- Data quality control in Sec. §B.4.2.
- License in Sec. §B.4.3.

In summary, we propose SpaceVista-1M, an open-source, real-world, all-scale dataset with spatial video QA. SpaceVista-1M contains 1 million QA pairs spanning 19 tasks, 5 scale types, and over 50 subscene categories.

## 4. Method

**Overview.** Our objective is to enhance spatial reasoning by elaborately designing and conditioning the model on explicit and detailed **all-scale information**. We first utilize a dense, expressive self-supervised encoder beyond semantics to strengthen the model's overall spatial perception. However, mixing different types of knowledge without distinction hinders, rather than facilitates, the model's reasoning in Fig. 4(a-d), a problem known as **knowledge conflict**. In all-scale reasoning, this conflict appears when similar visual patterns are interpreted differently at different scales. To mitigate such conflict, we propose a LoRA-like scale expert architecture to maintain the independence of scale-level knowledge, while maintaining parameter efficiency, as shown in Fig 4(e). Finally, drawing on human reasoning about scale, we introduce reward-based progressive reasoning paths that employ essential anchors to constrain the reasoning process to a reliable CoT path.

**Preliminaries.** The number of frames is first denoted as $T$ with the temporal patch size $\tau$. The visual representations from Qwen-2.5-VL visual encoder are denoted as $F_V \in \mathbb{R}^{t \times d_V \times H \times W}$, where $t = \frac{T}{\tau}$ is temporal dimension of the feature, $d_V$ is the feature dimension per patch, and $H$ and $W$ are the numbers of patches $p$ along the height and width of each frame, respectively. Then, each $i \in t \times d_V$ of $F_V$ is directly converted to an image token $T_V^i$ as input.

**Beyond Semantics.** Most open-sourced MLLM tokenizers including Qwen-2.5-VL visual encoder are pretrained on semantically rich text–image pairs via contrastive training, and thus often lack a well-formed understanding of information beyond semantics. Meanwhile, El Banani et al. (2024); Tong et al. (2024b;a) draw a valuable conclusion that self-supervised vision models, such as DINO series, learn rich depth, normal, and pattern representations. Therefore, leveraging popular DINOv3 (Siméoni et al., 2025)'s strong dense features seems to be a natural approach beyond simple semantics. The last layer of DINOv3 produces patch-level dense features $F_D \in \mathbb{R}^{T \times d_D \times H_D \times W_D}$. We pad and regularize the original image to align with the patch size $p$, enforcing $H_D = H$ and $W_D = W$. We then apply a simple MLP, $\mathbb{R}^{d_D} \to \mathbb{R}^{d_V}$, to map channel dimensions. For the temporal dimension, we use the same temporal pooling with the previously mentioned temporal patch size $\tau$ to aggregate across $T$, yielding features $F_D' \in \mathbb{R}^{t \times d_V \times H \times W}$. The fusion of the video feature $F_V$ and dense feature $F_D'$ is shown as:

$$F_V' = \mathrm{CA}(F_V, F_D', F_D') + F_V \qquad (1)$$

where $\mathrm{CA}(q, k, v)$ denotes multi-layer cross-attention over

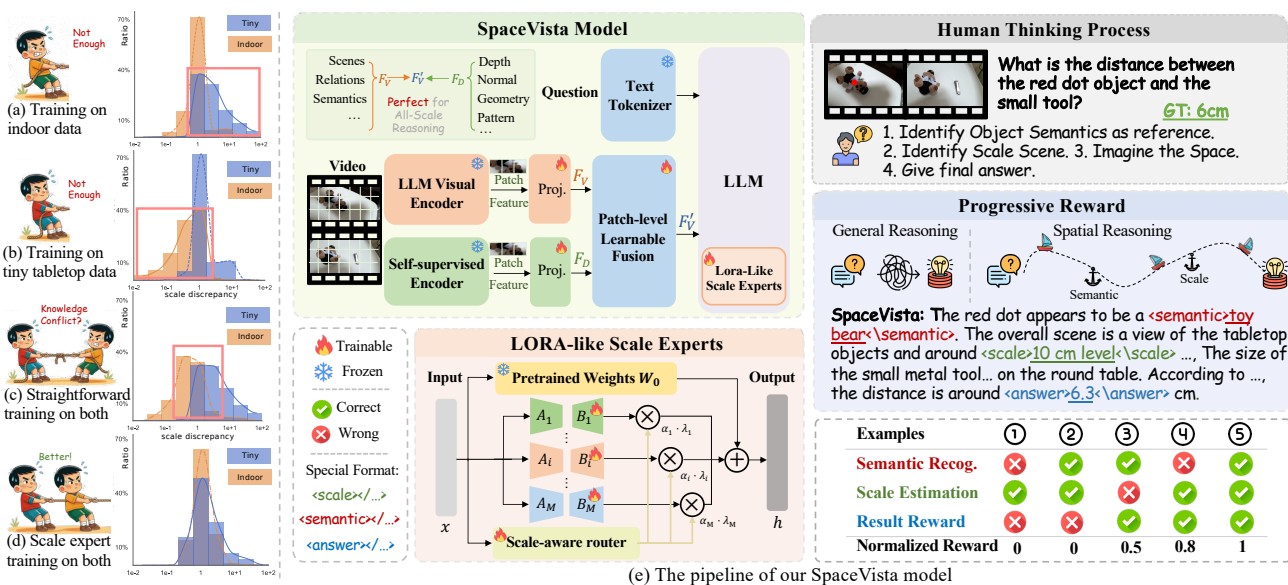

**Figure 4.** The left part (a-d) shows that the undifferentiated **mixture of cross-scale knowledge hinders**, rather than facilitates, the model's **reasoning process**. The horizontal axis represents the scale discrepancy, defined as $\frac{answer}{gt}$ (=1 for the ideal situation), and the vertical axis denotes the proportion of answers. Fig.(e) is our SpaceVista model, where "`<think>`" is omitted for clarity.

the query, key, and value inputs. Then, we convert $F_V'$ into a fused image token $T_V^i$, and the remaining calculations proceed as before.

**Scale Experts Design.** During all-scale mixed training in Fig.4(a-d), potential cross-scale knowledge conflicts lead to suboptimal results. This underscores the importance of preserving knowledge independence between scales during training. Inspired by Wu et al. (2024); Buehler & Buehler (2024); Chen et al. (2024b), we further introduce a LoRA-like module that adds scale experts by fine-tuning only 0.5% of the overall parameters for each expert. The original LoRA is using $B \in \mathbb{R}^{d \times r}$ and $A \in \mathbb{R}^{r \times d}$ with the rank $r \ll \min(d, k)$ to approximate original weights $W_0$. To construct scale LoRA experts, We attach $M$ scale experts $\{(A_i, B_i)\}_{i=1}^M$ to mitigate potential scale-level knowledge interference. Each expert $i$ has a base weight $\alpha_i$ and is dynamically scaled by a learned factor $\lambda_i$:

$$h = W_0 x + \sum_{i=1}^M \alpha_i^* B_i A_i x, \text{where } \alpha_i^* = \alpha_i \cdot \lambda_i, \quad (2)$$

where $x, h$ are the input and output of the projection layer, and $\alpha_i^*$ is the scaled factor. The learned factor $\lambda_i$ is obtained through a scale router, primarily an MLP and a softmax. We apply $M$ scale experts to each layer of the foundation LLM. Therefore, different layers, according to their respective conditions, obtain appropriate $\lambda_i$ to allocate the experts within the layer. Given that scenarios of scales can overlap (for example, an indoor scene may include some tabletop context), in the ideal case, the routers can select the suitable experts at different layers.

**Process Reward Design.** After basic SFT training, RL is used to align the model with human perception. Inspired by

how humans approach spatial observation tasks, we model the reasoning process explicitly. Humans typically proceed by: 1) identifying the task-specified semantics (if they help), 2) perceiving the global scale by inspecting surrounding objects (if it helps), and 3) inferring the answer from spatial relations. Following this paradigm, we construct 3 different anchors for RL that enforce the reasoning path to traverse the resulting anchor states. With the minimal, sufficient ground-truth anchors, we design the following three reward components based on these anchor formats: `<semantics>`, `<scale>`, and `<answer>`. Semantic reward $R_{\text{semantic}}$ is used to identify the referenced objects; Scale reward $R_{\text{scale}}$ is used to estimate the scale of the overall scene; Correctness reward $R_{\text{answer}}$ is used to ensure the answer is well derived. The updated correctness reward $\bar{R}_{\text{answer}}$ can be formed into

$$\bar{R}_{\text{answer}} = \sum_{k=1}^3 \prod_{n=1}^k R_{j_n},$$

$$\text{where} \quad R_{\text{scale}} = \max\left(0, 1 - \frac{|\log C_{\text{ans}} - \log C_{\text{gt}}|}{2}\right), \quad (3)$$

$$R_{\text{semantic}} = \frac{S_{\text{ans}} S_{\text{gt}}}{\|S_{\text{ans}}\|\|S_{\text{gt}}\|},$$

with $(j_1, j_2, j_3) = (\text{answer}, \text{scale}, \text{semantic})$ $C_{\text{ans}}, C_{\text{gt}}$ is the estimated scene scale in the same measurement; $S_{\text{ans}}, S_{\text{gt}}$ is the calculated semantic embedding. $C_{\text{gt}}$ and $S_{\text{gt}}$ can be easily obtained from Sec.§3. It is crucial to note that the order of $(j_1, ..., j_n)$ matters; rewards at the beginning are stricter and more important. Also, because tasks differ, for example in the camera rotation task, $R_{\text{semantic}}$ and $R_{\text{scale}}$ are not needed. Thus, $\bar{R}_{\text{answer}}$ under such circumstances collapses to a standard $R_{\text{answer}}$. The calculation of format reward $R_{\text{format}}$ and answer reward $R_{\text{answer}}$ remains the same

*Table 2.* **Performance comparison across five spatial reasoning benchmarks.** Open-sourced general models are evaluated with a comparable size. The highest performance of the open-sourced model is marked **bold**.

| | Multi-Image | | | Video | |
| Model | MMSI-Bench | SPAR-Bench | VSI-Bench | STI-Bench | SpaceVista-Bench |
|---|---|---|---|---|---|
| Human | 97.2 | 67.3 | 79.2 | - | 81.3 |
| *Closed-Sourced Commercial Models & 70B-Class Models* | | | | | |
| GPT-5 (OpenAI, 2025) | 40.7 | 37.4 | 44.2 | 39.3 | 33.7 |
| Gemini-2.5-pro (DeepMind, 2025) | 36.9 | 36.3 | 45.0 | 41.4 | 33.8 |
| InternVL3.5-38B (Wang et al., 2025c) | 36.9 | 31.0 | 66.3 | 39.2 | 30.7 |
| Qwen2.5-VL-72B (Bai et al., 2025) | 30.7 | 32.4 | 30.7 | 40.7 | 31.1 |
| *Open-Sourced General Models* | | | | | |
| LLAVA-Onevision-7B (Li et al., 2024a) | 24.5 | 30.6 | 32.4 | 29.0 | 13.6 |
| LLaVA-NeXT-Video-7B (Liu et al., 2024a) | 26.8 | 31.3 | 35.6 | 29.9 | 23.7 |
| InternVL3.5-8B (Wang et al., 2025c) | 30.9 | 36.0 | 38.2 | 33.2 | 24.5 |
| Qwen2.5-VL-7B (Bai et al., 2025) | 31.7 | 33.1 | 32.7 | 32.1 | 28.9 |
| *Open-Sourced Specialized Models* | | | | | |
| SpaceR-7B (Ouyang et al., 2025) | 26.1 | 37.6 | 46.9 | 37.0 | 21.2 |
| SpatialMLLM-4B (Wu et al., 2025c) | 27.0 | 31.5 | 48.4 | 30.5 | 24.2 |
| VILASR-7B (Wu et al., 2025d) | 30.2 | 37.6 | 45.4 | 31.5 | 23.6 |
| VG LLM-4B (Zheng et al., 2025) | - | - | 46.1 | 29.3 | 28.8 |
| Qwen2.5-VL-7B $w/.$ SpaceVista-1M | 27.3 | 36.9 | 42.0 | 35.0 | 29.5 |
| SpaceVista-7B (Ours) | 29.1 | 38.1 | 46.3 | 35.9 | 35.4 |
| SpaceVista-7B (Ours) $w/.$ RL | **32.3** | **41.6** | **48.6** | **38.2** | **39.2** |

as common practice (Feng et al., 2025; Guo et al., 2025) to encourage the generation of valid and executable answers. Therefore, our reward design forms the accurate reward signals to ensure all-scale spatial compliance and encourage human-like thinking. It is worth noting that the evaluation does not involve these anchors besides the actual answer.

**RL Training Objective.** For each question $i$, we define the reward $R_i$ to include both the updated correctness reward $\bar{R}_{\mathrm{answer}}$ and $R_{\mathrm{format}}$ following Guo et al. (2025), and use this overall reward $R_i$ to compute groupwise normalized advantages $A_i = \frac{R_i - \mathrm{mean}(\{R_j\})}{\mathrm{std}(\{R_j\})}$. $\{R_j\}$ is the response group related to $R_i$. The final policy $\pi_\theta$ is updated by maximizing

$$\mathbb{J}(\theta) = \mathbb{E}_{q,\{o_i\}} \left[ \frac{1}{G} \sum_{i=1}^{G} \min \left( \frac{\pi_\theta(o_i \mid q)}{\pi_{\theta_{\mathrm{old}}}(o_i \mid q)} A_i, \quad \mathrm{clip}\left( \frac{\pi_\theta(o_i \mid q)}{\pi_{\theta_{\mathrm{old}}}(o_i \mid q)}, 1 - \epsilon, 1 + \epsilon \right) A_i \right) - \beta \, \mathbb{D}_{\mathrm{KL}}(\pi_\theta \parallel \pi_{\mathrm{ref}}) \right], \quad (4)$$

where $\pi_{\theta_{\mathrm{old}}}$ and $\pi_\theta$ are the old and new policy model respectively. $\mathbb{D}_{\mathrm{KL}}$ represents KL divergence.

**Training Strategy.** We start with a cold-start phase on SpaceVista-1M, optimizing the input projection, feature-fusion modules, and scale experts. Next, we introduce the scale router to further train each scale-specific expert on the appropriate inputs, encouraging specialization. Finally, building on the SFT model, we apply RL training to obtain the final SpaceVista-7B reasoning model.

## 5. Experiment

**Datasets.** We use SpaceVista-1M in Sec.§ 3 for SFT and RL; its sources are detailed in Appendix.§ B.2.

**Model Configurations.** Our model is built on Qwen2.5-VL-7B for main experiments and Qwen2.5-VL-3B for ablation. Our model is trained on up to 16 NVIDIA A800 (80GB) GPUs. We process a maximum of 32 frames during training, each with a resolution of $128 \times 28 \times 28$ pixels. During inference, we increase the resolution ($256 \times 28 \times 28$ pixels) to enhance performance. During the expert training phase, we employ 4 experts, each tailored to a distinct scenario. We set the group size of GRPO to 8. The training data already includes four scale and scene labels from the construction process. These labels are used to divide the data, and each expert is trained on the data that matches its scale. We first perform SFT on CoT data of SpaceVista-1M for two epochs to obtain the SFT model. This is followed by RL training for 2.5k steps to produce the final SpaceVista-7B checkpoints with detailed information shown in Appendix.§ C.1. The semantic embeddings are computed using all-MiniLM-L6-v2 with the Sentence-Transformer Toolkit.

**Comparison on Spatial Reasoning Datasets.** Our method attains competitive performance across all spatial reasoning benchmarks in Tab. 2. On VSI-Bench, we achieve comparable results approaching the state of the art. More importantly, our approach delivers substantially superior performance in our all-scale benchmark SpaceVista-Bench, markedly exceeding 3% compared with proprietary and open-source models. Thus, SpaceVista-1M represents a robust baseline for both indoor and all-scale scenes.

*Table 3.* **Ablation study on 3B-based SpaceVista model trained in the same condition.** We analyze the impact of different modules (top) and extra input modalities (bottom).

| Setting | VSI-Bench | SpaceVista-Bench |
|---|---|---|
| Vanilla SFT | 44.4 | 31.0 |
| Vanilla SFT $w/.$ RL | 44.9 | 32.1 |
| *Modality Ablation $w/.$ SFT* | | |
| $w/.$ VGGT | 44.3 (-0.1) | 31.4 (+0.4) |
| $w/.$ DINOv3 | 46.4 (+2.0) | 32.1 (+1.1) |
| $w/.$ VGGT + DINOv3 | 45.3 (+0.9) | 31.7 (+0.7) |
| *Module Ablation $w/.$ RL* | | |
| $w/.$ Scale | 46.3 (+1.4) | 34.8 (+2.7) |
| $w/.$ Scale + Semantic | 46.8 (+1.9) | 35.4 (+3.3) |
| $w/.$ Expert Finetuning | 45.8 (+0.9) | 34.8 (+2.8) |

*Table 4.* **Ablation of the number of experts** based on the same SFT settings on the 3B model. $M = 4$ is the default setting.

| Num of Expert(s) $(M)$ | Training Data (Each Expert) | VSI-Bench | SpaceVista-Bench (Ours) |
|---|---|---|---|
| None | All | 44.4 | 31.0 |
| 1 | All | 44.2 (-0.2) | 31.0 (0) |
| 2 | 1/2 | 46.1 (+1.9) | 33.1 (+2.1) |
| 4 | 1/4 | 46.3 (+2.1) | 34.7 (+3.7) |
| 6 | 1/6 | 43.1 (-1.1) | 26.7 (-4.3) |

**Benchmarks.** We evaluate our model on 5 benchmarks, VSI-Bench , STI-Bench , SpaceVista-Bench (Ours), MMSI-Bench and SPAR-Bench . Among the benchmarks, the former three are video-based, while the latter two are multi-image benchmarks. We argue that video and multi-image tasks share rather strong similarities and collectively serve as benchmarks for cross-frame understanding. For all evaluations, we follow the configuration used in the official Qwen demo with $\text{top}_p = 0.001$ and temperature $= 0.01$.

**Comparison on Spatial Reasoning Datasets.** Our method attains competitive performance across all spatial reasoning benchmarks in Tab. 2. On VSI-Bench, we achieve comparable results approaching the state of the art. More importantly, our approach delivers substantially superior performance in all-scale benchmark SpaceVista-Bench, markedly exceeding 3% on average compared with proprietary and open-source models. Thus, SpaceVista-1M represents a robust baseline for both indoor and all-scale scenes.

**Comparison on Subsets of SpaceVista-Bench.** In Tab.5, we analyze the performance of popular models on each subset of our SpaceVista bench. In general, the small-scale sub-sets challenge both commercial and general models, likely due to biases in the pretraining corpus. We also observe that most models perform at a relatively low level on SpaceVista-Bench, indicating that it has the expected discriminative ability for all-scale reasoning and can serve as a foundational benchmark to help the community enrich the overall evaluation ecosystem. Our SpaceVista-7B, although exhibiting minor improvements on indoor scenes, attains comparatively high comprehensive scores across other scenarios

*Table 5.* **The SpaceVista-Bench leaderboard**. We utilize green (1st), blue (2nd), and yellow (3rd) backgrounds to distinguish the top three results within each scene. We employ **bold** and underlined text to denote the best and second-best results across all open-source models. All the baselines are instruction-tuned and are evaluated on the same resolution and fps. The citations here are listed in Appendix §B.5.

| Models | SpaceVista-Bench | | | | |
|---|---|---|---|---|---|
| | Tiny | Tabletop | Indoor | Outdoor | Overall |
| *Closed-sourced Commercial Models* | | | | | |
| 🥉 GPT-5 | 32.3 | 20.3 | 39.0 | 43.0 | 33.7 |
| GPT-4o | 21.7 | 13.3 | 34.3 | 38.3 | 26.9 |
| 🥈 Gemini-2.5-pro | 33.0 | 38.7 | 34.5 | 29.0 | 33.8 |
| Gemini-2.5-flash | 20.7 | 30.0 | 19.9 | 26.9 | 24.4 |
| Claude-Sonnet-4 | 27.3 | 19.3 | 38.1 | 34.1 | 29.7 |
| Claude-Opus-4.1 | 21.7 | 29.5 | 24.3 | 30.0 | 26.4 |
| *Open-Source General Models* | | | | | |
| Internvl3.5-38B | 29.3 | 25.2 | 41.2 | 27.0 | 30.7 |
| Internvl3.5-14B | 27.7 | 22.3 | 31.3 | 24.3 | 26.4 |
| Internvl3-78B | 38.3 | 23.3 | 42.2 | 30.3 | 33.5 |
| Internvl3-38B | 18.7 | 14.3 | 34.8 | 38.0 | 26.5 |
| GLM-4.5V | 23.0 | 17.8 | 27.3 | 25.2 | 23.3 |
| GLM-4.1V-Thinking | 30.7 | 19.3 | 29.0 | 13.3 | 23.1 |
| Qwen2.5VL-72B | 27.7 | 20.3 | 29.6 | 28.0 | 26.4 |
| Qwen2.5VL-32B | 25.3 | 19.3 | 38.1 | 30.7 | 28.4 |
| LLAVA-Onevision-72B | 25.0 | 12.0 | 15.3 | 11.7 | 16.0 |
| LLAVA-Onevision-7B | 17.5 | 8.0 | 13.3 | 11.6 | 12.6 |
| *Open-Source Specialized Models* | | | | | |
| SpaceR | 12.9 | 17.3 | 34.9 | 19.8 | 21.2 |
| Spatial-MLLM | 17.3 | 20.3 | 36.1 | 23.1 | 24.2 |
| VLM-3R | 15.1 | 24.6 | 45.1 | 26.9 | 27.9 |
| 🥇 SpaceVista-7b (Ours) | 35.3 | 38.2 | 44.1 | 39.1 | 39.2 |

and in overall evaluations. The results indicate a clear boost of around 6% compared with any size of the open-source models in comprehensive all-scale spatial reasoning.

**Ablation on Each Component.** 1) Scale Expert: We examine how potential information conflicts during cross-scale training are mitigated. As shown in Tab.3, the experts yield substantial gains. The ablation on the number of experts in Tab. 4. However, more routers are not always better; increasing the number of experts places greater pressure on the router and leads to load imbalance. 2) Reward: In Tab. 3, the progressive reward achieves higher performance than the unconstrained reasoning path. These optional anchors indeed serve as a valuable halfway point in the all-scale reasoning process. This highlights the importance of specifying thinking anchors when designing all-scale reasoning. 3) Modality: As shown in Tab. 3, incorporating DINO v3 yields greater gains than VGGT with its advantage of semantically dense cues.

**More Experiments.** We provide more visualization, experiments, and analysis in the appendix, for example,

- Patch-level encoder ablation in Sec. § C.2;
- All-Scale observation and insights in Sec. §C.5, C.6;
- Out-of-distribution analysis on customized data including Guinness Records in Sec. §C.4 and Fig. C18.

# 6. Conclusion

In this work, we introduce a novel task for all-scale reasoning from visual spatial context, which requires the machine to understand multimodal information and respond with the correct answer and rationale. To advance this field, we develop the first open-source, all-scale, spatial reasoning dataset, SpaceVista-1M, for cold start and reinforcement learning. Then, we handcraft SpaceVista-Bench, an accurate, multi-scale, video-based benchmark that adheres to physical world measurements and perceptions. During experiments, we compare our SpaceVista-7B model with popular models and demonstrate our proposed model's promising performance as a robust baseline in all-scale reasoning.

# Acknowledgement

This work is partially supported by the National Natural Science Foundation of China (No. 62306261), HK RGC-Early Career Scheme (No. 24211525), ITSP Platform Project (No. ITS/600/24FP) and the SHIAE Grant (No. 8115074). This study is supported in part by the Centre for Perceptual and Interactive Intelligence, a CUHK-led InnoCentre under the InnoHK initiative of the Innovation and Technology Commission of the Hong Kong Special Administrative Region Government. This work is partially supported by Hong Kong RGC Strategic Topics Grant (No. STG1/E-403/24-N), and CUHK-CUHK(SZ)-GDST Joint Collaboration Fund (No. YSP26-4760949). This work is also supported by Astribot Inc. Special thanks to Xinran Chen for her crucial and diligent data collection.

# Impact Statement

We anticipate that SpaceVista will catalyze widespread advancements across diverse domains by enabling robust all-scale spatial reasoning. Beyond enhancing core capabilities like spatial captioning, guided visual generation, and interactive world models, its impact extends to critical real-world applications ranging from the micro-scale—such as precision manufacturing ($\mu m$) and medical surgery ($mm$)—to the macro-scale of remote sensing ($km$) and cartography ($10km$). Ultimately, this work provides a foundational step for industrial automation, embedded systems, and autonomous driving, empowering intelligent agents to perceive and reason complex, unconstrained environments in the wild.

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

# Appendices Contents

# A. Important Information

## A.1. Task Distribution

Our SpaceVista-1M consists of a wide range of tasks, including both general tasks and scale-specific tasks. Fig. A5 illustrates the data composition for each scene task, where bubble sizes indicate the relative data volume.

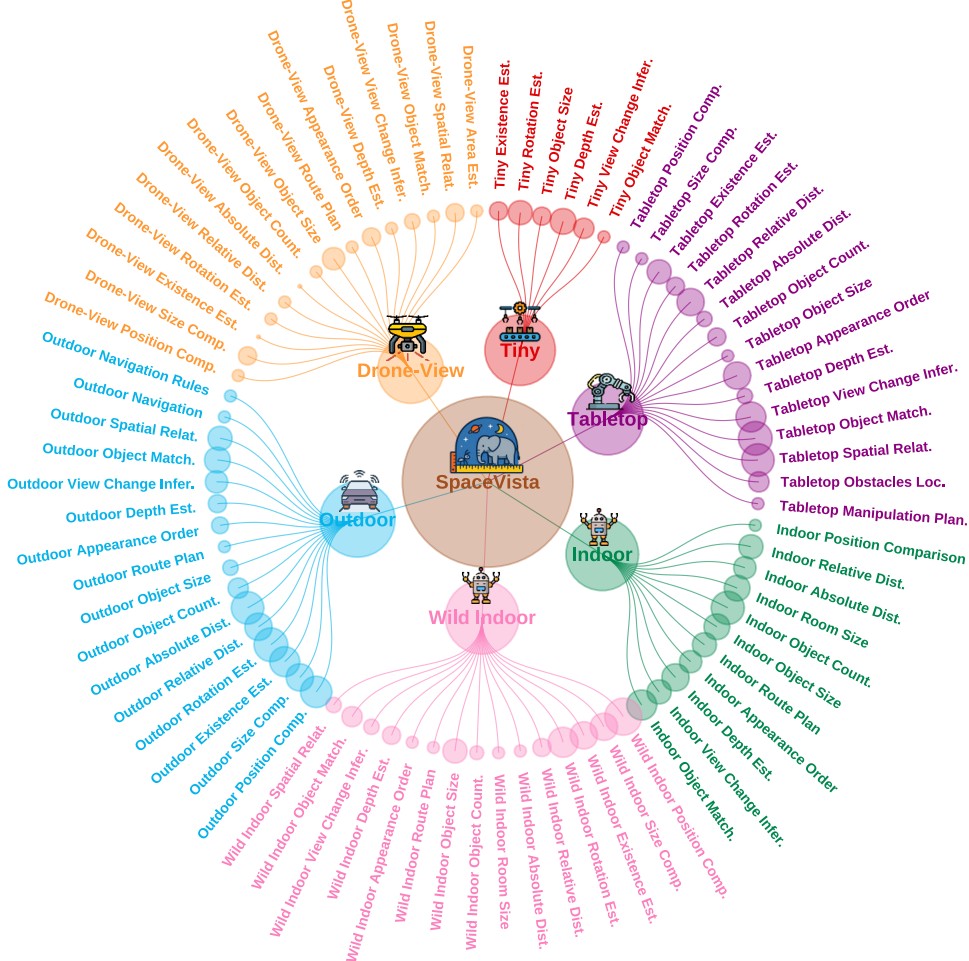

*Figure A5.* **Statistical chart of QA types.** The spatial reasoning tasks for various scenes include abbreviations, for example, "Est." for Estimation, "Dist." for Distance, "Loc." for Location, and "Com." for Comparison.

## A.2. Performance Radar

The comparison across models is carried out on multiple spatial reasoning benchmarks. We evaluate eight multimodal large models on five distinct benchmarks, with the results visualized in the radar chart in Fig. A6.

SpaceVista-7B achieves significant improvement across the benchmarks, highlighting its superiority in spatial reasoning tasks. While models, including LLAVA-Onevision-7B (Li et al., 2024a), demonstrate competitive performance, SpaceVista-7B consistently exhibits superior robustness and adaptability across a range of tasks, thereby solidifying its position as a robust model in spatial reasoning.

# B. Data Construction

Our SpaceVista-1M dataset spans 19 spatial reasoning task types, including scale-specific tasks, comprising 1 million QA pairs and 38 thousand videos collected across diverse scenes. This scale and variety enable large-scale training of perceptual understanding and spatial reasoning, and support comparative analysis across tasks and environments.

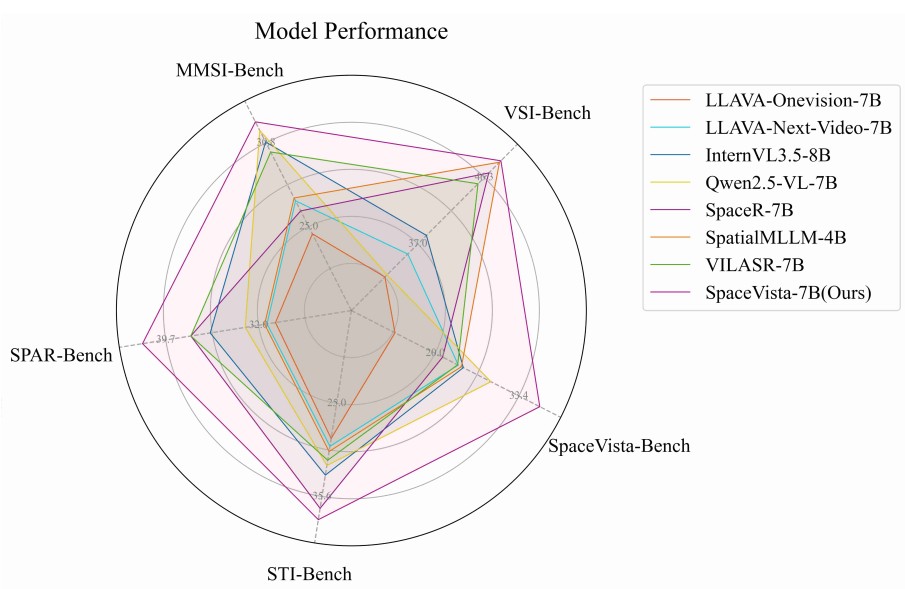

*Figure A6.* **Performance comparison across popular spatial reasoning benchmarks.** Our SpaceVista-7B model achieves certain performance boosts across all benchmarks.

This chapter details the data sources for each scene category (Sec.§ B.2), the end-to-end task construction pipeline (Sec.§ B.3.1), and key dataset statistics (Sec.§ B.4).

### B.1. Data Comparison

*Table B6.* **The datasets we used to build SpaceVista-1M and SpaceVista-Bench.** "†" means the datasets are only used for evaluation in SpaceVista-Bench. "‡" means data collected by us and used for accurate evaluation. The definition of scenes is the number of unique spaces, and one scene can be transformed into multiple questions.

| Dataset | Type | Scenes |
|---|---|---|
| uCO3D(Liu et al., 2025a) | Tiny, Tabletop | 10,000 |
| WildRGB-D(Xia et al., 2024) | Tabletop | 11,300 |
| SMOT(Park et al., 2020) | Tabletop | 13 |
| SpaceR(Ouyang et al., 2025) | Indoor | 1,500 |
| Spar-Bench(Zhang et al., 2025c) | Indoor | 4,500 |
| Scannet Series(Dai et al., 2017; Yeshwanth et al., 2023) | Indoor | 460 |
| VSI-Bench†(Yang et al., 2025a) | Indoor | 288 |
| MMSI-Bench†(Yang et al., 2025b) | Indoor | 231 |
| DL3DV(Ling et al., 2024) | Drone, Indoor, Outdoor | 10,510 |
| STI-bench†(Li et al., 2025d) | Indoor, Outdoor, Tabletop | 372 |
| Our own collected data ‡ | Tiny, Tabletop, Outdoor | 500 |

Our current dataset encompasses a broad diversity of scene categories, as summarized in Tab. B6. The data sources span a wide range of scenarios, including tiny, tabletop, indoor, outdoor, and drone-view.

To ensure evaluation quality and robustness, we apply multiple rounds of processing and rigorous filtering to all collected data. We remove redundant or inconsistent samples across datasets. Because scenes may overlap across sources, which can compromise the independence of the training and test splits, we removed from the training set any scene that appears in all the benchmarks. This strict separation prevents leakage and enables a fair assessment of generalization. Consequently, the SpaceVista-1M provides broad scene diversity, with a clean, reliable benchmark SpaceVista-Bench.

### B.2. Data Source

Sec.§ B.2 presents data sources that form our dataset, and systematically describes the provenance and acquisition of seven scene sources. These sources combine multiple public datasets and our own collected data, as detailed in Sec.§ B.2.1- B.2.7.

These scenes span object-centric through scene-level contexts and exhibit substantial variation in scale, shape, pattern, and illumination.

When building the dataset, our foundational data construction process must adhere to the following key criteria:

- **Video Data with 3D Modeling**: The data must consist of video sequences accompanied by either official or third-party 3D modeling. This enables effective use of camera parameters for robust data processing.

- **Multi-Frame & Multi-Scale**: The dataset should support meaningful spatial reasoning across multiple frames and scales. Its complexity must be sufficient to prevent trivial single-frame assessments from representing the full sequence.

- **Comprehensive Annotations & Metadata**: Each sample must include the following: (a) camera intrinsics and extrinsics, (b) detection and segmentation labels, and (c) dense depth maps. These elements support a broad range of downstream tasks.

### B.2.1. TINY TABLETOP SCENE

We curate small-scale, small-object videos from uCO3D (Liu et al., 2025a), selecting sequences where the object size falls below a predefined threshold to instantiate the tiny tabletop scenario. uCO3D comprises approximately 170,000 high-resolution, object-centric 360-degree videos captured via crowdsourcing, covering more than 1,000 LVIS (Gupta et al., 2019) categories grouped into 50 categories. For each video, uCO3D applies VGGSfM (Wang et al., 2024) for motion analysis and 3D Gaussian Splatting to generate accurate camera poses, depth maps, sparse and dense point clouds, and semantic captions. The resulting subset contains everyday small objects, such as stationery, food, and decorative items, placed on flat surfaces such as tables, counters, and shelves. These scenes provide complete viewpoint coverage, precise geometry, and rich semantic labels, which make them well-suited for fine-grained 3D object modeling and spatial video reasoning. Here, we only select a small part of uCO3D for around 10,000 videos for tiny objects after filtering.

### B.2.2. TABLETOP SCENE

For tabletop scene modeling, we select two datasets: WildRGB-D (Xia et al., 2024) and SMOT (Park et al., 2020). WildRGB-D consists of approximately 8,500 objects across 46 categories, recorded in around 20,000 RGB-D videos, with iPhones rotating 360 degrees around objects to replicate real-world interactions. It includes single-object, multi-object, and hand-occlusion videos, all automatically annotated via SLAM-generated camera poses and reconstructed point clouds, making it suitable for spatial reasoning tasks. To select samples for spatial reasoning, we specifically choose around 10,000 videos with multiple objects in a scene. SMOT (Park et al., 2020) is a challenging small dataset collected by a mobile robot, comprising 13 video sequences.

The tabletop, commonly referred to as the "table" scene, encompasses not only the planar surface of a table but also extends to various other surfaces, including sand, beds, wardrobes, floors, and similar environments. In combination, these datasets offer richly varied planar scenes, providing a robust foundation for challenging spatial video reasoning benchmarks.

### B.2.3. INDOOR SCENE

Indoor scenes are among the earliest domains studied in spatial video reasoning. Key datasets, including ScanNet (Dai et al., 2017) and ScanNet++ (Yeshwanth et al., 2023), collect RGB-D scans using handheld cameras, yielding aligned RGB images, depth maps, and 3D reconstructions. ScanNet contains more than 1,500 scenes and 2.5 million frames spanning common indoor spaces, such as offices and bedrooms, with annotations for over twenty object categories. ScanNet++ extends this setting with higher geometric fidelity and more complex layouts. The combination of focused object classes, structured environments, and rich annotations makes these datasets central benchmarks for spatial reasoning.

### B.2.4. WILD INDOOR SCENE

Beyond scan-based indoor modeling, DL3DV (Ling et al., 2024) adopts a video-based pipeline that replaces active scanning with video capture and camera parameter estimation. Building on this framework, and further compressed using 3D Gaussian Splatting (Chen et al., 2024c), DL3DV enables high-precision 3D reconstruction of wild indoor scenes. The dataset covers a broad range of object categories, including challenging reflective and transparent instances. Compared with conventional scan-based datasets, these scenes exhibit greater geometric and appearance variability, providing a more realistic and demanding benchmark for spatial video reasoning.

### B.2.5. OUTDOOR SCENE

In addition to tabletop and indoor scene modeling, DL3DV (Ling et al., 2024) collects extensive in-the-wild outdoor videos encompassing landmarks, street corners, private courtyards, and urban parks. Camera parameters are calibrated using COLMAP (Schönberger et al., 2016; Schönberger & Frahm, 2016). The DL3DV-10K dataset includes 10,510 videos in 4K resolution, totaling about 51.2 million frames, covering 65 types of locations. Each video is annotated for whether it is indoors or outdoors as well as for levels of reflection, transparency, and lighting conditions. Compared to conventional scan-based indoor datasets, these outdoor scenes exhibit richer geometric complexity, greater diversity of materials, and wider environmental variation, offering more challenging benchmarks for spatial video reasoning.

### B.2.6. DRONE SCENE

DL3DV (Ling et al., 2024) extends outdoor scene modeling by incorporating drone-captured videos that provide aerial perspectives to complement ground level views. Videos are recorded using unmanned aerial vehicles (UAVs), and camera parameters are calibrated through COLMAP (Schönberger et al., 2016; Schönberger & Frahm, 2016), following the same reconstruction pipeline applied to handheld footage. The DL3DV Drone subset consists of more than 100 videos covering a variety of scenes, including open plazas, tree-lined pathways, rooftop platforms, and landmark facades. DL3DV enhances spatial video reasoning by introducing unique geometric structures and varied viewpoints.

Although the data scale is not as large as tabletop or indoor, the drone-view scenes establish a more rigorous benchmark for aerial mapping and spatial video reasoning by expanding scene diversity and viewpoint range.

### B.2.7. OUR OWN COLLECTED DATA

The data collection methods described above rely on advanced specialized models and fully automated pipelines. While we incorporate limited manual filtering, whether the resulting data can be used as an accurate evaluation of real-world perception is still a question. This limitation motivates our collection of higher-fidelity data to better align with physical world perception.

Our dataset consists of two types: 1) measured, recorded, and manually annotated data, and 2) existing video data enhanced by retrieving and verifying publicly available information. The former is suitable for tiny objects, tabletop objects, whereas the latter is designed for indoor and outdoor scenarios.

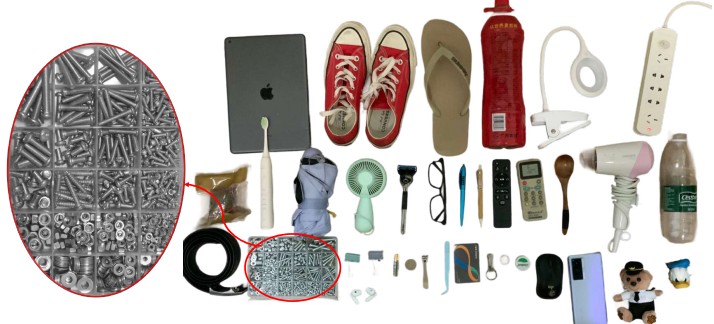

*Figure B7.* **The objects involved in our hand-crafted dataset.** Our self-collected data features various categories of objects, with tabletops and tiny tabletops ranging from 0.4m to 3mm, even including transparent and reflective objects.

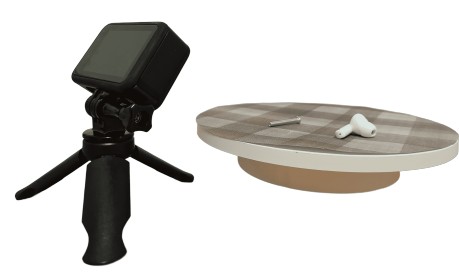

*Figure B8.* **A photo of the real scene** for the collection of tiny tabletop.

**Data from self-recording and measurement.** Precise spatial annotations (e.g., location and dimensions) are scarce in existing datasets such as uCO3D and WildRGB-D. To address this, we captured length and positional data for nearly 50 object categories across diverse scenarios. Using GoPro 11, iPhone 15, and Vivo X70, we systematically varied object arrangements, distances, lighting conditions, and backgrounds into over 200 videos and 1,000 QA pairs. As illustrated in Fig. B7 and Fig. B8, they show the objects used for self-collected data and a real scene of tiny tabletop data collection. Although we collected the raw high-resolution videos up to 2.7K/60fps, it is still necessary to resize and resample it for better comparison. The resulting measurements are consolidated into a unified perceptual space that closely approximates physical world geometry.

**Data Retrieved from authoritative sources.** Adopting a similar rationale, it is apparent that spatial information derived

solely from wild videos lacks the precision required for robust evaluation. Consequently, alternative methodologies must be explored. To address this, we propose a systematic approach that first identifies landmark objects within existing datasets and then manually retrieves images of these objects from authoritative sources, such as Wikipedia[1], architectural drawings, and official design documents, to obtain accurate spatial information, as shown in Fig. B9. This method ensures that the evaluation data is not only more precise but also more consistent with human perceptual judgments and preferences.

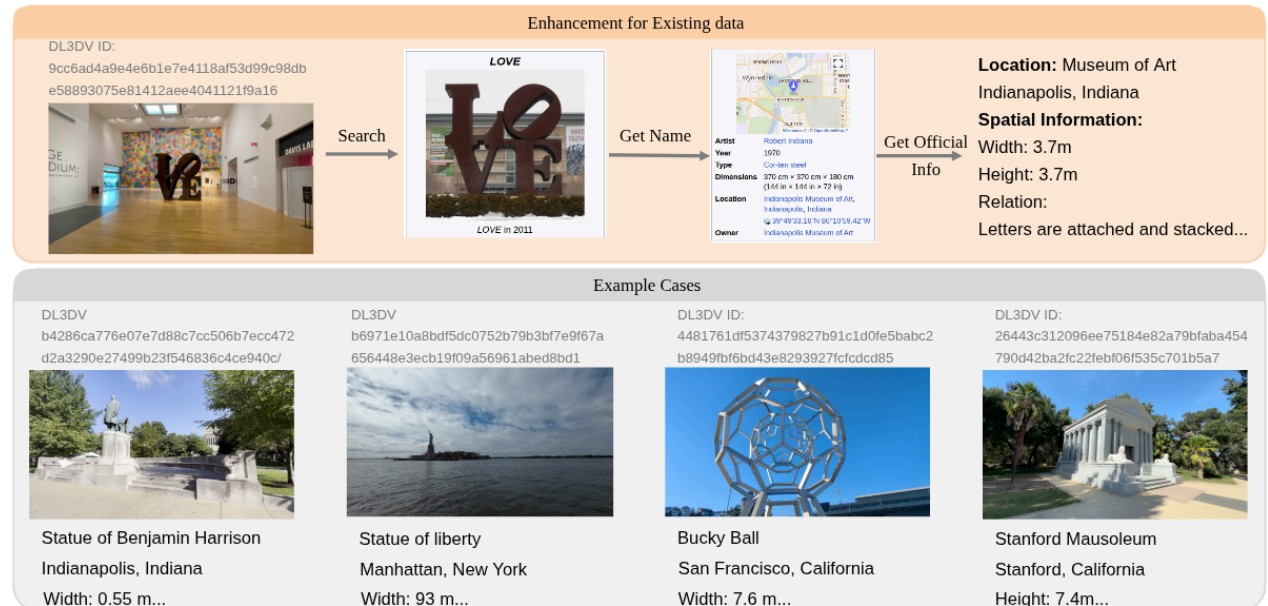

*Figure B9.* **Examples of identifying outdoor landmark objects** from existing datasets and retrieving their scale-related ground truth data.

### B.3. Task Construction

Upon acquiring the appropriate dataset, we initially perform necessary data preparation and processing in Sec.§ B.3.1. Subsequently, we carefully design workflow for each task (Sec.§ B.3.3-B.3.5), and we present detailed task explanations in Tab. B7. The final output consists of high-quality QA pairs, facilitating the cold-start and reinforcement learning processes of MLLMs.

### B.3.1. DATA PREPARATION

Previous popular approaches, such as InternSpatial (Deng et al., 2025b), required estimating camera intrinsic and extrinsic parameters, which introduced cumulative errors that propagated through subsequent tasks. However, since we exclusively utilize datasets with known camera parameters (as detailed in Sec B.2), our framework operates under conditions close to ground truth.

We first employ Metric3Dv2 (Hu et al., 2024) and UniDepthV2 (Piccinelli et al., 2025) to obtain accurate metric depth maps and normal maps. The metric depth maps provide precise distance measurements between the camera and scene objects, while the normal maps facilitate robust plane estimation. There are two challenges during construction. **1) Video consistency**: According to observation, the metric depth model may not have that level of consistency across frames. So, we use Video-Depth-Anything (Chen et al., 2025) to ensure consistency by minimizing the energy function,

$$D^* = \underset{D}{\arg\min} \left\{ \|D - M\|_F^2 + \lambda \|\nabla_t(D) - \nabla_t(N)\|_F^2 \right\}, \tag{5}$$

where $M$,$N$ represent metric depth model maps and Video-Depth-Anything map . **2) Extreme Scale**: Although the metric depth model is trained on the datasets as DDAD (Guizilini et al., 2020) and NYUv2 (Silberman et al., 2012), it may have a certain level of adaptation to the extreme situations. For extreme situations, including drone-view and tiny objects, it is still necessary to provide a prerequisite to adjust the depth normalization accordingly.

---

[1]https://www.wikipedia.org/

*Table B7.* **Detailed explanation of 19 tasks included in SpaceVista-1M.**

| Task | Description |
|---|---|
| *General Indoor Scenes* | |
| Position Comparison | Compare the positions of two objects within or across frames, assessing their spatial relationships in terms of left/right, above/below, and near/far. |
| Size Comparison | Compare the positions of two objects within or across frames, involving three pairs of size relationships: wider/thinner, taller/shorter, larger/smaller. |
| Existence Estimation | Determine whether there are objects across frames whose positional/size relationships with the specified object meet the constraint conditions. |
| Object Counting | Estimate how many objects meet the constraint conditions across frames. |
| Rotation Estimation | Estimate the rotation angle of an object across multiple frames. |
| Absolute Distance | Estimate the closest distance between two objects within or across the frames. |
| Object Size | Estimate the longest dimension of an object within or across the frames. |
| Route Planning | Choose what action should be performed between a sequence of actions within or across the frames in order to route from a start point to a target. |
| Appearance Order | Given a video, determine the $N$-th appearance order of several objects. |
| Depth Estimation | Estimate the relative or absolute distance of objects from the camera viewpoint in a single image or across multiple images. |
| View Change Inference | Infer how the camera viewpoint has changed (position and orientation) across the video frames. |
| Object Matching | Determine whether two objects in the beginning and end frames of a video are the same physical object instance or different instances of the same object type. |
| Spatial Relation | Analyze and describe the spatial relationships (e.g., support, hanging, adhesion, stacking, encircling, plug-in) between multiple objects or cameras across the frames. |
| *Indoor Scenes* | |
| Every Type in General | All task types from Indoor Scenes can be applied to drone-view perspectives. |
| Room Size | Estimate the volume of the room(s) across the frames. |
| *Outdoor Scenes* | |
| Every Type in General | All task types from Indoor Scenes apply to Outdoor Scenes except for Room Size estimation. |
| Navigation | Determine the optimal path or movement strategy to navigate from one location to another across different views (similar to the Route Planning mentioned in Indoor Scenes). |
| *Drone-View Scenes* | |
| Every Type in General | All task types from Indoor Scenes can be applied to drone-view perspectives. |
| Route Plan | Given a series of aerial images, choose what action should be performed between a sequence of actions in order to route from a start point to a target (similar to the Route Planning mentioned in Indoor Scenes). |
| Area Estimation | Estimate the size or area of regions or objects from an aerial perspective. |
| *Tabletop Scenes* | |
| Every Type in General | All task types from Indoor Scenes can be applied to drone-view perspectives. |
| Object Location | Determine the precise position of objects on a table surface, typically corresponding to other objects. |
| Destination Location | Identify target positions related to single objects (i.e. left, right, front ...) as part of manipulation planning. |
| Obstacles Location | Identify and locate objects with the AABB box that may interfere with manipulation as part of manipulation planning. |
| Manipulation Planning | Determine the sequence of actions needed to rearrange objects or achieve a specific configuration on the table. |

For fine-grained semantic understanding at the pixel level, we leverage the advanced proprietary model DINO-X (Ren et al., 2024) to extract semantic information and bounding boxes for complex scenes, while relying on Grounding DINO (Liu et al., 2023) for simpler samples. To address cross-frame consistency challenges in video data, we integrate the aforementioned grounding models with SAM2's (Ravi et al., 2024) advanced tracking capabilities, generating temporally consistent masks and unique object IDs across frames based on Grounded-SAM2[2].

By this stage, we obtain a comprehensive understanding of each frame, including bounding boxes, masks, categories, and object IDs, laying a solid foundation for downstream task formulation.

### B.3.2. TYPE: DISTANCE

The distance-related tasks, including object size, room size, object distance, and relative distance, rely on depth maps and computer vision techniques to measure object and spatial dimensions from monocular images. The method converts 2D depth keypoints into 3D point clouds using camera calibration parameters and applies Principal Component Analysis (PCA) to extract dimensional information, focusing on objects larger than 20×20 pixels. For object size estimation, the system segments visible objects using instance masks and projects the masked depth values into 3D space. PCA determines the principal axes of the point cloud, with height measured along the vertical axis and width derived from the convex hull of points projected onto the dominant plane. Relative distances are calculated by comparing 3D centroids in world coordinates, and room dimensions are estimated by analyzing the spatial distribution of depth points and identifying major planar surfaces corresponding to walls.

The method uses camera intrinsics and extrinsics to express all measurements in a consistent world coordinate system, addressing the scale ambiguity of monocular systems. Multiple frames are processed to improve robustness, with temporal averaging reducing noise in the estimates. The technique assumes piecewise rigid scenes, operates on standard RGB images, and produces metric-scale measurements. Accuracy depends on the quality of depth estimation and segmentation. Overall, it demonstrates how 2D computer vision pipelines can be extended to 3D measurement tasks through precise geometric reasoning.

### B.3.3. TYPE: COUNTING

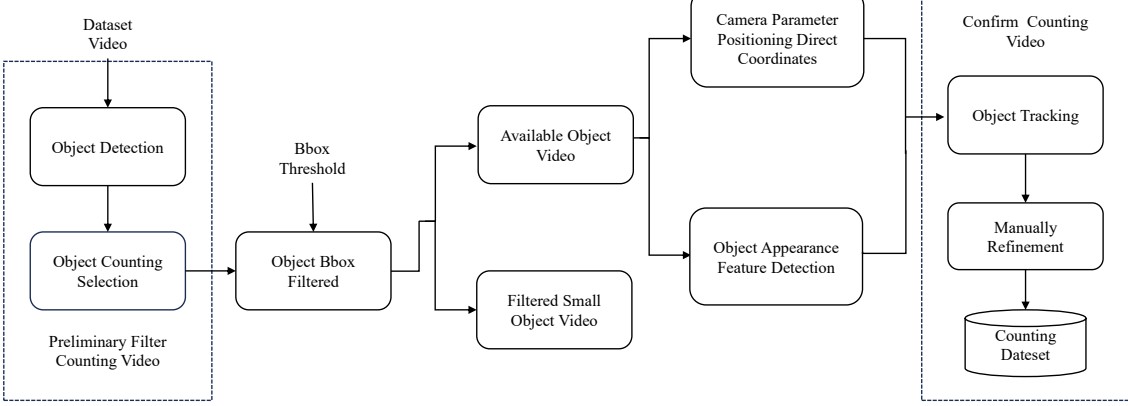

*Figure B10.* **Automatic Processing Pipeline for Counting Task Scenes.** Through data filtering, object tracking, and counting, the final counting video is obtained after data confirmation.

Object counting across real-world scenes faces diverse visual conditions and a high cost of manual labeling, which motivates an automatic pipeline that adapts to scene type. The automatic pipeline addresses object counting through two methodologies tailored to specific scenarios, and Fig. B10 illustrates the workflow that maintains high accuracy while reducing manual effort across indoor, outdoor, and tabletop scenes. For outdoor video sequences, the open-vocabulary detection model (Ren et al., 2024; Cheng et al., 2024) uses text prompts with a confidence threshold of 0.3 for zero-shot detection, projects 2D observations into 3D world coordinates to enforce spatial consistency, and tracks objects via motion prediction with confirmation after at least ten consistent detections. Given the difficulty of reliably detecting very small objects in outdoor scenes and to mitigate ID switching and trajectory fragmentation under severe occlusions, scenes are prefiltered to those

---

[2]https://github.com/IDEA-Research/Grounded-SAM-2

containing 2 to 10 objects with a minimum bounding-box size of 32 pixels. For tabletop scenarios, grounding model (Ren et al., 2024; Liu et al., 2023) and SAM2 (Ravi et al., 2024) are employed, where open-vocabulary detection uses text and bounding box thresholds of 0.4, and mask propagation applies IoU and center distance thresholds of 0.4 and 32 pixels, respectively, to distinguish instances. Both methodologies output object categories and their corresponding counts for each video.

### B.3.4. TYPE: PLANNING

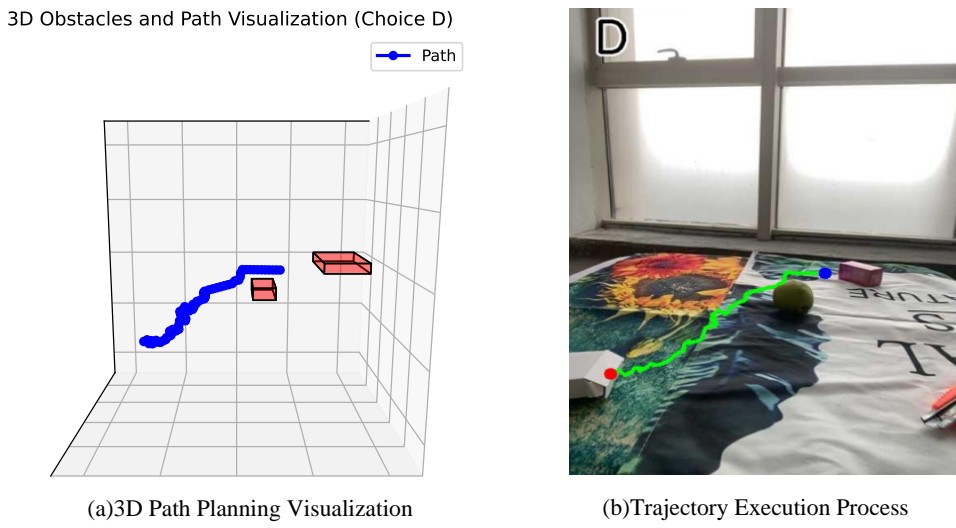

(a)3D Path Planning Visualization  (b)Trajectory Execution Process

*Figure B11.* **Visualization of robotic manipulation planning.** Fig.(a) visualizes the option for moving the red box to the left of the upper box. Fig.(b) represents the key frame to carry out the manipulation.

In robotic manipulation tasks, effective route planning is essential to ensuring smooth and accurate object movement. The route planning pipeline proceeds as follows. First, depth information and object detection are utilized to identify the category, position, shape, and size of all objects within the image. Subsequently, an arbitrary object is selected as the manipulation source and another as the target position, with the objective being to relocate the source object to a designated position (e.g., front, back, left, right, or above) relative to the target object. Based on this configuration, an LLM generates corresponding manipulation instructions, such as *"What is the correct route of placing the apple on the box"*. Next, the actual spatial positions of the objects are computed using both intrinsic and extrinsic camera parameters. The Rapidly-exploring Random Tree (RRT) (LaValle, 1998; Xu, 2024) algorithm is then employed to plan a collision-free path, where the bounding boxes of objects serve as obstacle constraints during path computation. Finally, two types of data are generated from the planned path: 1) multiple paths are projected onto the camera plane, with the correct trajectory serving as the ground truth answer, and 2) the coordinate variations along the path are translated into natural language instructions via the LLM. For instance, when the x-coordinate of the object decreases while the y-coordinate remains constant in the camera space, the LLM produces the instruction *"move the object to the left."* Fig. B11 demonstrates the visualization of robotic manipulation under the option, showing the planned movement of the red box to the left of the upper box. This figure highlights the spatial relationship and intended positioning within the manipulation task.

### B.3.5. TYPE: RELATION

In spatial relation analysis, we combine semantic information with 3D positional data through an automatic reasoning process to ensure consistency in both semantic and spatial aspects. Our analysis operates primarily at the semantic level. We first identify and extract common candidate relations, such as support, attach, insert, and surround. Based on the consistent 3D keypoint semantics established earlier, we generate potential relation pairs that may exhibit these spatial relationships. These candidate pairs are then evaluated for spatial plausibility by integrating 3D positional data with the few-shot prompt through Chain-of-Thought (CoT) reasoning using the foundation model. Finally, the validated pairs are processed by GPT for transformation and answer generation, ensuring semantically and spatially consistent outputs.

### B.3.6. DATA POST-PROCESSING

To address the cold-start challenge in SFT, we prioritize the acquisition of explicit "thinking process" rationales—step-by-step explanations that clarify how answers are derived. For example, in object counting, the model is prompted to articulate intermediate reasoning (e.g., *"there are 2 cups on the table and 3 on the chair, totaling 5"*), enriching task understanding and facilitating more robust generalization.

Following common practice (Feng et al., 2025), we acquire high-quality rationales by distilling from advanced open-source and proprietary large models. Specifically, we use Qwen2.5-VL-72B and Gemini-2.5-Pro for complex tasks, and Qwen2.5-VL-32B for simpler ones, balancing reasoning depth with efficiency. We then compare these generated rationales and their corresponding answers with previously collected cases. When GPT answers are different from the answers from previous workflows, we apply a confidence-based filtering strategy to curate the training set, retaining only instances with consistent, well-supported reasoning. This pipeline generates a cleaner, rationale-augmented dataset, mitigating SFT cold-start effects and enhancing downstream performance.

### B.3.7. BENCHMARK CONSTRUCTION

Our benchmark comprises two components: **1) Measurement-Related.** For the scale-related portion requiring precise scale annotations, we collect approximately 300 videos across diverse scenes using the two methods described in Appendix B.2.7 and human annotation for other spatial tasks, covering tiny, tabletop, and outdoor settings. For the indoor evaluation set, we instead selected suitable data from ScanNet-based datasets (e.g., VSI-bench and SPAR-bench) and constructed a series of scale-focused questions on top of these bases. **2) Non-Measurement.** For the non-measurement questions, we manually annotate the data collected in the previous step to produce additional spatial reasoning QA pairs. In total, we curate about 1,600 fully human-annotated QA pairs for model evaluation.

## B.4. Data Statistics

From a visual perspective, our dataset comprises wild scenes spanning scales from millimeters to kilometers. Although the raw dataset contains over 100 million frames, we calculate unsupervised annotations as intermediate information at both the pixel and semantic levels for a curated subset of 10 million frames. These frames vary in resolution from 480p to 2.7K, with frame rates ranging from 24 to 30 fps. During data processing, we preserve the original resolution whenever possible and apply uniform sampling during training as needed.

In terms of the QA component, we employ a combination of templated generation and GPT-based methods to produce 1 million QA pairs with a theoretical duplication rate of only 0.0005%. These pairs are structured into diverse answer formats, including free-form, multiple-choice, and regression-based responses, catering to different analytical needs. Rigorous quality control measures are implemented, with detailed analyses provided in Sec.§ B.4.2.

### B.4.1. TARGET CATEGORY DISTRIBUTION

The introduction of diverse scenarios, such as tabletop, indoor, and outdoor, aims to establish a more inclusive object composition system. Due to the limited drone data, we incorporate drone-view data into the outdoor analysis. By approximating complex object distribution patterns to the real world, this approach enhances the scene adaptation capabilities of visual reasoning models. To quantitatively assess the impact of scene diversity on model generalization, we use the word cloud to compare object distribution characteristics across different scenarios, as shown in Figs. B12–B17. The results reveal that indoor scenes are predominantly composed of rigid objects such as furniture and electronics, exhibiting a highly structured spatial layout. In contrast, outdoor scenes feature more scale-varying objects like vehicles and natural landscapes, demonstrating spatial openness. Meanwhile, tabletop scenes focus on manipulable items such as tools and daily necessities, reflecting precise spatial arrangements. These cross-scene differences provide complementary training samples, effectively mitigating the risk of overfitting to specific scenarios. Thus, the necessity of a multi-scenario strategy to enhance cross-domain generalization is validated.

Overall, each subset scenario differs significantly from the previous indoor-dominated setting, highlighting the diversity of our scenes.

### B.4.2. DATA QUALITY CONTROL

During the construction of our dataset, we distinguish between two notions of answer assessment: **1) strict correctness**, which requires that an answer conform to objective physical reality, and **2) human preference**, which requires that an answer align with typical human judgments. Since strict correctness is difficult to establish for training data derived from in-the-wild videos (due to issues like missing calibration, occlusions, and limited metadata), we adopt the human-preference criterion

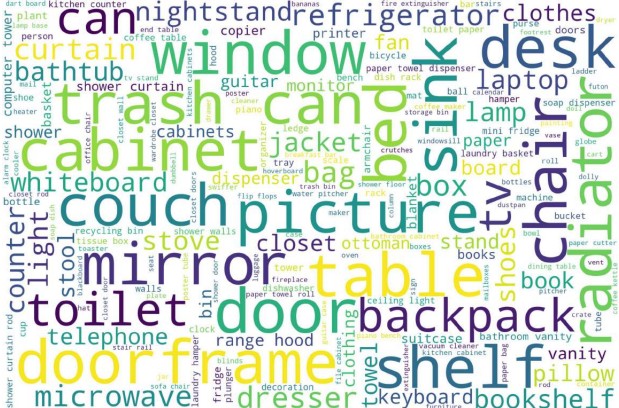

*Figure B12.* **The word cloud of the previous indoor spatial reasoning datasets.**

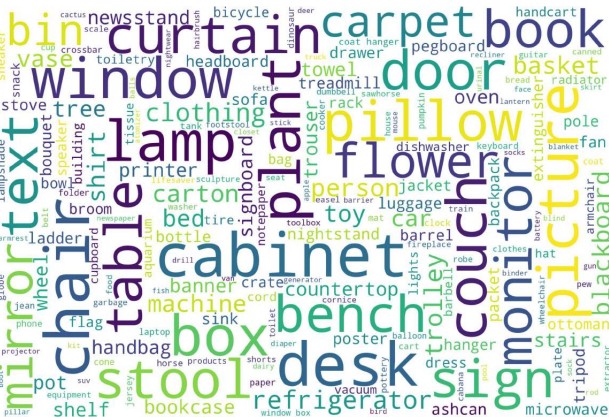

*Figure B13.* **The word cloud of our indoor subset.**

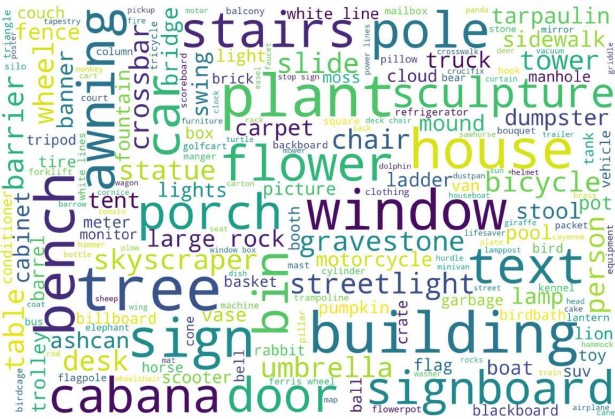

*Figure B14.* **The word cloud of our outdoor subset.**

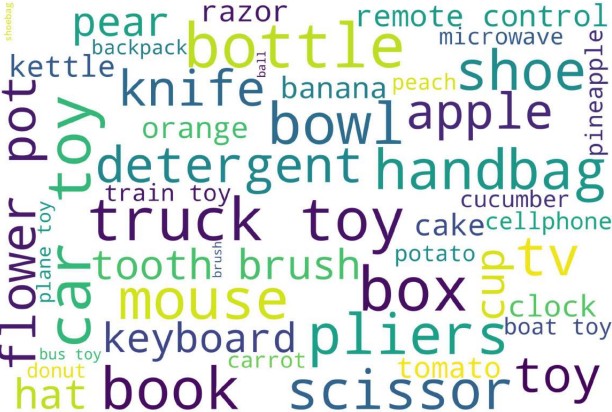

*Figure B15.* **The word cloud of our tabletop subset.**

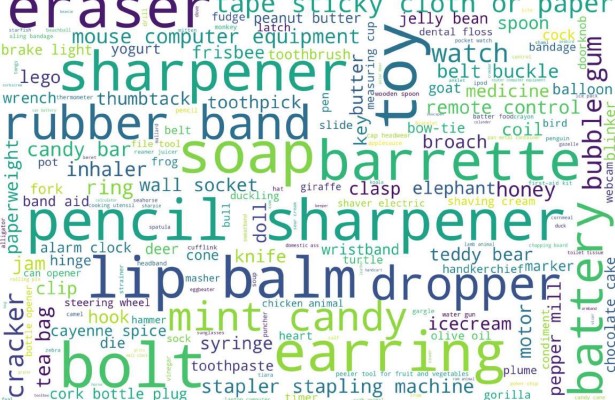

*Figure B16.* **The word cloud of our tiny tabletop subset.**

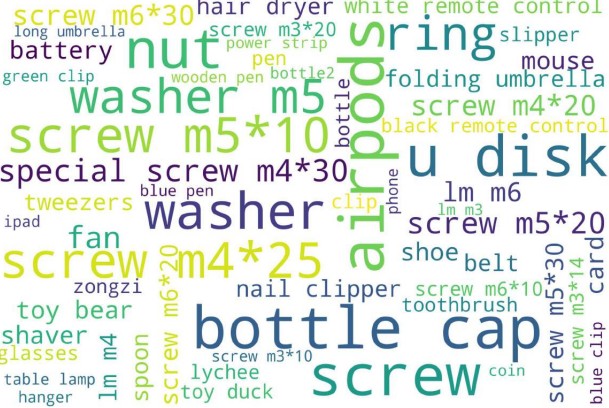

*Figure B17.* **The word cloud of the self-collected subset.** Note: We use standard ISO 7046 to denote the models of the screw, which looks like *"m4*10"*.

for training data quality control. Specifically, during validation, we present annotators with both the question and a candidate answer and ask them to judge whether the answer is acceptable from a human perceptual perspective. Consequently, the reported human preference rate should be interpreted as agreement with human perception rather than strict fidelity to physical-world quantities or metric scale. For these statistics and the user study, we use MTurk[3]. SpaceVista-1M human preference rates are shown in Tab. B8. SpaceVista-Bench is curated through source verification and manual review, while model evaluation follows strict correctness.

*Table B8.* **Human preference rate over each task category in our training set, SpaceVista-1M.** "∼" means we observe unusual variation for different annotators. The standards for Route Planning, Navigation, and Obstacle Avoidance are **notably stringent**, as these are inherently multi-step processes where a single error can invalidate an entire sample. However, even if a training sample contains minor discrepancies in step, sequence, or distance, the descriptive knowledge within the remaining sections remains valuable for comprehension.

| | Task Categories in SpaceVista-1M for training | | | | |
|---|---|---|---|---|---|
| **Task** | Position Comp. | Size Comp. | Existence Est. | Rotation Est. | Relative Dist. |
| **Human Preference Rate** | 95% | 84% | 94% | 95% | 82% |
| **Task** | Room Size | Object Count | Object Size | Route Plan | Appear. Order |
| **Human Preference Rate** | 84% | 87% | 81% | ∼69% | 80% |
| **Task** | View Change | Object Match | Spatial Rel. | Navigation | Area Est. |
| **Human Preference Rate** | 96% | 93% | 95% | ∼67% | 78% |
| **Task** | Manip. Plan | Absolute Dist. | Depth Est. | Obstacles | |
| **Human Preference Rate** | 73% | 84% | 95% | 74% | |

### B.4.3. LICENSE

We conduct a systematic review of the open-source licenses for the datasets we use, with the results summarized in Tab. B9. The analysis indicates that CC BY 4.0 and Apache License 2.0 are the most widely adopted. After comprehensive consideration, our SpaceVista-1M dataset adopts the **Creative Commons Attribution (CC BY) 4.0** or **Apache License 2.0** for different sources of data, which is already used by most of the source data.

*Table B9.* **The licenses for the dataset and benchmark** included in this paper.

| Dataset | Type | License |
|---|---|---|
| *Benchmarks* | | |
| VSI-Bench(Yang et al., 2025a) | Indoor | Apache License 2.0 |
| STI-bench(Li et al., 2025d) | Indoor | Apache License 2.0 |
| MMSI-Bench(Yang et al., 2025b) | Indoor | CC BY 4.0 |
| STI-Bench(Li et al., 2025d) | Outdoor, Tabletop | Apache License 2.0 |
| Spar-Bench(Zhang et al., 2025c) | Indoor | Apache License 2.0 |
| SpaceVista-Bench (Ours) | Tiny, Tabletop, Indoor, Outdoor | Apache License License 2.0 & CC BY 4.0 |
| *Training Datasets* | | |
| uCO3D(Liu et al., 2025a) | Tiny, Tabletop | CC BY 4.0 |
| SMOT(Park et al., 2020) | Tabletop | Unknown |
| WildRGBD(Xia et al., 2024) | Tabletop | None |
| SpaceR(Ouyang et al., 2025) | Indoor | CC BY-NC 4.0 |
| Scannet Series(Yeshwanth et al., 2023) | Indoor | ScanNet Terms of Use |
| DL3DV(Ling et al., 2024) | Indoor, Outdoor, Drone | DL3DV-10K Terms of Use |
| SpaceVista-1M (Ours) | Tiny, Tabletop, Outdoor | Apache License License 2.0 & CC BY 4.0 |

### B.5. Supplementary Citation

Due to the page limit, we have omitted some citations in Tab. 1. Here, we provide a supplementary table of citations.

## C. Model Detail

### C.1. Parameter Setting

**SFT.** The model architecture is based on Qwen2.5-VL-7B-Instruct, a 7-billion parameter vision-language model capable of processing both images (resized to 100,352 pixels) and videos (16,384 pixels at 16/32 frames). In the ablation study, we use the 3B model for efficiency. For fine-tuning, we employ a selective freezing strategy: while the vision tower and multi-modal

---

[3]https://www.mturk.com/

*Table B10.* **Supplementary citation** of Tab. 1 dataset comparison.

| Dataset | Citation | Dataset | Citation |
|---------|----------|---------|----------|
| SpaceR | Ouyang et al. (2025) | All-Angles | Yeh et al. (2025) |
| SPAR-7M | Zhang et al. (2025c) | MVBench | Li et al. (2024b) |
| Spatial-MLLM | Wu et al. (2025c) | VSI-Bench | Yang et al. (2025a) |
| InternSpatial | Deng et al. (2025b) | MMSI-Bench | Yang et al. (2025c) |
| Video-MME | Fu et al. (2024) | SPAR-Bench | Zhang et al. (2025c) |
| TempCompass | Liu et al. (2024b) | STI-Bench | Li et al. (2025d) |

*Table B11.* **Supplementary citation** of models in Tab. 5 SpaceVista-Bench leaderboard.

| Model | Citation | Model | Citation |
|-------|----------|-------|----------|
| GPT-5 | (OpenAI, 2025) | Internvl3-38B | (Zhu et al., 2025) |
| GPT-4o | (Hurst et al., 2024) | GLM-4.5V | (Team et al., 2025) |
| Gemini-2.5-pro | (DeepMind, 2025) | GLM-4.1V-Thinking | (GLM et al., 2024) |
| Gemini-2.5-flash | (DeepMind, 2025) | Qwen2.5VL-72B | (Bai et al., 2025) |
| Claude-Sonnet-4 | (Anthropic, 2025a) | Qwen2.5VL-32B | (Bai et al., 2025) |
| Claude-Opus-4.1 | (Anthropic, 2025b) | LLAVA-Onevision-72B | (Li et al., 2024a) |
| Internvl3.5-38B | (Wang et al., 2025c) | LLAVA-Onevision-7B | (Li et al., 2024a) |
| Internvl3.5-14B | (Wang et al., 2025c) | SpaceR | (Ouyang et al., 2025) |
| Internvl3-78B | (Zhu et al., 2025) | Spatial-MLLM | (Wu et al., 2025c) |

projector remain frozen to preserve pretrained visual representations, the language model is fully trainable. Training utilizes full parameter fine-tuning with a DeepSpeed[4] ZeRO-2 configuration for memory optimization. The model is trained on our proposed dataset for spatial understanding in indoor environments, with samples truncated at 32,768 tokens. We implement a cosine learning rate schedule (initial LR=5e-7) with 10% warmup over 2 epochs. We maintain computational efficiency through mixed-precision bfloat16 training.

**RL.** We conduct our experiments using the Qwen2.5-VL (Bai et al., 2025) on a custom spatial dataset. The training utilizes 7 GPUs with DeepSpeed acceleration and mixed-precision bf16 training with flash attention. Key hyperparameters include a batch size of 1 per device, gradient accumulation steps of 1, an initial learning rate of 1e-6 with cosine scheduling, and weight decay of 0.01. The model processes input sequences up to 16,384 tokens long while generating outputs up to 1,024 tokens. Training runs for 2 epochs with evaluation performed every 200 steps. For inference, we use vLLM on a separate GPU with temperature 1.0 and generate 8 samples per input.

**Other Setting.** We set the number of experts $M$ to 4 in most cases. We also add LoRA with the same default behavior as PEFT. Additionally, we apply expert scaling factors on a layer-wise basis rather than globally.

**Ablation Setting.** Unless otherwise noted, we conduct all ablation experiments using the Qwen2.5-VL-3B model because of resource constraints; all other settings are identical to those described above.

## C.2. Patch Level Encoder Ablation

We evaluate several visual encoders with dense feature or geometry-aware representations, including VGGT-1B (Deng et al., 2025a)(the only publicly available model) and the generalDINOv3 ViT-Base, and perform ablations on the patch encoder. Tab. C12 reports the performance gains and computational costs associated with each model. Across encoders, DINOv3 achieves more favorable efficiency–accuracy trade-offs with a smaller parameter budget. We attribute this to its self-supervised pretraining, which is not constrained by labeled data and thus confers stronger generalization. In contrast, VGGT exhibits strong reconstruction capabilities but depends on annotations that lack rich semantic content and further relies on a large decoder to recover geometry. Consequently, compared to VGGT, DINOv3 features are more readily consumed by the fusion module, facilitating more effective mapping.

## C.3. LoRA Like Expert Ablation

On top of the same 3B pretrained base model, we compare three training strategies: **1) Full-parameter Fine-tuning**, **2) Vanilla LoRA**, and **3) LoRA-like Expert**, with the results shown in Fig. C13. We observe that vanilla SFT-based fine-tuning still suffers from latent cross-scale information conflicts. The difference between model-wise and layer-wise is that, for

---

[4]https://github.com/deepspeedai/DeepSpeed

*Table C12.* **Ablation Comparison of the patch-level encoder** across different sizes of models on the indoor set VSI-Bench based on the same SFT training settings.

| Model&Parameter | Video-Only | +VGGT | +DINO v3 | +VGGT +DINO v3 |
|---|---|---|---|---|
| SpaceVista-3B (Ours) | 41.9 | 43.3 | 43.5 | 43 .3 |
| SpaceVista-3B (Ours) *w/o.* fusion module | - | 42.0 | 44.8 | 44.7 |
| SpaceVista-7B (Ours) | 45.0 | 45.7 | 46.3 | 46.0 |
| **Extra Parameter** | 0 | 909M | 303M | 1,320M |

*Table C13.* **Ablation comparison of the LoRA-like expert** in the SFT training stage.

| Model | Benchmark | w/. Full-parameter Fine-tuning | w/. Vanilla LoRA Fine-tuning | w/. LoRA-like Expert (model-wise) | w/. LoRA-like Expert (layer-wise) |
|---|---|---|---|---|---|
| SpaceVista-3B | VSI-Bench | 43.5 | 42.9 | 43.9 | 45.3 |
| | SpaceVista-Bench | 29.5 | 29.4 | 32.5 | 33.0 |
| **Trainable Parameters** | | 3B | 20M | 80M+30M | 80M+34M |

each input, the router is calculated and implemented to the whole model or to separate layers, respectively. In contrast, the model-wise LoRA-like Expert yields clear gains over both full-parameter fine-tuning and vanilla LoRA. Furthermore, scaling to a higher-capacity, layer-wise LoRA-like Expert delivers additional improvements.

## C.4. Memorization effect observation (Out-of-Distribution Problem)

In our experiments, we observe that models often exhibit a strong bias toward memorizing fixed sizes for certain objects—for instance, chairs are typically assumed to be 50-70 cm tall. Consequently, the network tends to rely on memorized size priors rather than reasoning about object scale. However, this phenomenon presents a dual nature. On one hand, human perception of size and scale also depends on reference objects and familiar benchmarks, which are essential for intuitive understanding. On the other hand, since real-world spatial relationships can vary significantly, such biases may lead to erroneous judgments in atypical cases.

We argue there is two types of Out-of-Distribution (OOD) that should be discussed separately. 1) **OOD category with normal size** 2) **normal category with OOD size**.

For **normal category with OOD size**, we need to develop a dataset with precise annotation. The Guinness World Records (GWR) is a globally recognized organization[5] that catalogs uncommon objects and forms. We obtain precise size measurements along with the corresponding images/videos, and construct a series of QA pairs about object sizes as shown in Fig. C18. The GWR data comprises diverse scenes, including outdoor, indoor, and drone, with over 50 images and over 50 questions. Because only a small portion of the records is documented on the website, we used nearly all available website content to construct this GWR test set. All questions were created through human annotation to ensure dataset quality. This data is used solely for insight and analysis, not for official purposes. The licensing status of GWR content is unclear. If the license permits, we will release this GWR set on Hugging Face.

*Table C14.* **Performance comparison across GWR dataset.**

| Size-Related QA | Qwen2.5VL-7B | Qwen2.5VL-3B | SpaceVista-7B | SpaceVista-3B |
|---|---|---|---|---|
| SpaceVista-Bench | 49.9 | 44.0 | 58.3 | 49.3 |
| GWR set | 27.8 | 23.1 | 31.1 | 27.3 |

As shown in Table C14, we evaluate the popular Qwen2.5-VL model and our SpaceVista-7B model. Because the GWR data contain only size-related questions, we select the size-related subset of SpaceVista-Bench to ensure a fair comparison. We find that these OOD data are challenging for both the general-purpose model and our specialist model. However, the OOD challenge does not produce a clear performance gap between Qwen2.5-VL and SpaceVista. Although our model is not designed for purely image-based tasks, this potential bias suggests a promising direction for future work in VLLMs.

For **OOD category with normal size**, to systematically evaluate the impact of this bias and its potential implications for

---

[5]https://www.guinnessworldrecords.com/records/showcase

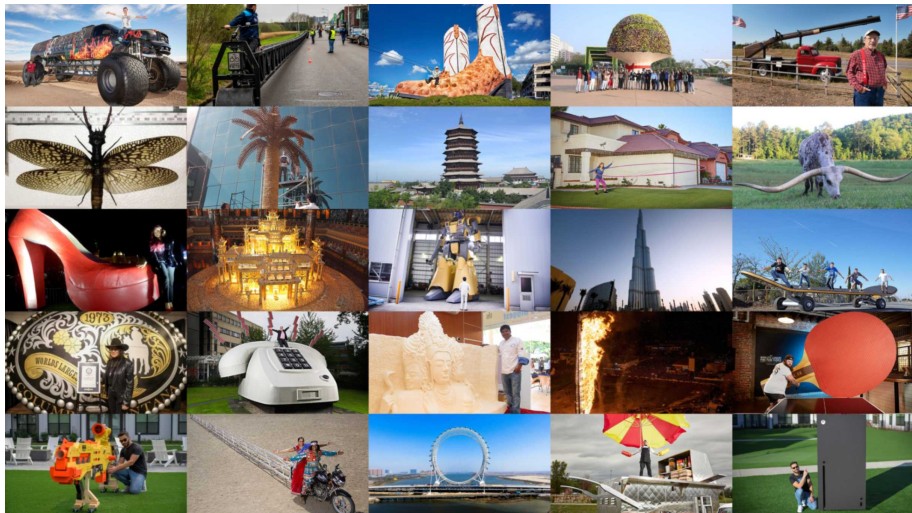

*Figure C18.* **Data preview in the Guinness World Records (GWR).** GWR is a globally recognized organization that catalogs uncommon objects and forms. We scraped precise size measurements along with the corresponding images/videos, and constructed a series of QA pairs about object sizes.

advancing the field, we design three specialized subsets at the same scale:

- **Seen Set:** Common object categories from the training distribution (i.e., bicycle, table, chair).

- **Seen Set with Various Scales:** bjects of the same category (i.e., different sizes and shapes of screw).

- **Unseen Set:** Rare or culturally specific objects requiring contextual size reasoning (i.e., ethnic items with regional characteristics, such as a traditional food).

The Seen Set provides baseline performance metrics for familiar objects but may overlook biases due to training conformity. The Seen Set with scale variety directly probes size generalization for known categories, but it is limited to variations within seen objects. The Unseen Set evaluates robustness to novel, culturally diverse scenarios but risks introducing confounders beyond scale bias. Collectively, these subsets balance ecological validity with experimental control, offering a comprehensive framework to diagnose size-related biases. This structured approach enables us to analyze how size biases manifest under different conditions, combining ecological validity with controlled experimentation. As shown in Fig. C15, all-scale training benefits the overall reasoning model; however, the general models still tend to memorize the regular size of the target object.

*Table C15.* **Reasoning VS memorizing analysis** of different subsets.

| Model | Seen Set (Normal) | Seen Set (Various Scales) | Unseen Set |
|---|---|---|---|
| Qwen2.5-VL-3B-Instruct | 35.7 | 34.7 | 23.1 |
| Qwen2.5-VL-7B-Instruct | 37.0 | 38.9 | 28.0 |
| **SpaceVista-7B (Ours)** | **37.3** | **41.0** | **32.8** |

Our analysis of potential bias has two parts:

1. **Depth Knowledge.** Current metric depth models estimate distance primarily based on accurate camera parameters, such as focal length. These parameters vary across different scales, which is why our model performs slightly better than a general model.

2. **Scale Prior.** Human distance estimation also strongly relies on reference objects (i.e., scale priors in question). When these references are unusual, humans also unavoidably exhibit bias. Thus, scale priors are a double-edged sword and cannot be simply described as good or bad.

## C.5. Challenging Scenario Analysis

*Table C16.* **Results analysis of different scenes.** The model mentioned below is trained in a balanced subset of SpaceVista-1M for better control of experiment conditions.

| Model | SpaceVista-Bench (Ours) | | | |
| --- | --- | --- | --- | --- |
| | Indoor | Outdoor | Tabletop | Tabletop |
| Qwen2.5-VL-7B | 30.34 | 18.31 | 23.79 | 19.37 |
| $w/.$ balance training | 38.77 | 24.90 | 30.17 | 20.86 |

When testing scenes at varying scales, several critical questions arise: Which scenarios pose greater challenges, and to what extent is data complexity the primary bottleneck? To systematically investigate these issues, we design a controlled observational experiment.

We identify tasks that exhibit consistent properties across different scales, including object size, object comparison, absolute and relative distance, and depth estimation. For fairness in comparison, we train models using videos from diverse scenes while maintaining similar quantities of QA pairs and video samples. Under these controlled conditions, we evaluate and compared performance across different scale-dependent scenarios. In Tab.C16, it seems indoor data is the easiest task. We hypothesize that a human-scale estimation bias—arising because both humans and GPT focus on objects expressible in basic units like meters in pretraining corpora—leads to this preference.

## C.6. 3D Geometric Feature Observation

*Table C17.* **Comparison of the robustness of the model training of 3D and 2.5D.** All the models are trained on 3D or 2.5D data along with the video. However, we vary the evaluation input of these models to see the robustness. "–" denotes experiments we consider unnecessary. "low" means using low resolution visual for 3D reconstruction. This table includes only the popular model for which a detailed score is available. For average-score comparisons, see Table 2. "($n\%$)" means the relative decrease compared to the original input.

| Settings | Eval Input | VSI-bench | SpaceVista-Bench |
| --- | --- | --- | --- |
| Training with $w/.$ 3D | visual $w/.$ 3D | 44.3 | 31.4 |
| | visual $w/.$ 3D (low) | 38.1 (-14%) | – |
| | visual $w/o.$ 3D | 34.0 (-23%) | – |
| Training with $w/.$ 2.5D | visual $w/.$ 2.5D | 45.6 | 33.0 |
| | visual $w/.$ 2.5D (low) | 43.9 (-4%) | 32.3 (-2%) |
| | visual $w/o.$ 2.5D | 40.7 (-10%) | 29.1(-12%) |

In addition to introducing VGGT(Wang et al., 2025a) and DINO v3(Siméoni et al., 2025) as extra signals, we conduct a series of targeted ablation studies. This suggests that representation formats like VGGT, when used in their native encoder output, are wonderful for capturing geometry information, but suboptimal for capturing semantic information or overall scenes, especially for low resolution and uncommon scenarios. In Tab.C17, we use "3D" to denote the pure geometric features from VGGT, and "2.5D" to denote the additional 12 viewing angles of the overall scene rendered by the decoder and the renderer. We use the special prompt and the image token to provide

As shown in Tab.C17, 2.5D is usually more robust in spatial reasoning. Rendering to 2.5D enables effective exploitation of pretrained image tokenizers, which in turn provides more reliable semantic information.

Below is the special prompt for 2.5D finetuning.

"*Please think about this question as if you were a human pondering deeply. Consider detailed information from the video frames and coarse spatial information from the 3D point cloud image. Provide the model's thought process and reasoning between the <think> </think> tags, and give your final answer between the <answer> </answer> tags. <video> The images below are obtained from the 3D point clouds based on the video frames above. The following point cloud images are randomly selected viewpoints; some may be completely unhelpful, while others may contain important information. Please discern carefully. <image> Provide your reasoning between the <think> </think> tags and your final answer between the <answer> </answer> tags.*"

## C.7. Leaderboard Settings Detail

To assess the spatial reasoning ability of both closed-source and open-source models, we evaluate the latest available versions. Tab. 5 presents their performance across the Tiny Tabletop, Tabletop, Indoor, and Outdoor scenarios, whereas Tab. C18 provides an overview of their release dates and sources.

*Table C18.* **The release time and model source of LLMs used.**

| Model | Release Time | Source |
|---|---|---|
| GPT-5(OpenAI, 2025) | 2025-08 | https://openai.com/gpt-5/ |
| GPT-4o(Hurst et al., 2024) | 2024-05 | https://gpt4o.ai/ |
| Claude-Opus-4.1(Anthropic, 2025b) | 2025-08 | https://www.anthropic.com/news/claude-opus-4-1 |
| Claude-Sonnet-4(Anthropic, 2025a) | 2025-05 | https://www.anthropic.com/claude/sonnet |
| Gemini-2.5-Pro(DeepMind, 2025) | 2025-06 | https://deepmind.google/technologies/gemini/pro/ |
| Gemini-2.5-Flash(DeepMind, 2025) | 2025-06 | https://deepmind.google/models/gemini/flash/ |
| Internvl3.5-38B (Wang et al., 2025c) | 2025-08 | https://huggingface.co/OpenGVLab/InternVL3_5-38B-Instruct |
| Internvl3.5-14B (Wang et al., 2025c) | 2025-08 | https://huggingface.co/OpenGVLab/InternVL3_5-14B-Instruct |
| Internvl3-78B (Zhu et al., 2025) | 2025-04 | https://huggingface.co/OpenGVLab/InternVL3-78B |
| Internvl3-38B (Zhu et al., 2025) | 2025-04 | https://huggingface.co/OpenGVLab/InternVL3-38B |
| GLM-4.5V (Team et al., 2025) | 2025-08 | https://www.glm45.com/glm45v |
| GLM-4.1V-Thinking (GLM et al., 2024) | 2025-07 | https://huggingface.co/zai-org/GLM-4.1V-9B-Thinking |
| Qwen2.5VL-72B (Bai et al., 2025) | 2025-01 | https://huggingface.co/Qwen/Qwen2.5-VL-72B-Instruct |
| Qwen2.5VL-32B (Bai et al., 2025) | 2025-01 | https://huggingface.co/Qwen/Qwen2.5-VL-32B-Instruct |
| LLAVA-Onevision-72B (Li et al., 2024a) | 2024-08 | https://huggingface.co/llava-hf/llava-onevision-qwen2-72b-ov-hf |
| LLAVA-Onevision-7B (Li et al., 2024a) | 2024-08 | https://huggingface.co/lmms-lab/llava-onevision-qwen2-7b-ov |

*Table C19.* **Parameter settings for Closed-Source LLMs generation.**

| Model | Generation Setup |
|---|---|
| GPT-5 | "model" : "gpt-5", "temperature" : 0, "max_tokens" : 1024 |
| GPT-4o | "model" : "gpt-4o", "temperature" : 0, "max_tokens" : 1024 |
| Claude-Opus-4.1 | "model" : "claude-opus-4.1", "temperature" : 0, "max_tokens" : 1024 |
| Claude-Sonnet-4 | "model" : "claude-sonnet-4", "temperature" : 0, "max_tokens" : 1024 |
| Gemini-2.5-Pro | "model" : "gemini-2.5-pro", "temperature" : 0, "max_tokens" : 1024 |
| Gemini-2.5-Flash | "model" : "gemini-2.5-flash", "temperature" : 0, "max_tokens" : 1024 |

For closed-source models accessed via API and open-source models, the generation configurations are summarized in Tab. C19 and C20, respectively.

*Table C20.* **Generating parameters for Open-Source LLMs.**

| Model | Generation Setup |
|---|---|
| Internvl3.5-38B | do_sample = False, temperature = 0, max_new_tokens = 512 |
| Internvl3.5-14B | do_sample = False, temperature = 0, max_new_tokens = 512 |
| Internvl3-38B | do_sample = False, temperature = 0, max_new_tokens = 512 |
| Internvl3-78B | do_sample = False, temperature = 0, max_new_tokens = 512 |
| GLM-4.5V | do_sample = False, temperature = 0, max_new_tokens = 1024 |
| GLM-4.1V-Thinking | do_sample = False, temperature = 0, max_new_tokens = 1024 |
| Qwen2.5VL-32B | do_sample = False, max_new_tokens = 1024 |
| Qwen2.5VL-72B | do_sample = False, max_new_tokens = 1024 |
| LLAVA-Onevision-7B | do_sample = False, temperature = 0, max_new_tokens = 1024 |
| LLAVA-Onevision-72B | do_sample = False, temperature = 0, max_new_tokens = 1024 |

# D. FAQ

## D.1. Error Accumulation

Our data construction pipeline is primarily based on metric depth estimation and the corresponding transformation to canonical view space. It should be noted that this approach may introduce potential error accumulation, especially considering that current metric depth estimation models have not yet achieved high performance at full scale.

To address concerns regarding error accumulation, we justify our methodology from the following perspectives: **1) data quality assurance:** To ensure alignment with human perception, we implement a multi-tiered validation process. Specifically,

we conduct manual verification on a subset of the training set, perform full human annotation on the entire test set, and additionally collect real-world measured data to construct a dedicated test subset. These measures effectively ensure that the automatically generated data remains suitable for learning human perceptual models. We argue that even if minor error accumulation exists, it does not compromise the overall quality and contribution of the dataset. **2) forward-looking methodological contribution:** The proposed data construction framework and model architecture will have a significant impact on the field of all-scale spatial reasoning. Importantly, as more accurate all-scale inference methods emerge in the future, we will continuously integrate higher-quality data to refine this work. This dynamic updating mechanism ensures the long-term relevance and value of our research.

### D.2. All Scale Possibilities

Currently, our data coverage remains limited in addressing the full spectrum of spatial scales, despite the equal importance of spatial understanding across these domains. At fine scales, domains such as minimally invasive surgery call for millimeter-level models, while precision manufacturing—especially semiconductor production—pushes into the nanometer range. These capabilities underpin progress in healthcare and technology. In contrast, large-scale applications, including satellite remote sensing and cartography, typically work with resolutions of 10 kilometers or greater.

While spatial understanding is equally essential across these extremes, the imaging and 3D modeling techniques involved extend well beyond conventional real-world sensing methods. As a result, our current work does not fully address these diverse scales. Nevertheless, we aim to expand our capabilities in the future by integrating modeling across a broader range of dimensions, thereby bridging these gaps and enabling more unified spatial analysis.

### D.3. Dataset Usage Discussion

We use the free-form subset of SPAR-7M(Zhang et al., 2025c), which consists of approximately 100K samples, about 1% of the original dataset. This part of the data is later processed and filtered with original Scannet (Dai et al., 2017), Scannet++ (Yeshwanth et al., 2023), and ARKitScenes (Baruch et al., 2021) to fit the requirements of our dataset. However, we do not consider our model to be trained on SPAR-7M, nor do we compare it against models trained on SPAR-7M in SparBench. We observe that SPAR-7M's data design leads to over 200 QA pairs per scene on average, which can cause overfitting in indoor scenarios. Instead, we leverage SPAR-7M's scan-based characteristics to construct our own CoT for cold-start purposes. It is important to note that neither SpaceR nor SPAR-7M includes CoT reasoning. We generate CoT following the method described in Sec.§§ 3 and apply filtering and screening to ensure quality. These processed data sources, along with the wild video dataset, are integrated into SpaceVista-1M, while acknowledging the additional labeling and filtering steps involved in our pipeline. Overall, these decisions support our position that our data retains a meaningful degree of independence from SPAR-7M and SpaceR.

## E. Preview

### E.1. Scene Preview

**Indoor Scenes.** Our indoor dataset consists of simple and clean room-scale environments such as living rooms, meeting rooms, and classrooms. An overview of the data is provided in Fig. E19, highlighting the simplicity and cleanliness of our indoor scenes compared to more complex wild indoor environments. Living rooms feature sofas, coffee tables, and shelves arranged along walls with open floor space. Meeting rooms include evenly spaced chairs around a central table, while classrooms have rows of desks facing a blackboard or screen. These scenes show limited object variety and limited scene complexity.

**Wild Indoor Scenes.** Representative wild indoor scenes, captured via multi-view smartphone recordings in complex and unconstrained environments such as shopping malls, banquet halls, and art galleries, are illustrated in Fig. E20. These scenes exhibit diverse architectural layouts and high object density. Like in shopping malls, elements such as escalators, display shelves, and glass facades create multi-layered structures with frequent reflections and occlusions. Compared to previous indoor scenes, wild indoor scenes have irregular layouts, dense furniture, diverse objects, and uneven lighting, leading to more complex spatial arrangements. This contrast underscores the structured and clear nature of our data, which supports controlled spatial reasoning evaluation.

**Outdoor Scenes.** Our outdoor scenes include various environments such as parks, tourist landmarks, and others, captured

from both ground and aerial views, as shown in Fig. E21. Parks contain irregularly shaped walking paths winding through dense clusters of trees, shrubs, and open lawns, creating a mix of natural textures and spatial variations. These areas often include water features, benches, and varied terrain elevations. Therefore, outdoor scene layouts usually involve plazas, staircases, and structured open spaces that introduce rich geometric complexity.

**Drone Scenes.** Fig. E22 shows examples from a drone's perspective. Aerial, low-angle, and oblique views offer detailed spatial structures that are not easily visible from the ground. Playgrounds exhibit clear arrangements of play equipment and open spaces, while parking lots display orderly rows of vehicles and marked boundaries. Parks show clusters of trees, pathways, and water bodies, revealing a layered combination of natural and built elements. These diverse viewpoints provide a more complete understanding of scene layout and environmental features, supporting improved spatial reasoning.

**Tabletop Scenes.** Examples of tabletop scenes are illustrated in Fig. E23. These scenes capture everyday objects such as keyboards, boxes, and fruits arranged on tabletops, characterized by natural occlusions, varying object placements, and diverse background textures. The dataset employs dynamic multi-view acquisition using mobile devices, enabling richer structural coverage compared to traditional static indoor datasets. This approach captures subtle interactions between objects and background elements, as well as changes in viewpoint and lighting conditions.

**Tiny Tabletop Scenes.** The Fig. E24 shows the tiny tabletop scenes from our dataset. These data are 360-degree turntable videos to capture objects from every angle, solving occlusion issues and improving scene completeness.

**Our Collected Scenes.** We use mobile devices to capture and collect data for some Tabletop and Tiny Tabletop scenes. Our collected data, shown in Fig. E25, features diverse objects and detailed multi-view coverage, enabling fine-grained spatial analysis. The data is similar to the previously mentioned tabletop and tiny tabletop. Tabletop scenes have relatively large objects and rich and diverse backgrounds, which are suitable for capturing diverse objects and natural environments in daily life; while Tiny Tabletop scenes focus on smaller objects, emphasizing detail integrity and multi-view coverage, which facilitates in-depth research on the subtle structure and morphology of these scenes.

### E.2. Template Preview

As shown in Tab. E21, we present three exemplar applications: point input for Object Counting, bounding box input for Object Distance, and original input for Spatial Relation. Other scenes and tasks are similar to the example template.

### E.3. QA Preview

You can have as much text here as you want. The main body must be at most 8 pages long. For the final version, one more page can be added. If you want, you can use an appendix like this one.

The \onecolumn command above can be kept in place if you prefer a one-column appendix, or can be removed if you prefer a two-column appendix. Apart from this possible change, the style (font size, spacing, margins, page numbering, etc.) should be kept the same as the main body.

*Table E21.* **Multi-type template preview.** Examples using the point input for Object Counting, the bounding-box input for Object Distance, and the original input for Spatial Relation.

**Point Input Template**
- Refer to the red point in the starting frame and count how many objects are of that type.
- Count the number of objects whose class is referred to by the red point in the first frame throughout the video.
- Using the red point in the first frame as reference, count how many objects of that class appear in the entire video.
- Count every object like the one highlighted by the red point in the video's first frame.
- Find all video objects that are of the same kind as the one identified by the red point.
- Identify the class from the red point in frame one and tally all instances of that class in the video.
- How many objects in the video resemble the one tagged with the red point in the first frame?
- Search for all items that belong to the same class as the one shown by the red point in frame one.
- Track all objects of the same category as the red-point one from the first frame and count them.
- Count the total number of objects in the video that correspond to the class defined by the red point in the first frame.
- Use the red point to find a class and count how many such instances are there in the video.
- Using the initial frame's red point as a guide, total up all objects of that class.
- From the first frame's red point, find that class and count its appearances across the video.
- Match the object under the red point to others in the video and count them.
- Take the red-pointed object as example and count all others like it in the video.

**Bounding Box Input Template**
- How far apart do the objects enclosed by the red bounding box and blue bounding box appear in these frames?
- What space lies between the red bounding box and the blue bounding box in these frames?
- What is the distance measurement between the red bounding box and blue bounding box in the video?
- What is the distance between the red-bounded object and the blue-bounded object in the video?
- Measure the distance separating the red bounding box and blue bounding box in the video frames.
- What is the estimated distance between the red bounding box and the blue bounding box in the video?
- What is the measured distance between the red bounding box and blue bounding box in the footage?
- Calculate the ground distance from the red bounding box to the blue bounding box based on the frames.
- Find the ground distance between the red bounding box and the blue bounding box in these images.
- How wide is the space between the red bounding box and the blue bounding box in the video?
- Based on the frames, what is the distance from the red bounding box to the blue bounding box?
- Please estimate the ground distance between the red bounding box and the blue bounding box in these images.
- What is the approximate distance between the red bounding box and blue bounding box in these images?
- Provide an estimate for the distance between the red bounding box and blue bounding box seen in the footage.
- How far is the red bounding box from the blue bounding box in the frames?

**Original Input Template**
- Describe how desk and chair are spatially positioned relative to each other.
- What is the spatial relation type between desk and chair in the video?
- What type of spatial relationship exists between desk and chair in these frames?
- Estimate the spatial relation (such as support, stacking, adhesion, hanging, plug-in) between desk and chair in these frames.
- What is the most likely spatial relationship (support, stacking, adhesion, hanging, plug-in) between cabinet and book?
- Can you describe the spatial relationship type of awning and awning?
- Identify how picture and ceiling are spatially related in the video sequence.
- Between desk and chair, what spatial link exists?
- What spatial relation links tag to hat in the given frames?
- What spatial relation best fits cable and computer mouse in the video frames?
- Identify how cable and socket are spatially related in the video sequence.
- Describe the spatial relation (e.g., support, stacking, adhesion, hanging, plug-in) between fork and spoon.
- Explain the spatial relation between toy camera and building blocks in the video.
- How would you classify the spatial relation between sticky note and tumbler?
- What type of spatial relationship exists between toy block and toy train in these frames?.

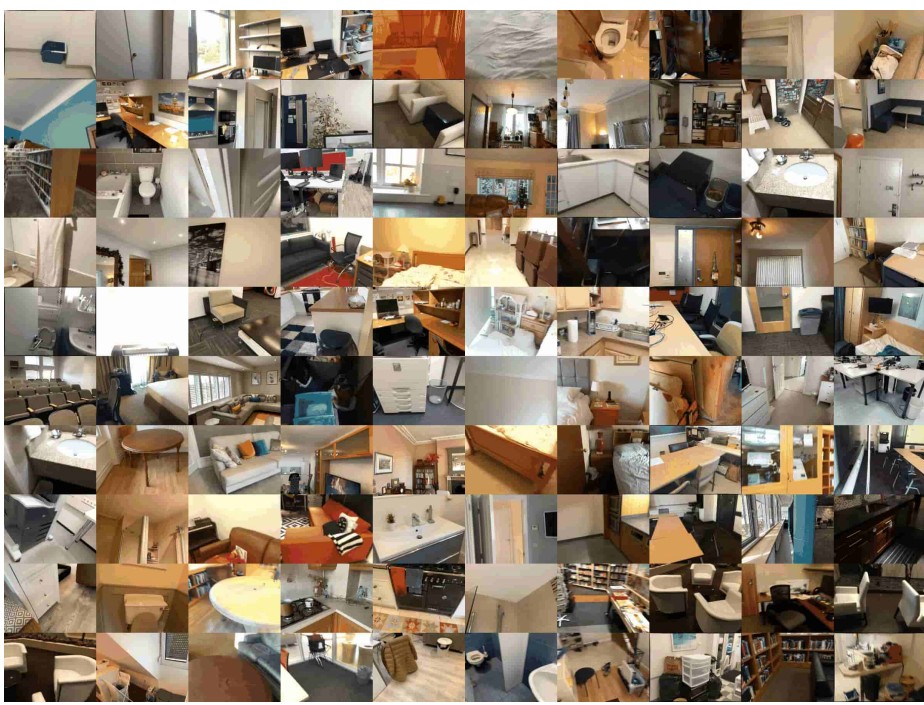

*Figure E19.* **Preview of indoor data.** Indoor data are rather simple and clean scenes inside a room. The overall scene is not as complex as the wild indoor scene.

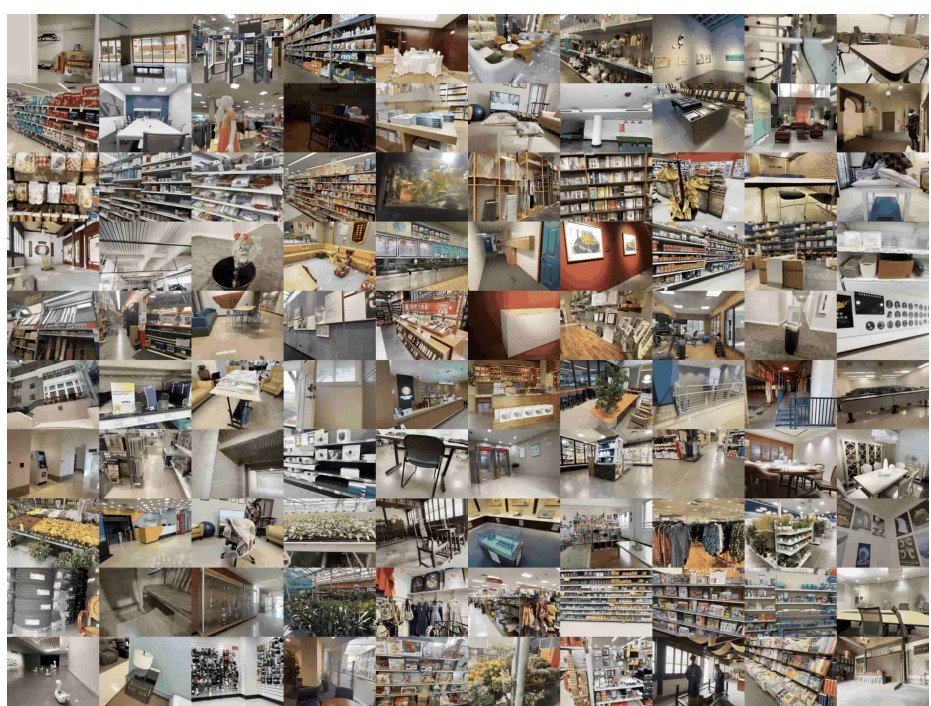

*Figure E20.* **Preview of wild indoor data.** Wild indoor data includes more light changes, reflections, and transparency. The objects included are more diverse.

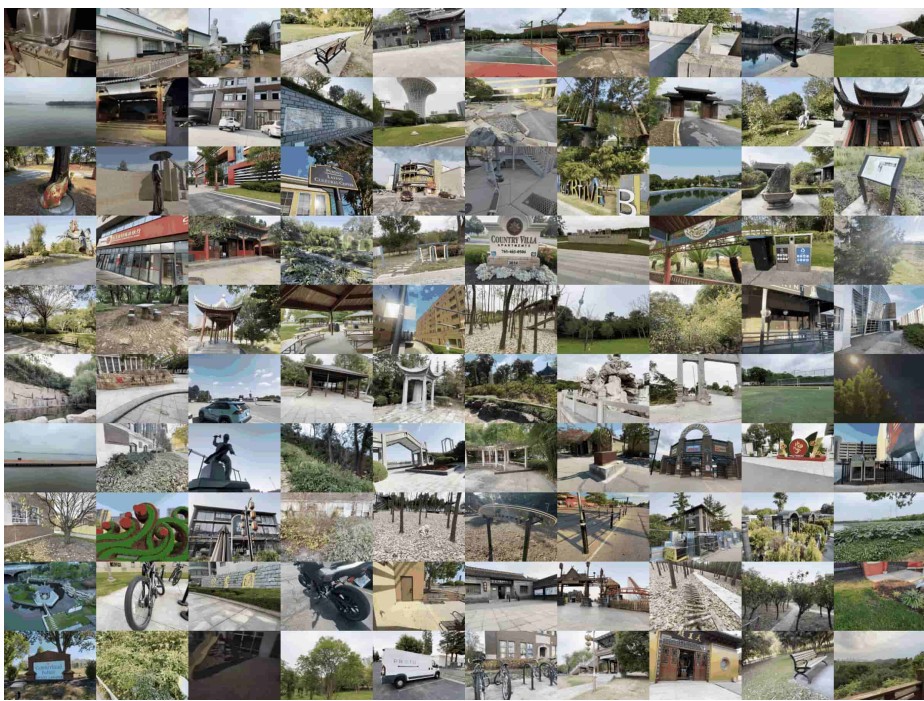

*Figure E21.* **Preview of outdoor data.** Outdoor data is jointly collected from ground views, incorporating street, park, building and so on.

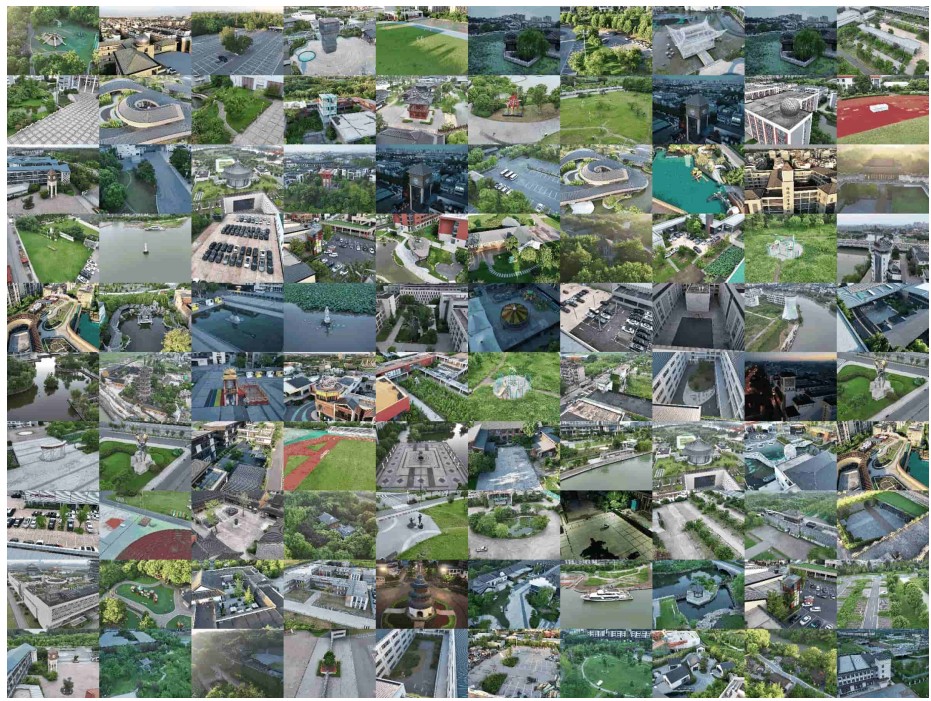

*Figure E22.* **Preview of drone data.** Drone data captures ground objects from above at oblique angles, providing more complete structural coverage than traditional ground-based capture methods.

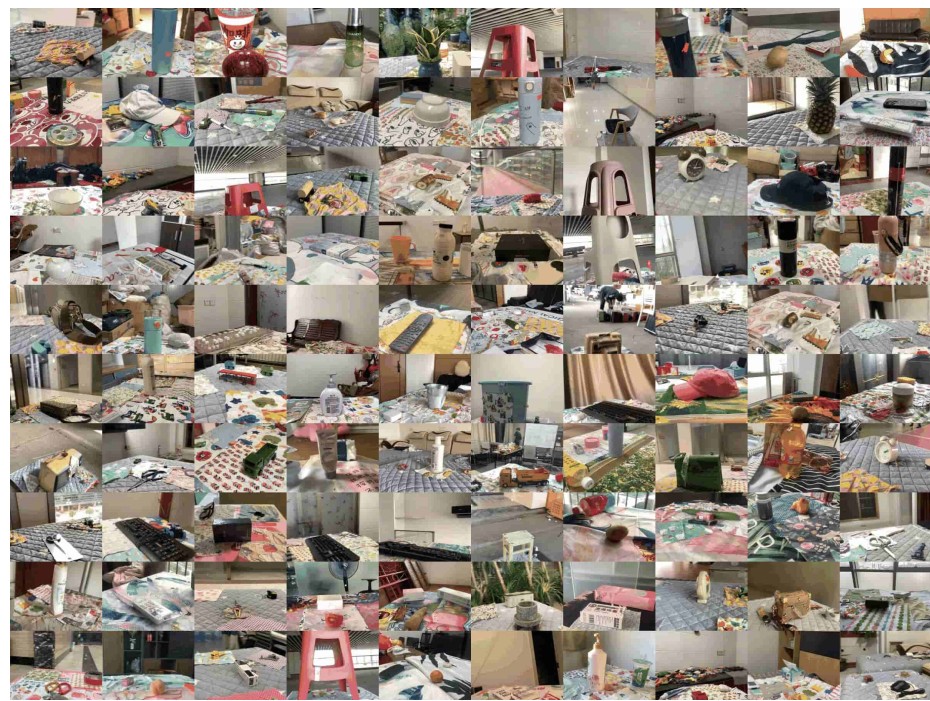

*Figure E23.* **Preview of tabletop data.** In this tabletop scene, videos capture tabletop objects exhibiting rich background variation and natural occlusions, delivering clearer structural coverage of the objects than traditional static indoor datasets.

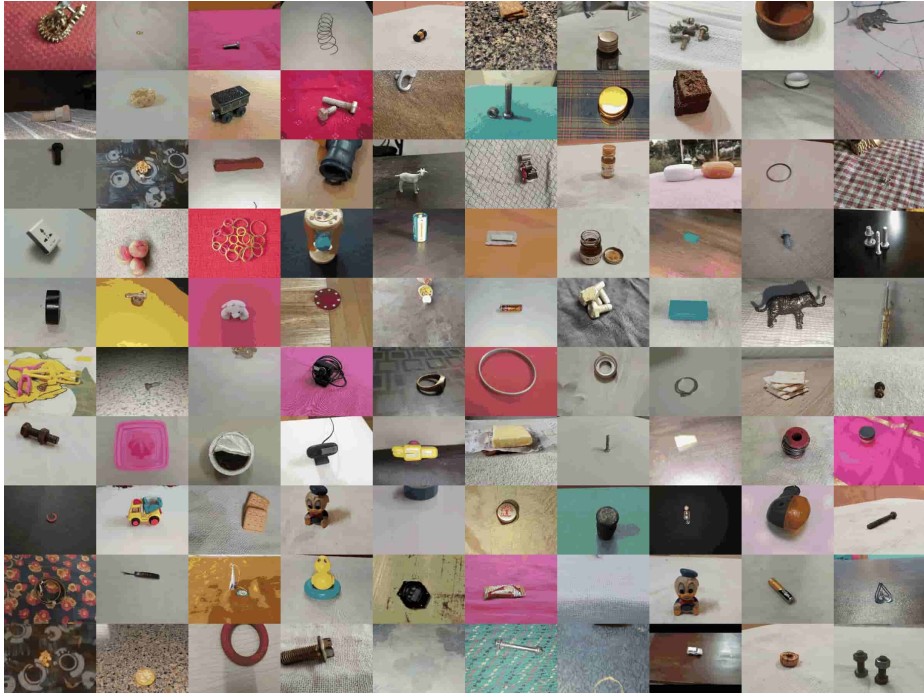

*Figure E24.* **Preview of tiny tabletop data.** Tiny tabletop objects captured with rich details for small objects, focusing on fine-scale scenes, unlike typical large or complex indoor or outdoor datasets.

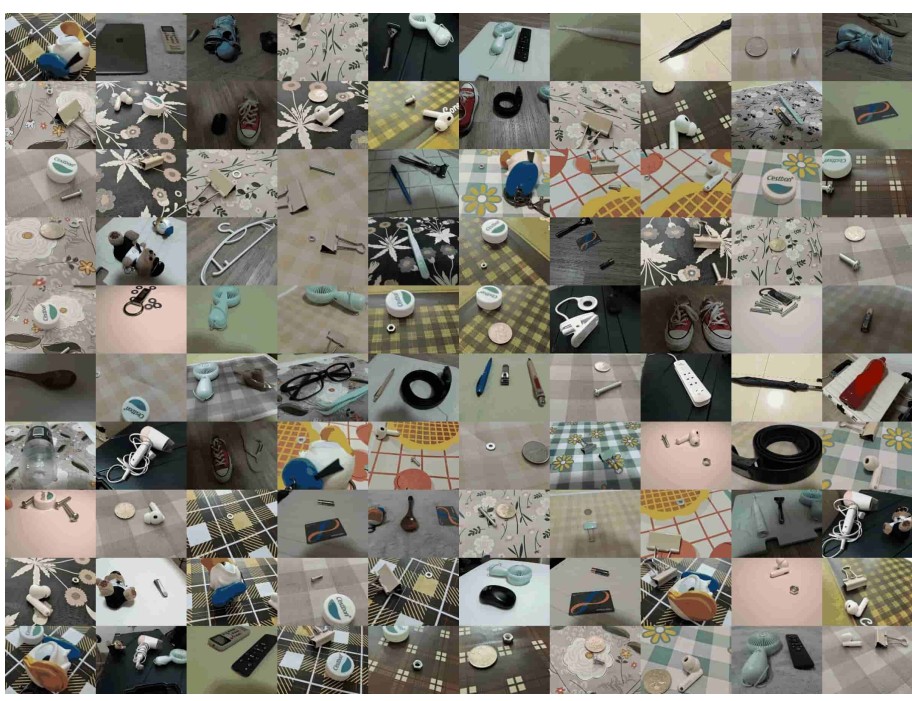

*Figure E25.* **Preview of self-collected data.** These samples are collected by us. As small-scale, Tabletop, and Tiny Tabletop datasets offer rich details with accurate annotations.

