In this section, we first conduct a diversity analysis of the visual scenes, examining their composition, categories, and object size distributions (Sec.§ B.4.1-Sec.§ B.4.5). We then present a statistical overview of the QA pairs, along with an evaluation of quality control mechanisms (Sec.§ F.3-Sec.§ B.4.7). And also, at the beginning of the appendix, Fig. A5 illustrates the data composition for each scene task, where bubble sizes indicate the relative data volume.

B.4.1. TARGET CATEGORY DISTRIBUTION

The introduction of diverse scenarios, such as tabletop, indoor, and outdoor, aims to establish a more inclusive object composition system. Due to the limited drone data, we incorporate drone-view data into the outdoor analysis. By approximating complex object distribution patterns to the real world, this approach enhances the scene adaptation capabilities of visual reasoning models. To quantitatively assess the impact of scene diversity on model generalization, we use the word cloud to compare object distribution characteristics across different scenarios, as shown in Figs. B12–B17. The results reveal that indoor scenes are predominantly composed of rigid objects such as furniture and electronics, exhibiting a highly structured spatial layout. In contrast, outdoor scenes feature more scale-varying objects like vehicles and natural landscapes, demonstrating spatial openness. Meanwhile, tabletop scenes focus on manipulable items such as tools and daily necessities, reflecting precise spatial arrangements. These cross-scene differences provide complementary training samples, effectively mitigating the risk of overfitting to specific scenarios. Thus, the necessity of a multi-scenario strategy to enhance cross-domain generalization is validated.

Overall, each subset scenario differs significantly from the previous indoor-dominated setting, highlighting the diversity of

1375 our scenes.

1377 ### B.4.2. SCALE DISTRIBUTION

1379 To evaluate the dynamic range of depth across different scenes, we statistically analyze the distributions of the maximum
1380 and minimum depth values in each scenario, with the results visualized in Fig. B18. This analysis reveals the variation in
1381 extreme depth ranges. Notably, the farthest depth point progressively decreases from the drone scene to the tiny tabletop

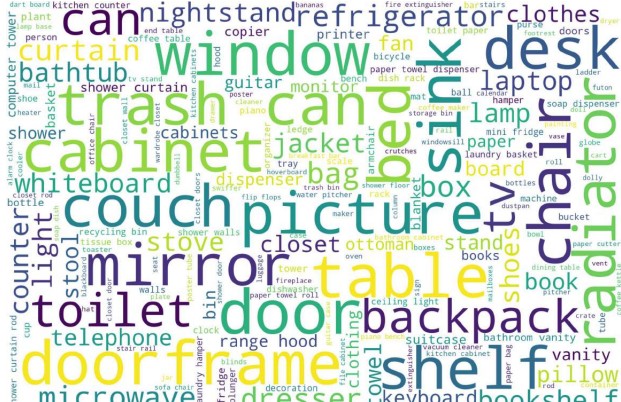

*Figure B12.* **The word cloud of the previous indoor spatial reasoning datasets.**

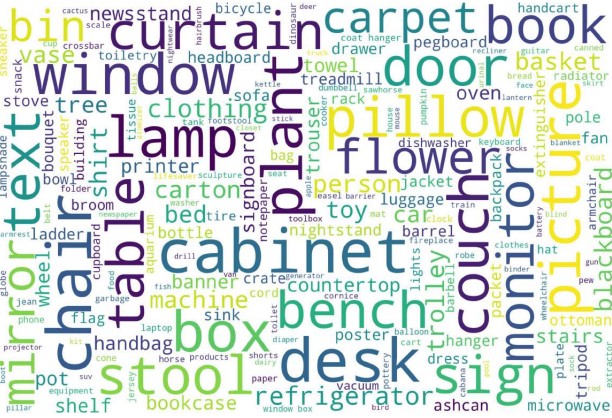

*Figure B13.* **The word cloud of our indoor subset.**

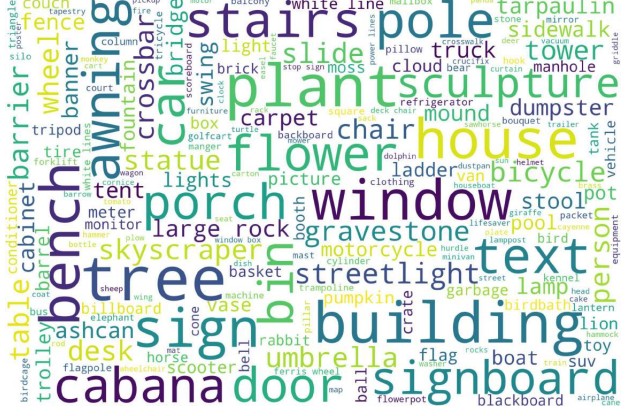

*Figure B14.* **The word cloud of our outdoor subset.**

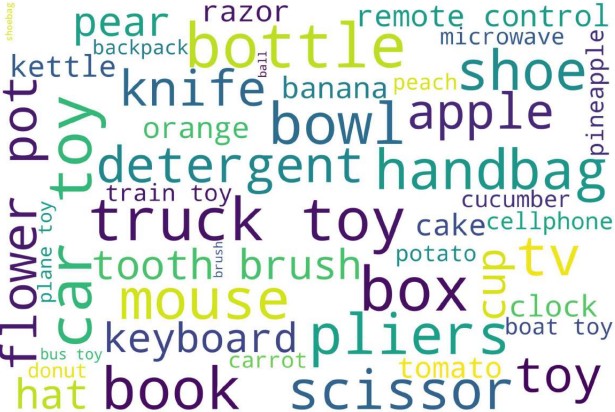

*Figure B15.* **The word cloud of our tabletop subset.**

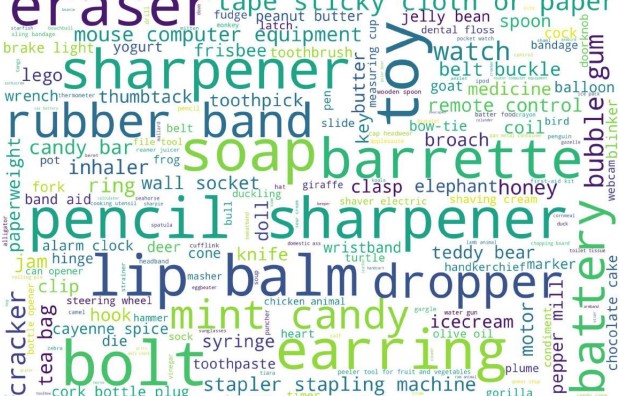

*Figure B16.* **The word cloud of our tiny tabletop subset.**

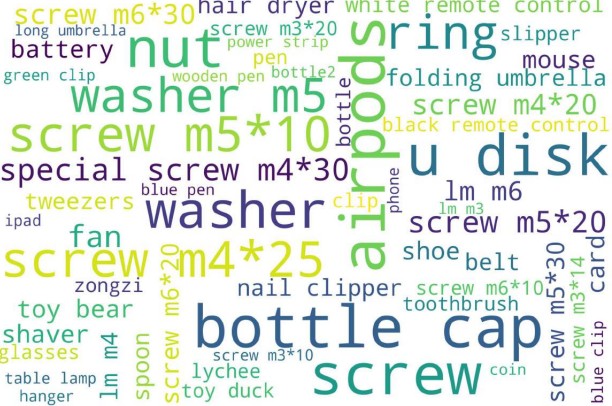

*Figure B17.* **The word cloud of the self-collected subset.** Note: We use standard ISO 7046 to denote the models of the screw, which looks like *"m4*10"*.

scene, indicating a consistent reduction in the overall scene. While we can see some extreme values for tiny object scenes, it might be the small object around the window, and extreme depth represents the outside of the window view. It is not unavoidable for data construction and will not affect the overall quality.

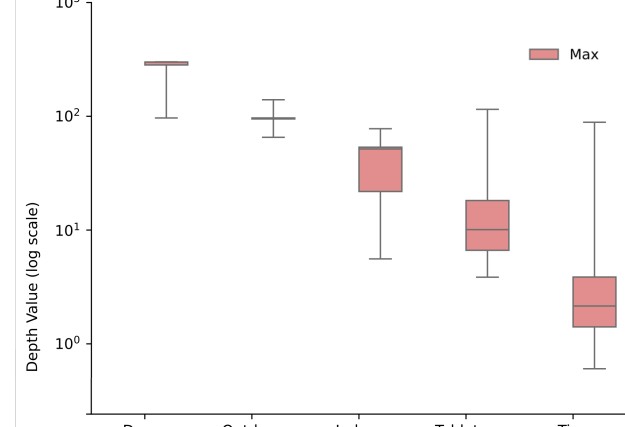

*Figure B18.* **The distribution of the maximum depth value of our SpaceVista-1M.** The maximum distance denotes the farthest point observed.

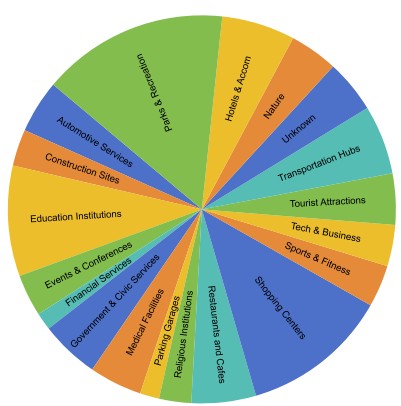

*Figure B19.* **The distribution of the specific sceneries.** Note: this chart is just for basic knowledge. Due to the latter filtering policy, there might be some vague or inaccurate analysis.

### B.4.3. SUBSCENE TYPE DISTRIBUTION

While our dataset is largely derived from multiple existing sources, we perform a thorough analysis of its scene type diversity. As shown in Fig. B19, the dataset covers a broad range of real-world scenarios, enhancing its complexity and generalizability. To quantify this diversity, we utilize LLM for scene understanding, leveraging object-level annotations from the video data. However, certain subsets, such as partial tabletop scenes and most of the tiny tabletop data, are excluded from the analysis due to limited visual cues. As a result, these statistics primarily illustrate the dataset's variety rather than providing an exact distribution for downstream tasks.

### B.4.4. OBJECT SIZE DISTRIBUTION

To enhance spatial understanding at design scales, we analyze the distribution of object sizes in the dataset. The results, shown in Fig. B20, reveal a relatively uniform distribution for objects smaller than 50m, while those exceeding 100m exhibit a certain tail distribution. This trend likely reflects real-world bias in object sizing, with high-rise buildings, common in urban environments, dominating the larger size categories. Consequently, the observed minor long-tail distribution aligns with real-world phenomena and is considered an acceptable characteristic of the dataset.

### B.4.5. CAMERA TO OBJECT DISTRIBUTION

To examine biases regarding camera positioning relative to the subject, we analyze the distance (depth) between the camera and the primary object, with the statistical results shown in Fig. B21. The distribution of object-camera distances follows a spindle-shaped pattern, with few instances where the object is positioned closer than 10 cm or farther than 500 m from the camera. This trend is largely influenced by the focusing limitations of most hardware, like lenses, which exhibit reduced sensitivity to objects at extreme distances. Notably, this distribution mirrors that of conventional optical devices in real-world settings and should not be interpreted as a dataset bias.

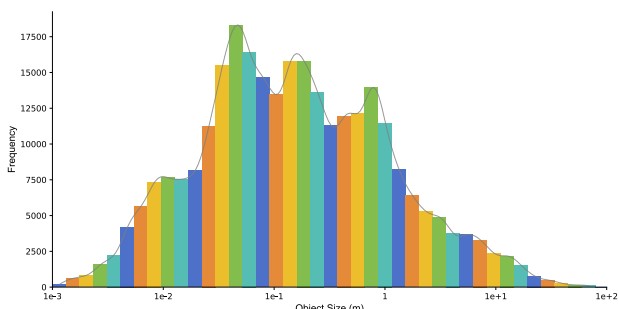

*Figure B20.* **The distribution of the size of the existing objects.**

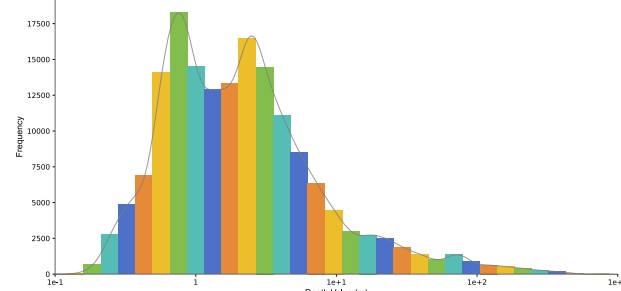

*Figure B21.* **The distribution of the distance between the target object and the camera.**

### B.4.6. QA STATISTICS ACROSS SCENES

We also provide the overall statistics of the SpaceVista-1M dataset in Tab. B8. The SpaceVista-1M dataset consists of approximately 1 million QA pairs, covering a wide range of tasks and scene types across all scales, from tiny tabletop objects to large-scale outdoor and drone-view scenarios, with scales ranging from 1 millimeter to 0.7 kilometers. Its diversity offers extensive challenges for model training and evaluation, enhancing the model's adaptability and reasoning capabilities across different environments.

*Table B8.* **Statistics of QA Pairs for different tasks in SpaceVista-1M.**

| Task Category
Scale Distribution | Total
1mm-0.7km | Tiny Tabletop
2mm - 5cm | Tabletop
5cm-2m | Indoor
0.5m-20m | Wild Indoor
0.3m-50m | Outdoor
0.5m-500m | Drone-View
10m-0.7km |
|---|---|---|---|---|---|---|---|
| **All Scenes** | 1,014K | 79K | 242K | 162.5K | 213.3K | 284.3K | 33.1K |
| *General Scenes Tasks* | | | | | | | |
| Position Comparison | 70.5K | – | 10K | 22K | 18K | 20K | 0.5K |
| Size Comparison | 88K | – | 8K | – | 30K | 40K | 10K |
| Existence Estimation | 82K | 15K | 25K | – | 20K | 20K | 2K |
| Rotation Estimation | 85.5K | 18K | 20K | – | 22K | 25K | 0.5K |
| Relative Distance | 81K | – | 24K | 11K | 15K | 30K | 1K |
| Absolute Distance | 99K | – | 25K | 26K | 13K | 34K | 1K |
| Object Counting | 21.3K | – | 1K | 11K | 3.5K | 5.5K | 0.3K |
| Object Size | 157K | 15K | 30K | 33K | 38K | 34K | 7K |
| Route Plan | 2.5K | – | – | - | 1K | 1K | 0.5K |
| Appearance Order | 27.3K | – | 4K | 15K | 3K | 4.5K | 0.8K |
| Depth Estimation | 102K | 19K | 32K | 10K | 15K | 23K | 3K |
| View Change Inference | 51.7K | 6K | 27K | 8K | 4K | 6.5K | 0.2K |
| Object Matching | 102K | 3K | 24K | 12K | 26K | 32K | 5K |
| Spatial Relation | 19K | - | 6K | – | 4K | 8K | 1K |
| *Indoor Scenes Tasks* | | | | | | | |
| Room Size | 15.3K | – | – | 14.5K | 0.8K | – | – |
| *Outdoor Scenes Tasks* | | | | | | | |
| Navigation | 0.8K | – | – | – | – | 0.8K | – |
| *Drone-View Scenes Tasks* | | | | | | | |
| Area Estimation | 0.3K | – | – | – | – | – | 0.3K |
| *Tabletop Scenes Tasks* | | | | | | | |
| Obstacles Location | 3K | – | 3K | – | – | – | – |
| Manipulation Planning | 6K | 3K | 3K | – | – | – | – |

### B.4.7. DATA QUALITY CONTROL

During the construction of our dataset, we distinguish between two notions of answer correctness: **1) strict correctness**, which requires that an answer conform to objective physical reality, and **2) perceptual correctness**, which requires that an answer align with typical human judgments. Since strict correctness is difficult to 1 for training data derived from in-the-wild videos (due to issues like missing calibration, occlusions, and limited metadata), we adopt the perceptual criterion.

Specifically, during validation, we present annotators with both the question and a candidate answer and ask them to judge its acceptability. Consequently, the reported accuracy should be interpreted as agreement with human perception rather than strict fidelity to physical-world quantities or metric scale. For these statistics and the user study, we use MTurk[3] for these statistics and the user study. SpaceVista-1M human checking accuracies are shown in Tab. B9 and SpaceVista-Bench human checking accuracies are shown in Tab. B10. It is important that perceptual correctness is only used in training data quality control, while model evaluation still follows strict correctness.

*Table B9.* **Human checking accuracy over each task category in our training set, SpaceVista-1M.** "∼" means we observe unusual variation for different annotators. The standards for Route Planning, Navigation, and Obstacle Avoidance are **notably stringent**, as these are inherently multi-step processes where a single error can invalidate an entire sample. However, even if a training sample contains minor discrepancies in step, sequence, or distance, the descriptive knowledge within the remaining sections remains valuable for comprehension.

| **Task Categories in SpaceVista-1M for training** | | | | |
|---|---|---|---|---|
| **Task** | Position Comp. | Size Comp. | Existence Est. | Rotation Est. | Relative Dist. |
| **Accuracy** | 95% | 84% | 94% | 95% | 82% |
| **Task** | Room Size | Object Count | Object Size | Route Plan | Appear. Order |
| **Accuracy** | 84% | 87% | 81% | ∼69% | 80% |
| **Task** | View Change | Object Match | Spatial Rel. | Navigation | Area Est. |
| **Accuracy** | 96% | 93% | 95% | ∼67% | 78% |
| **Task** | Manip. Plan | Absolute Dist. | Depth Est. | Obstacles | |
| **Accuracy** | 73% | 84% | 95% | 74% | |

*Table B10.* **Human checking accuracy over each scene's category in our benchmark, SpaceVista-Bench.** Our SpaceVista-Bench, which is annotated by humans, maintains an average accuracy rate exceeding 95%.

| | **Scene Categories in SpaceVista-Bench** | | | |
|---|---|---|---|---|
| **Scenes** | Tiny | Tabletop | Indoor | outdoor |
| **Accuracy** | 98% | 95% | 99% | 93% |

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

# D. Observation Results

## D.1. GRPO Reward Observation

During reinforcement learning training, we observe a relatively stable increase in reward without evidence of reward hacking, as shown in Fig. D22. In most settings, the reward reliably converges within a few thousand environment steps, after which further training yields minimal additional improvements. This suggests that the learning dynamics are well-behaved under our setup and that extending training beyond the convergence point offers limited marginal benefit. Additionally, this may be also be treated as the curve of the amount of data and its performance during post-training.

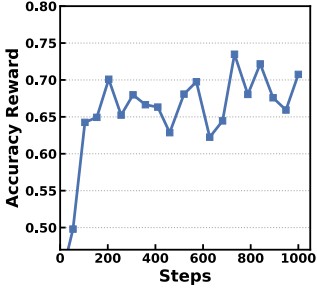

*Figure D22.* **Visualization of GRPO updated and normalized correctness reward chart.** This figure visualizes how the reward grows during the RL training stage.

## D.2. Expert Behavior Observation

We select 10 samples from tiny and indoor scenes and visualize the expert scale distribution in Fig. D23. As shown in Fig. D24, inputs from each scene type tend to activate the expert specialized for that scene. This demonstrates the model's ability to distinguish scene-specific characteristics and allocate resources accordingly. By activating the most relevant expert, the model ensures efficient processing and enhanced performance in scene-specific tasks, highlighting its ability to focus on distinct features and patterns within each scene.

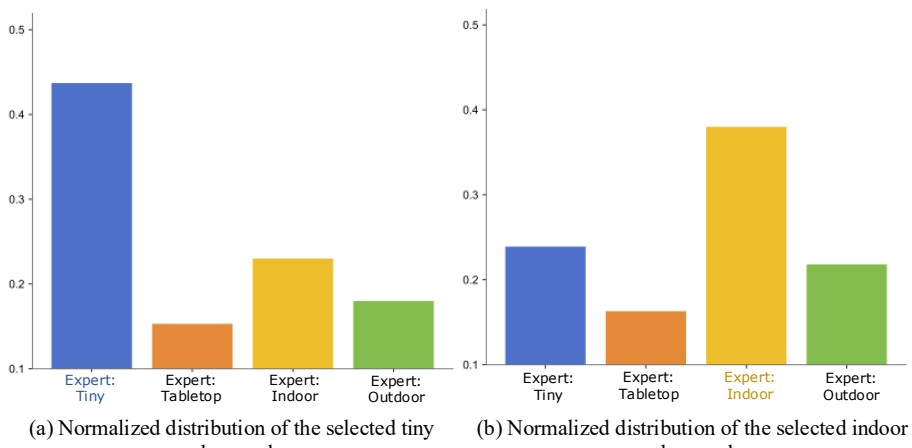

(a) Normalized distribution of the selected tiny scale samples

(b) Normalized distribution of the selected indoor scale samples

*Figure D23.* **Visualization of the normalized scale of each expert with different selected samples.** It reflects the model's capacity to allocate resources according to the inherent properties of each scene.

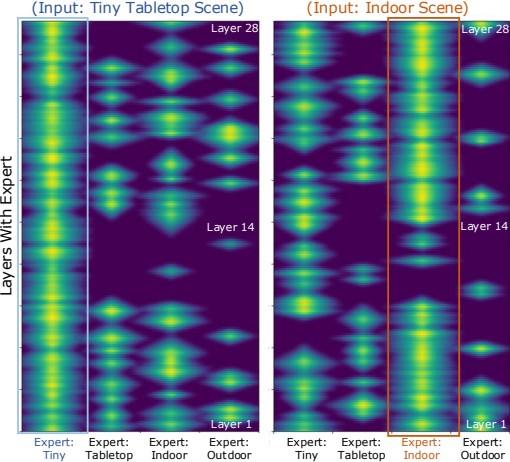

*Figure D24.* **Visualization of scale-expert activations** on salient tokens with an appropriate threshold. This shows the router selects experts based on the input during inference.

## D.3. Memorization effect observation (Out-of-Distribution Problem)

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

### D.4. Benchmark Performance Analysis

We conduct a comprehensive evaluation of SpaceVista-7B across multiple benchmarks, including STI-Bench (Li et al., 2025e), SPAR-Bench (Zhang et al., 2025d), MMSI-Bench (Yang et al., 2025b) and VSI-Bench (Yang et al., 2025a). In this section, we analyze SpaceVista-7B's performance on each benchmark and compare it to other state-of-the-art models. The results from these benchmarks provide a thorough assessment of SpaceVista-7B's spatial reasoning capabilities, highlighting its versatility and adaptability across diverse tasks.

*Table D18.* **Performance comparison of our SpaceVista-7B and other baselines on STI-Bench.** We use **bold** and underlined text for the top two within open-source categories, while ranks are computed across all model categories. In Static Understanding, "Dim. Meas." refers to Dimensional Measurement. In Dynamic Understanding, "Disp. & P.L.", "Speed & Acc.", "Ego Orient.", "Traj. Desc.", and "Pose Est." represent Displacement and Path Length, Speed and Acceleration, Ego-Centric Orientation, Trajectory Description, and Pose Estimation, respectively. This table includes only the popular model for which the detailed scores are available. For average-score comparisons, see Table 2.

| Model/Method | Rank | Avg. | Static Understanding | | | Dynamic Understanding | | | | |
|---|---|---|---|---|---|---|---|---|---|---|
| | | | Dim. Meas. | Spatial Relation | 3D Video Grounding | Disp. & P.L. | Speed & Acc. | Ego Orient. | Traj. Desc. | Pose Est. |
| *Closed-source Models* | | | | | | | | | | |
| GPT-4o (Hurst et al., 2024) | 8 | 34.8 | 27.1 | 51.8 | 29.0 | 23.2 | 35.4 | 33.7 | 32.0 | 53.6 |
| Gemini-2.0-Flash (Deepmind, 2024) | 3 | 38.7 | 31.9 | 50.0 | 31.8 | 27.7 | 32.1 | 10.8 | 38.5 | 61.3 |
| Claude-3.7-Sonnet (Anthropic, 2025a) | 2 | 40.5 | 29.8 | 45.5 | 35.7 | 28.9 | 38.8 | 40.0 | 47.4 | 62.6 |
| Gemini-2.5-Pro (DeepMind, 2025) | 1 | 41.4 | 38.7 | 53.8 | 36.9 | 33.9 | 33.1 | 52.5 | 47.4 | 50.4 |
| *Open-source Models* | | | | | | | | | | |
| VideoLLaMA3-7B (Zhang et al., 2025a) | 7 | 35.2 | 29.4 | 48.6 | 36.1 | 21.5 | 36.7 | 23.2 | **54.6** | 48.1 |
| MiniCPM-V-2.6 (Yao et al., 2024) | 10 | 26.9 | 27.7 | 44.5 | 29.0 | 19.0 | 25.7 | 7.0 | 30.8 | 35.6 |
| VideoChat-R1 (Li et al., 2025d) | 9 | 32.8 | 23.2 | 47.3 | 31.5 | 22.4 | 31.1 | 26.0 | 47.9 | 48.3 |
| InternVL2.5-78B (Chen et al., 2024e) | 4 | 38.5 | 29.9 | **52.8** | 31.6 | 24.9 | 37.2 | **49.2** | 43.6 | **53.6** |
| VideoChat-Flash (Li et al., 2024c) | 6 | 36.3 | **33.6** | 51.4 | 33.1 | **27.1** | 32.3 | 22.2 | 54.2 | 51.4 |
| SpaceVista-7B (Ours) | 5 | 38.2 | 33.1 | 47.2 | **37.6** | 23.6 | **37.3** | 39.6 | 43.1 | 51.2 |

*Table D19.* **Performance comparison of our SpaceVista-7B and other baselines on SPAR-Bench.** We use **bold** and underlined text for the top two within open-source categories, while ranks are computed across all model categories. OO, OC, and MV refer to object-object, object-camera, and multi-view, respectively. This table includes only the popular model for which the detailed scores are available. For average-score comparisons, see Table 2.

| Model/Method | Rank | Avg. | Low | Depth-OC | Depth-OC-MV | Depth-OO | Depth-OO-MV | Dist-OC | Dist-OC-MV | Dist-OO | Dist-OO-MV | Medium | PosMatch | CamMotion | ViewChg‡ | High | Dist-OO | Dist-OO-MV | ObjRel-OC-MV | ObjRel-OO | ObjRel-OO-MV | SpImg-OC | SpImg-OC-MV | SpImg-OO | SpImg-OO-MV |
|---|---|---|---|---|---|---|---|---|---|---|---|---|---|---|---|---|---|---|---|---|---|---|---|---|---|
| *Baseline* | | | | | | | | | | | | | | | | | | | | | | | | | |
| Chance Level (Random) | - | - | - | - | - | - | - | - | - | - | - | - | 22.65 | 24.50 | - | 25.09 | 23.82 | 22.02 | 31.25 | 25.27 | 22.16 | 25.81 | 24.42 | 24.17 | 26.89 |
| Chance Level (Frequency) | 5 | 32.74 | 31.19 | 43.09 | 43.51 | 17.38 | 13.05 | 41.90 | 30.99 | 27.40 | 32.17 | 38.25 | 29.01 | 26.75 | 59.00 | 32.29 | 52.94 | 50.60 | 28.25 | 26.92 | 26.59 | 26.34 | 26.74 | 26.49 | 25.77 |
| *SPAR-Bench(full)* | | | | | | | | | | | | | | | | | | | | | | | | | |
| InternVL2-2B (Chen et al., 2024f) | 12 | 28.06 | 21.74 | 18.06 | 24.81 | 23.20 | **20.97** | 19.47 | 19.95 | 26.83 | 20.61 | 22.83 | 39.69 | 23.00 | 5.81 | 35.42 | 51.18 | 55.95 | 46.00 | 31.59 | 23.82 | 36.02 | 34.30 | 17.55 | 22.41 |
| InternVL2-4B (Chen et al., 2024f) | 6 | 32.01 | 28.94 | 23.94 | 27.22 | 20.00 | 18.12 | 42.57 | 40.16 | 31.29 | 28.18 | 29.16 | 49.87 | 21.00 | 16.62 | 35.70 | 56.76 | 55.36 | 40.25 | 36.81 | 25.21 | 28.76 | 32.27 | 21.19 | 24.65 |
| InternVL2-8B (Chen et al., 2024f) | 4 | 33.02 | 26.83 | 25.75 | 30.88 | 20.76 | 20.78 | 39.03 | 36.19 | 19.15 | 22.19 | **36.49** | **63.36** | 28.00 | 18.11 | 37.37 | 64.71 | 54.46 | 42.75 | 37.36 | 26.32 | 34.14 | 31.10 | 20.86 | 24.65 |
| InternVL2.5-2B (Chen et al., 2024e) | 10 | 30.14 | 25.79 | 39.67 | 39.72 | 12.12 | 15.03 | 30.94 | 29.59 | 20.22 | 19.02 | 22.93 | 37.91 | 24.25 | 6.64 | 36.41 | 51.47 | 56.85 | 50.25 | 33.79 | 24.10 | 27.15 | 35.17 | 26.49 | 22.41 |
| InternVL2.5-4B (Chen et al., 2024e) | 9 | 30.55 | 25.66 | 29.06 | 32.97 | 21.77 | 16.83 | 20.84 | 26.85 | 28.13 | 28.79 | 29.75 | 47.07 | 33.25 | 8.92 | 35.16 | 54.12 | 58.93 | 35.50 | 29.67 | 34.63 | 24.73 | 31.39 | 19.21 | 28.29 |
| InternVL2.5-8B (Chen et al., 2024e) | 2 | 36.28 | 29.46 | 25.78 | 29.31 | 23.79 | 18.76 | 46.82 | 42.68 | 22.62 | 25.89 | 31.88 | 61.32 | 28.00 | 6.32 | 43.80 | 59.71 | 56.85 | 51.75 | 44.23 | 41.55 | 36.56 | 41.57 | 22.52 | 39.50 |
| LLaVA-Onevision-0.5B (Li et al., 2024a) | 11 | 29.48 | 30.14 | 49.22 | 42.72 | 18.04 | 14.92 | 31.48 | 25.67 | 28.98 | 30.10 | 15.89 | 24.43 | 21.75 | 1.50 | 33.42 | 50.88 | 50.00 | 32.00 | 27.75 | 26.04 | 30.91 | 34.01 | 24.50 | 24.65 |
| LLaVA-Onevision-7B (Li et al., 2024a) | 7 | 31.20 | 21.79 | 30.33 | 26.94 | 18.58 | 13.87 | 10.43 | 13.64 | 31.24 | 29.29 | 26.13 | 38.68 | 30.25 | 9.47 | 40.14 | 56.47 | 55.06 | 37.25 | 48.63 | 38.23 | 30.38 | 33.72 | 26.49 | 35.01 |
| Qwen2-VL-2B (Wang et al., 2024b) | 13 | 24.60 | 19.43 | 38.03 | 40.63 | 18.84 | 14.09 | 7.81 | 7.07 | 17.82 | 11.14 | 27.55 | 26.21 | 25.25 | **31.20** | 28.22 | 54.12 | 49.11 | 21.75 | 25.27 | 12.47 | 23.92 | 27.62 | 24.83 | 14.85 |
| Qwen2-VL-7B (Wang et al., 2024b) | 8 | 30.74 | 27.52 | 35.97 | 35.22 | 20.83 | 12.88 | 28.68 | 29.95 | 28.21 | 28.45 | 20.44 | 35.37 | 20.25 | 5.69 | 37.03 | 59.71 | 52.38 | 30.25 | 38.46 | **41.00** | 22.04 | 28.49 | 22.52 | 38.38 |
| Qwen2.5-VL-7b (Bai et al., 2025) | 3 | 33.07 | 28.75 | 31.33 | 33.66 | 21.99 | 14.97 | 42.88 | 37.73 | 23.83 | 23.64 | 22.97 | 33.33 | 28.75 | 6.83 | 40.27 | 58.24 | 51.49 | 44.75 | **50.00** | 32.13 | 33.87 | 32.85 | **27.15** | 31.93 |
| LLaVA-v1.5-7b (Liu et al., 2023a) | 14 | 23.65 | 10.85 | 5.17 | 12.53 | 17.37 | 11.34 | 7.25 | 5.26 | 18.73 | 9.12 | 26.50 | 24.43 | 26.75 | 28.31 | 34.09 | 51.18 | 52.38 | 34.25 | 24.18 | 26.87 | 34.68 | 29.94 | 22.52 | 30.81 |
| LLaVA-v1.6-7b (Liu et al., 2023a) | 15 | 13.21 | 8.53 | 12.14 | 0.00 | 20.35 | 0.27 | 10.76 | 0.41 | 24.27 | 0.00 | 4.79 | 6.62 | 7.75 | 0.00 | 20.18 | 51.76 | 7.74 | 6.25 | 32.14 | 6.37 | **39.52** | 10.47 | 21.52 | 5.88 |
| SpaceVista-7B (Ours) | 1 | 41.68 | **42.51** | **57.78** | **51.94** | **24.44** | 20.22 | **57.02** | **51.12** | **42.62** | **34.98** | **36.02** | 31.04 | **41.00** | 0.00 | **46.82** | **66.76** | **63.10** | **56.5** | **50.00** | **41.55** | **37.10** | **37.21** | **27.15** | **42.02** |

*Table D20.* **Performance Comparison of our SpaceVista-7B and other baselines on MMSI-Bench.** We use **bold** and underlined text for the top two within open-source categories, while ranks are computed across all model categories. Cam., Obj., Reg., Meas., and Appr. denote Camera, Object, Region, Measurement, and Appearance, respectively. This table includes only the popular model for which the detailed scores are available. For average-score comparisons, see Table 2.

| Model/Method | Rank | Avg. | Positional Relationship | | | | | | Attribute | | Motion | | MSR |
|---|---|---|---|---|---|---|---|---|---|---|---|---|---|
| | | | Cam.-Cam. | Obj.-Obj. | Reg.-Reg. | Cam.-Obj. | Obj.-Reg. | Cam.-Reg. | Meas. | Appr. | Cam. | Obj. | – |
| *Baseline* | | | | | | | | | | | | | |
| Blind GPT-4o | 32 | 22.7 | 20.2 | 17.0 | 29.6 | 13.9 | 29.4 | 19.2 | 21.8 | 12.1 | 20.2 | 29.0 | 20.2 |
| Random Guessing | 29 | 25.0 | 25.0 | 25.0 | 25.0 | 25.0 | 25.0 | 25.0 | 25.0 | 25.0 | 25.0 | 25.0 | 25.0 |
| Human Level | 1 | 97.2 | 95.7 | 98.9 | 97.5 | 94.2 | 98.8 | 96.4 | 95.3 | 98.5 | 98.6 | 98.7 | 97.0 |
| *Closed-source Models* | | | | | | | | | | | | | |
| o3 (OpenAI, 2025b) | 2 | 41.0 | 45.2 | 39.4 | 37.0 | 44.2 | 47.1 | 62.6 | 54.7 | 28.8 | 31.1 | 32.9 | 34.9 |
| GPT-4.5 (OpenAI, 2025a) | 3 | 40.3 | 34.4 | 29.8 | 39.5 | 51.2 | 47.1 | 55.4 | 39.1 | 33.3 | 41.9 | 40.8 | 36.4 |
| GPT-4o (Hurst et al., 2024) | 7 | 30.3 | 34.4 | 24.5 | 23.5 | 19.8 | 37.6 | 27.7 | 32.8 | 31.8 | 35.1 | 36.8 | 30.8 |
| Gemini-2.5-Pro (DeepMind, 2025) | 4 | 36.9 | 39.7 | 31.9 | 39.5 | 45.3 | 35.2 | 43.3 | 51.5 | 21.2 | 36.4 | 30.2 | 34.3 |
| Claude-3.7-Sonnet (Anthropic, 2025a) | 10 | 28.7 | 32.3 | 26.6 | 22.2 | 34.9 | 37.6 | 42.2 | 25.0 | 22.7 | 21.6 | 32.9 | 22.7 |
| Seed1.5-VL (Guo et al., 2025a) | 8 | 29.7 | 32.2 | 30.8 | 30.8 | 23.2 | 38.8 | 32.5 | 39.0 | 21.2 | 36.4 | 25.0 | 26.2 |
| *Open-source Models* | | | | | | | | | | | | | |
| InternVL3-78B (Zhu et al., 2025) | 12 | 28.5 | 34.4 | 23.4 | 32.1 | 12.8 | 37.6 | 26.5 | 37.5 | 19.7 | **28.4** | 31.6 | 29.3 |
| InternVL2.5-78B (Chen et al., 2024e) | 12 | 28.5 | 23.7 | 22.3 | **39.5** | 29.1 | 31.8 | **42.2** | 35.9 | 19.7 | 17.6 | 26.3 | 27.3 |
| Qwen2.5-VL-72B (Bai et al., 2025) | 5 | 30.7 | 25.8 | 34.0 | 34.6 | 23.3 | 34.1 | 36.1 | **45.3** | 27.3 | 27.0 | 30.3 | 27.3 |
| LLaVA-OneVision-72B (Li et al., 2024a) | 13 | 28.4 | **43.0** | 31.9 | 33.3 | 30.2 | 37.6 | 38.6 | 28.1 | 19.7 | 13.5 | 32.9 | 15.7 |
| InternVL3-38B (Zhu et al., 2025) | 23 | 26.3 | 21.5 | 20.2 | 33.3 | 23.3 | 35.3 | 25.3 | 39.1 | 21.2 | 16.2 | 31.6 | 25.8 |
| InternVL2.5-38B (Chen et al., 2024e) | 16 | 27.9 | 18.3 | 22.3 | 35.8 | 22.1 | 38.8 | 34.9 | 37.5 | 25.8 | 14.9 | 38.2 | 25.3 |
| Qwen2.5-VL-32B (Bai et al., 2025) | 17 | 27.7 | 24.7 | 26.6 | 29.6 | 22.1 | 32.9 | 31.3 | 31.2 | 24.2 | 18.9 | 35.5 | 27.8 |
| InternVL2.5-26B (Chen et al., 2024e) | 15 | 28.0 | 24.7 | 19.1 | 29.6 | 33.7 | 31.8 | 37.3 | 35.9 | 30.3 | 10.8 | 31.6 | 26.8 |
| NVILA-15B (Liu et al., 2024c) | 6 | 30.5 | 30.1 | **39.4** | 28.4 | 36.0 | **38.8** | 20.5 | 29.7 | **31.8** | 18.9 | 35.5 | 27.8 |
| InternVL3-14B (Zhu et al., 2025) | 20 | 26.8 | 19.4 | 24.5 | 24.7 | 23.3 | 37.6 | 24.1 | 31.2 | 22.7 | 24.3 | 31.6 | 29.3 |
| Llama-3.2-11B-Vision (Grattafiori et al., 2024) | 27 | 25.4 | 25.8 | 30.8 | 32.0 | 25.6 | 21.2 | 25.9 | 20.3 | 19.7 | 25.6 | 28.9 | 19.2 |
| InternVL3-9B (Zhu et al., 2025) | 21 | 26.7 | 18.3 | 25.5 | 32.1 | 29.1 | 31.8 | 22.9 | 29.7 | 24.2 | 16.2 | 38.2 | 26.8 |
| InternVL3-8B (Zhu et al., 2025) | 26 | 25.7 | 25.8 | 31.9 | 17.0 | 25.6 | 35.3 | 28.9 | 23.4 | 24.2 | 16.2 | 32.9 | 14.6 |
| InternVL2.5-8B (Chen et al., 2024e) | 10 | 28.7 | 32.3 | 27.7 | 29.6 | 32.6 | 24.7 | 32.5 | 26.6 | 27.3 | 16.2 | 31.6 | 30.3 |
| NVILA-8B (Liu et al., 2024c) | 14 | 28.1 | 17.2 | 29.8 | 24.7 | 30.2 | 22.4 | 34.9 | 34.4 | 25.8 | 25.7 | 34.2 | 29.8 |
| Qwen2.5-VL-7B (Bai et al., 2025) | 25 | 25.9 | 24.7 | 24.5 | 24.7 | 25.6 | 29.4 | 26.5 | 25.0 | 18.2 | 20.3 | **39.5** | 25.8 |
| LLaVA-OneVision-7B (Li et al., 2024a) | 30 | 24.5 | 20.4 | 33.0 | 29.6 | 29.1 | 25.9 | 30.1 | 29.7 | 25.8 | 18.9 | 34.2 | 11.6 |
| InternVL2.5-4B (Chen et al., 2024e) | 23 | 26.3 | 31.2 | 23.4 | 21.0 | 31.4 | 34.1 | 25.3 | 23.4 | 24.2 | 13.5 | 31.6 | **36.8** |
| Qwen2.5-VL-3B (Bai et al., 2025) | 22 | 26.5 | 26.9 | 27.7 | 30.9 | 29.1 | 28.2 | 34.9 | 31.2 | 16.7 | 17.6 | 27.6 | 23.2 |
| InternVL3-2B (Zhu et al., 2025) | 28 | 25.3 | 26.9 | 25.5 | 29.6 | 31.4 | 28.2 | 27.7 | 26.6 | 22.7 | 12.2 | 23.7 | 23.7 |
| InternVL2.5-2B (Chen et al., 2024e) | 9 | 29.0 | 28.0 | 27.7 | 24.7 | **37.2** | 29.4 | 36.1 | **43.8** | 15.2 | 21.6 | 31.6 | 26.8 |
| InternVL3-1B (Zhu et al., 2025) | 19 | 27.0 | 24.7 | **35.1** | 22.2 | 30.2 | 29.4 | 30.1 | 32.8 | 28.8 | 17.6 | 19.7 | 26.3 |
| InternVL2.5-1B (Chen et al., 2024e) | 24 | 26.1 | 23.7 | 26.6 | 24.7 | 25.6 | 31.8 | 25.3 | 31.2 | 30.3 | 17.6 | 25.0 | 26.3 |
| DeepSeek-VL2 (Wu et al., 2024b) | 18 | 27.1 | 23.7 | 31.9 | 22.2 | 36.0 | 30.6 | 22.9 | 28.1 | 15.2 | **28.4** | 26.3 | 28.3 |
| DeepSeek-VL2-Small (Wu et al., 2024b) | 11 | 28.6 | 24.7 | 28.7 | 18.5 | 33.7 | **38.8** | 27.7 | 28.1 | **33.3** | 24.3 | 25.0 | 29.8 |
| DeepSeek-VL2-Tiny (Wu et al., 2024b) | 31 | 24.0 | 29.0 | 27.7 | 21.0 | 23.3 | 17.6 | 31.3 | 14.1 | 24.2 | 14.9 | 25.0 | 27.3 |
| SpaceVista-7B (Ours) | 5 | 30.7 | 26.9 | 23.2 | 30.9 | **45.3** | 27.1 | 36.1 | 34.4 | 26.7 | 23.3 | 35.5 | 25.8 |

*Table D21.* **Performance comparison of our SpaceVista-7B and other baselines on VSI-Bench.** We use **bold** and underlined text for the top two within open-source categories, while ranks are computed across all model categories. This table includes only the popular model for which the detailed scores are available. For average-score comparisons, see Table 2.

| Model / Method | Rank | Avg. | Obj Appear ance Order | Object Abs Distance | Object Counting | Object Rel Distance | Object Size Estimation | Room Size Estimation | Route Planning | Object Rel Direction |
|---|---|---|---|---|---|---|---|---|---|---|
| *Proprietary Models(API)* | | | | | | | | | | |
| GPT-4o(Hurst et al., 2024) | 10 | 34.0 | 28.5 | 5.3 | 46.2 | 37.0 | 43.8 | 38.2 | 31.5 | 41.3 |
| Gemini-1.5 Flash (API)(Team et al., 2024) | 3 | 42.1 | 37.8 | 30.8 | 49.8 | 37.7 | 53.5 | 54.4 | 31.5 | 41.0 |
| Gemini-1.5 Pro (API)(Team et al., 2024) | 2 | 45.4 | 34.6 | 30.9 | 56.2 | 51.3 | 64.1 | 43.6 | 36.0 | 46.3 |
| *Open-source Models* | | | | | | | | | | |
| InternVL2-2B(Chen et al., 2024f) | 16 | 26.5 | 6.3 | 24.0 | 25.7 | 32.1 | 20.0 | 29.2 | 30.4 | 44.1 |
| InternVL2-8B(Chen et al., 2024f) | 6 | 37.5 | 46.4 | 29.0 | 31.3 | 38.0 | 48.9 | **44.2** | 28.9 | 33.4 |
| InternVL2-40B(Chen et al., 2024f) | 7 | 37.0 | 44.7 | 26.2 | 41.3 | **47.6** | 48.2 | 27.5 | 27.8 | 32.7 |
| LongVILA-8B(Chen et al., 2024d) | 17 | 21.6 | 25.5 | 9.1 | 29.1 | 29.6 | 16.7 | 0.0 | 32.5 | 30.7 |
| VILA-1.5-8B(Lin et al., 2023) | 14 | 28.9 | 24.8 | 21.8 | 17.4 | 32.1 | 50.3 | 18.8 | 31.0 | 34.8 |
| VILA-1.5-40B(Lin et al., 2023) | 12 | 31.2 | 32.9 | 24.8 | 22.4 | 40.5 | 48.7 | 22.7 | 31.5 | 25.7 |
| LongVA-7B(Zhang et al., 2024a) | 13 | 29.2 | 15.7 | 16.6 | 38.0 | 33.1 | 38.9 | 22.2 | 25.4 | 43.3 |
| LLaVA-Video-7B(Zhang et al., 2024c) | 8 | 35.6 | 30.6 | 14.0 | 48.5 | 43.5 | 47.8 | 24.2 | 34.0 | 42.4 |
| LLaVA-Video-72B(Zhang et al., 2024c) | 4 | 40.9 | 48.6 | 22.8 | 48.9 | 42.4 | 57.4 | 35.3 | 35.0 | 36.7 |
| LLaVA-NeXT-Video-7B(Zhang et al., 2024b) | 8 | 35.6 | 30.6 | 14.0 | 48.5 | 43.5 | 47.8 | 24.2 | 34.0 | 42.4 |
| LLaVA-NeXT-Video-72B(Zhang et al., 2024b) | 4 | 40.9 | 48.6 | 22.8 | 48.9 | 42.4 | 57.4 | 35.3 | 35.0 | 36.7 |
| LLaVA-OneVision-0.5B(Li et al., 2024a) | 15 | 28.0 | 5.8 | 28.4 | 46.1 | 28.3 | 15.4 | 28.3 | 34.5 | 36.9 |
| LLaVA-OneVision-7B(Li et al., 2024a) | 11 | 32.4 | 24.4 | 20.2 | 47.7 | 42.5 | 47.4 | 12.3 | 29.4 | 35.2 |
| LLaVA-OneVision-72B(Li et al., 2024a) | 5 | 40.2 | 44.6 | 23.9 | 43.5 | 42.5 | 57.6 | 37.5 | 32.5 | 39.9 |
| Qwen2.5-VL-7B (Bai et al., 2025) | 9 | 34.4 | 32.7 | 17.5 | 34.0 | 35.8 | 51.9 | 36.6 | 29.4 | 37.7 |
| SpaceVista-7B (Ours) | 1 | 48.6 | **56.3** | **36.0** | **62.9** | 44.2 | **58.1** | 42.0 | **38.9** | **49.7** |

On **STI-Bench**, SpaceVista-7B ranks fifth overall and exhibits strong performance on 3D video grounding as well as speed and acceleration estimation. It achieves 37.6% on 3D video grounding and 37.3% on speed-related tasks. Gemini-2.5-Pro (DeepMind, 2025) attains the highest average score of 41.4%, followed by Claude-3.7-Sonnet (Anthropic, 2025a). In contrast, Ego-Centric Orientation, Trajectory Description, and Displacement and Path Length remain highly challenging, as they require accurate modeling of egocentric camera motion, long-range temporal integration, and stable 3D reasoning under viewpoint changes and occlusions. Dynamic, long-term spatiotemporal reasoning remains a challenge for current vision-language models. The evaluation results are presented in Tab. D18.

SpaceVista-7B attains the highest overall performance among all compared models on **SPAR-Bench**, with an average accuracy of 41.68% and rank 1. SPAR-Bench evaluates spatial compositional reasoning over object–object(OO), object–camera(OC), and multi-view(MV) relations under low, medium, and high difficulty settings. Across all difficulty levels, SpaceVista-7B consistently ranks within the top two, and on the most challenging OC and MV subsets it reaches up to 66.76%, indicating robust modeling of complex object–camera relations under large viewpoint changes, as summarized in Tab. D19. Meanwhile, most OO subsets remain highly challenging for all models, and reasoning about fine-grained multi-object spatial relations in heavily occluded scenes with subtle depth and ordering differences is still problematic.

On **MMSI-Bench**, SpaceVista-7B achieves an average accuracy of 30.7% and ranks fifth overall, representing the strongest performance among all open-source models. It performs particularly well on positional-relationship tasks, such as camera–object reasoning with 45.3%, and maintains competitive results on attribute and motion categories, as summarized in Tab. D20, indicating a reasonably balanced multi-dimensional spatial understanding. Nevertheless, all models, including SpaceVista-7B, remain far below the human upper bound of 97.0%, and sub-tasks involving camera motion and the composite MSR metric are still notably difficult.

Finally, in the **VSI-Bench** evaluation, SpaceVista-7B outperforms all other models, excelling in object counting, appearance sequencing, and absolute distance tasks, achieving 62.9% in object counting and 36.0% in absolute distance, surpassing several open-source models, including LLaVA-Video-72B (Zhang et al., 2024c) and LLaVA-OneVision-72B (Li et al., 2024a). The results of this evaluation are shown in Tab. D21.

*Table D22.* **Comparison on general understanding ability on popular video benchmark Video-MME.**

| Model | Video-MME |
|---|---|
| VideoLLaMA2 | 47.9 |
| LLaVA-OneVision-7B | 58.2 |
| Qwen2.5VL-7B | 63.8 |
| InternVL3-8B | 65.3 |
| VG-LLM-8B (Spatial Model) | 59.3 |
| Qwen2.5VL-7B ($w/.$ 1/5 SpaceVista-1M) | 59.1 |
| SpaceVista-7B (Spatial Model) | 59.6 |

In general, breakthroughs in specialized domains tend to lead to a decline in general VLM capabilities. This phenomenon has been widely explored in mathematical reasoning, code reasoning, and spatial reasoning. To analyze general ability, we evaluate the performance of SpaceVista-7B on the widely accepted video benchmark Video-MME (Fu et al., 2025). Video-MME is a full-spectrum, multi-modal benchmark of MLLMs in general video analysis. The comparison is shown as Tab. D22.

Therefore, we consider our SpaceVista general ability comparable, and also don't believe it has "lost" general ability or merely follows a preset spatial template. It is still undeniable that specialist models are inspiring for future explorations of general MLLMs.

### D.5. Challenging Scenario Analysis

*Table D23.* **Results analysis of different scenes.** The model mentioned below is trained in a balanced subset of SpaceVista-1M for better control of experiment conditions.

| Model | SpaceVista-Bench (Ours) | | | |
|---|---|---|---|---|
| | Indoor | Outdoor | Tabletop | Tabletop |
| Qwen2.5-VL-7B | 30.34 | 18.31 | 23.79 | 19.37 |
| $w/.$ balance training | 38.77 | 24.90 | 30.17 | 20.86 |

When testing scenes at varying scales, several critical questions arise: Which scenarios pose greater challenges, and to what extent is data complexity the primary bottleneck? To systematically investigate these issues, we design a controlled observational experiment.

We identify tasks that exhibit consistent properties across different scales, including object size, object comparison, absolute and relative distance, and depth estimation. For fairness in comparison, we train models using videos from diverse scenes while maintaining similar quantities of QA pairs and video samples. Under these controlled conditions, we evaluate and compared performance across different scale-dependent scenarios. In Tab.D23, it seems indoor data is the easiest task. We hypothesize that a human-scale estimation bias—arising because both humans and GPT focus on objects expressible in basic units like meters in pretraining corpora—leads to this preference.

### D.6. 3D Geometric Feature Observation

*Table D24.* **Comparison of the robustness of the model training of 3D and 2.5D.** All the models are trained on 3D or 2.5D data along with the video. However, we vary the evaluation input of these models to see the robustness. "–" denotes experiments we consider unnecessary. "low" means using low resolution visual for 3D reconstruction. This table includes only the popular model for which a detailed score is available. For average-score comparisons, see Table 2. "($n\%$)" means the relative decrease compared to the original input.

| Settings | Eval Input | VSI-bench | SpaceVista-Bench |
|---|---|---|---|
| | visual $w/.$ 3D | 44.3 | 31.4 |
| Training with $w/.$ 3D | visual $w/.$ 3D (low) | 38.1 (-14%) | – |
| | visual $w/o.$ 3D | 34.0 (-23%) | – |
| | visual $w/.$ 2.5D | 45.6 | 33.0 |
| Training with $w/.$ 2.5D | visual $w/.$ 2.5D (low) | 43.9 (-4%) | 32.3 (-2%) |
| | visual $w/o.$ 2.5D | 40.7 (-10%) | 29.1 (-12%) |

In addition to introducing VGGT(Wang et al., 2025a) and DINO v3(Siméoni et al., 2025) as extra signals, we conduct a series of targeted ablation studies. This suggests that representation formats like VGGT, when used in their native encoder output, are wonderful for capturing geometry information, but suboptimal for capturing semantic information or overall scenes, especially for low resolution and uncommon scenarios. In Tab.D24, we use "3D" to denote the pure geometric features from VGGT, and "2.5D" to denote the additional 12 viewing angles of the overall scene rendered by the decoder and the renderer. We use the special prompt and the image token to provide

As shown in Tab.D24, 2.5D is usually more robust in spatial reasoning. Rendering to 2.5D enables effective exploitation of pretrained image tokenizers, which in turn provides more reliable semantic information.

Below is the special prompt for 2.5D finetuning.

"*Please think about this question as if you were a human pondering deeply. Consider detailed information from the video frames and coarse spatial information from the 3D point cloud image. Provide the model's thought process and reasoning between the <think> </think> tags, and give your final answer between the <answer> </answer> tags. <video> The images below are obtained from the 3D point clouds based on the video frames above. The following point cloud images are randomly selected viewpoints; some may be completely unhelpful, while others may contain important information. Please discern carefully. <image> Provide your reasoning between the <think> </think> tags and your final answer between the <answer> </answer> tags.*"

### D.7. Scaling-Up Analysis

We investigate prospective scaling behavior across three model sizes—3B, 7B, and 32B—to inform future model development. Our analysis is conducted using the same SpaceVista-1M dataset while holding all model settings nearly constant. However, there is a minor difference between different scale models. We use LoRA rather than full scale to finetune 32B model. Since using more experts will inevitably increase the inference time, we use fewer experts as the scale increases. However, we still hold the strong belief that it does not affect the overall scaling exploration of our SpaceVista-1M data and model.

*Table D25.* **Scaling model with SpaceVista-1M.** "Qwen2.5-VL-*B" indicates that the SFT model used for evaluation is trained on the corresponding base model.

| Foundation Model | Qwen2.5-VL-3B | Qwen2.5-VL-7B | Qwen2.5-VL-32B |
|---|---|---|---|
| VSI-Bench | 43.5 | 46.3 | 49.0 |
| SpaceVista-Bench | 29.5 | 35.4 | 36.3 |

As summarized in Tab. D25, the dataset affords a certain degree of support for the 32B model's capabilities. Nevertheless, beyond this observation, the main results are achieved by the 7B configuration, whereas ablation studies are primarily conducted with the 3B model.

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

### F.3. QA Preview

We provide a comprehensive set of SpaceVista-1M QA pairs here for preview in Tab.F30-Tab.F48. Note that the RL-oriented multiple-choice and regression formats omit anchors like `<semantic>` and `<scale>`, since they can be easily injected during training from the meta information. Since if objects are referred to by a bounding box, the only changes needed are to change the object name into the corresponding object point/bbox/mask. Each question takes only one video with one form of referring. For example, *"Where is the toothbrush relative to the keyboard from the view of the start frame?"* → *"Where is the red mask referred object relative to the keyboard from the view of the start frame?"*. So, in this preview, we only provide the natural language questions for clarity.

Overall, these previews highlight the diversity of our all-scale reasoning SpaceVista-1M dataset.

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

Engage in an internal dialogue using expressions such as 'let me think',
'wait', 'hmm', 'oh, I see', 'let's break it down', etc, or other natural
language thought expressions. It's encouraged to include self-reflection.
Video: <video>
Question: <text>
During RL: Please provide the thinking process within the <think> </think> tags. Please provide only the single option letter
(e.g., A, B, C, D, etc.) within the <answer> </answer> tags.

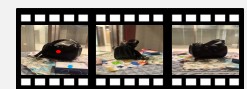

---

**Text Input**
What is the most likely spatial relationship between the red point and the blue point?
Options: A.Stacking  B. Dhesion  C. Support  D. Adjacent  E. Plug-in

| **Bounding Box Input** | **Point Input** | **Mask Input** |
|---|---|---|
| [{"x1":91, "y1":181, "x2":439, "y2":458}, {"x1":241, "y1":471, "x2":479, "y2":627}] | [[328, 287], [548, 360]] | |

---

**SFT GT**
<think>Based on the spatial relationship in the image, the red point appears to be positioned below or supporting the blue point. Given their relative arrangement and visual positioning, the red point is most likely the foundation or support for the blue point.</think> <answer>The red point likely serves as the foundation or support for the blue point.</answer>
**RL GT**
<think>*</think><answer>D</answer>

*Table F31.* **The camera moving task QA preview.**

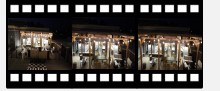

**Camera Moving Task**

Please think about this question as if you were a human pondering deeply.
Engage in an internal dialogue using expressions such as 'let me think',
'wait', 'hmm', 'oh, I see', 'let's break it down', etc, or other natural
language thought expressions. It's encouraged to include self-reflection.
Video: `<video>`
Question: `<text>`
During RL: Please provide the thinking process within the <think> </think> tags. Please provide only the single option letter
(e.g., A, B, C, D, etc.) within the <answer> </answer> tags.

**Text Input**
What directional path did the camera follow in the video, ignoring rotation?
Options: A.Front  B. Back  C. Left  D. Right  E: Down

| **Bounding Box Input** | **Point Input** | **Mask Input** |
|---|---|---|
| Not applicable. | Not applicable. | Not applicable. |

**SFT GT**
<think>The images show a sequence of shots from the same scene, with each frame zooming in slightly more. This indicates
that the camera is moving forward towards the structure, capturing more detailed parts of the scene as it progresses. The
camera keeps moving forward.</think> <answer>Front</answer>
**RL GT**
<think>*</think><answer>A</answer>

*Table F32.* **The position comparison task QA preview.**

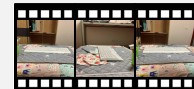

**Position Comparison Task**

Please think about this question as if you were a human pondering deeply.
Engage in an internal dialogue using expressions such as 'let me think',
'wait', 'hmm', 'oh, I see', 'let's break it down', etc, or other natural
language thought expressions. It's encouraged to include self-reflection.
Video: `<video>`
Question: `<text>`
During RL: Please provide the thinking process within the <think> </think> tags. Please provide only the single option letter
(e.g., A, B, C, D, etc.) within the <answer> </answer> tags.

**Text Input**
Where is the toothbrush relative to the keyboard from the view of the start frame?
Options: A.Left  B. Right  C. Front  D. Back

| **Bounding Box Input** | **Point Input** | **Mask Input** |
|---|---|---|
| Not applicable. | [237, 9] | 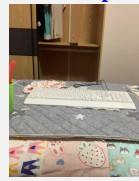 |

**SFT GT**
<think>The video starts with the toothbrush on the left side in the green cup. As the view rotates, its position shifts from left
to right and then back to left. Before disappearing, it remains on the left side. So the toothbrush is on the left from the view
of the start frame.</think> <answer>Left</answer>
**RL GT**
<think>*</think><answer>A</answer>

*Table F33.* **The size comparison task QA preview.**

**Size Comparison Task**

Please think about this question as if you were a human pondering deeply.
Engage in an internal dialogue using expressions such as 'let me think',
'wait', 'hmm', 'oh, I see', 'let's break it down', etc, or other natural
language thought expressions. It's encouraged to include self-reflection. 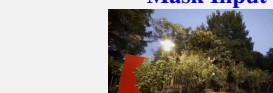
Video: `<video>`
Question: `<text>`
During RL: Please provide the thinking process within the <think> </think> tags. Please provide only the single option letter
(e.g., A, B, C, D, etc.) within the <answer> </answer> tags.

**Text Input**
Based on the height of the referred object shown in the video, which is taller: the red point object or the blue point object?
Options: A.Streetlight B. Sign

| **Bounding Box Input** | **Point Input** | **Mask Input** |
|---|---|---|
| `[{"x1":17, "y1":110,` `"x2":94, "y2":248},` `{"x1":126, "y1":74,` `"x2":163, "y2":220}]` | `[[185, 52], [101, 144]]` | |

**SFT GT**
<think>By analyzing the position of the light source in the image, it's clear that the streetlight is situated above the sign.
Additionally, due to its height and the relatively distant position, the streetlight has a broader light coverage, further suggesting
it is taller than the sign.</think> <answer>Streetlight.</answer>
**RL GT**
<think>*</think><answer>A</answer>

*Table F34.* **The existence estimation task QA preview.**

**Existence Estimation Task**

Please think about this question as if you were a human pondering deeply.
Engage in an internal dialogue using expressions such as 'let me think',
'wait', 'hmm', 'oh, I see', 'let's break it down', etc, or other natural
language thought expressions. It's encouraged to include self-reflection.
Video: `<video>`
Question: `<text>`
During RL: Please provide the thinking process within the <think> </think> tags. Please provide only the single option
letter (e.g., A, B, C, D, etc.) within the <answer> </answer> tags.

**Text Input**
Determine which objects in the cabinet, printer, dishwasher, and trash bin are found in the footage.
Options: A.Printer B. Cabinet C. Dishwasher D. Trash bin

| **Bounding Box Input** | **Point Input** | **Mask Input** |
|---|---|---|
| Not applicable. | Not applicable. | Not applicable. |

**SFT GT**
<think>Based on the content in the images, I can see a clearly visible trash bin placed on the ground. Its presence is
obvious in the scene, as it is clearly shown, and other objects in the image provide a reference for its position and shape.
Therefore, it is reasonable to infer the existence of the trash bin.</think> <answer>Trash bin</answer>
**RL GT**
<think>*</think><answer>D</answer>

*Table F35.* **The rotation estimation task QA preview.**

**Rotation Estimation Task**

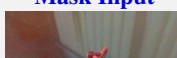

Please think about this question as if you were a human pondering deeply. Engage in an internal dialogue using expressions such as 'let me think', 'wait', 'hmm', 'oh, I see', 'let's break it down', etc, or other natural language thought expressions. It's encouraged to include self-reflection.
Video: <video>
Question: <text>
During RL: Please provide the thinking process within the <think> </think> tags. Please provide the numerical value (e.g., 42 or 3.14) within the <answer> </answer> tags.

**Text Input**
Calculate the rotation angle of the object in the red bounding box between the first and last frame.

| **Bounding Box Input** | **Point Input** | **Mask Input** |
| --- | --- | --- |
| `[{"x1":911, "y1":605,`
`"x2":1201, "y2":853}]` | `[[480, 989]]` | 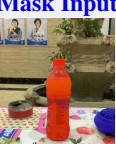 |

**SFT GT**
<think>By observing the change in the object's position between the frames, I estimate its rotation angle. Based on the relative rotation in the two images and using reference points, I approximate the object has rotated by about 136 degrees. The exact angle might vary slightly, but it's within this range.</think> <answer>136 degrees</answer>
**RL GT**
<think>*</think><answer>136</answer>

*Table F36.* **The relative distance task QA preview.**

**Relative Distance Task**

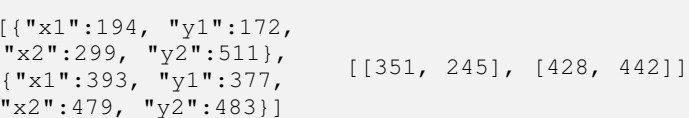

Please think about this question as if you were a human pondering deeply. Engage in an internal dialogue using expressions such as 'let me think', 'wait', 'hmm', 'oh, I see', 'let's break it down', etc, or other natural language thought expressions. It's encouraged to include self-reflection.
Video: <video>
Question: <text>
During RL: Please provide the thinking process within the <think> </think> tags. Please provide the numerical value (e.g., 42 or 3.14) within the <answer> </answer> tags.

**Text Input**
How close does the red mask get to the blue mask in the frames (in centimeters)?

| **Bounding Box Input** | **Point Input** | **Mask Input** |
| --- | --- | --- |
| `[{"x1":194, "y1":172,`
`"x2":299, "y2":511},`
`{"x1":393, "y1":377,`
`"x2":479, "y2":483}]` | `[[351, 245], [428, 442]]` | |

**SFT GT**
<think>Based on the content of frames, the relative position of the red and blue objects changes minimally. The bottle seems 20 centimeters tall. Therefore, the relative distance between them in the frames is estimated to be approximately 7.8 centimeters. The contrast between their shapes and background positions helps in making this estimation.</think> <answer>8 centimeters</answer>
**RL GT**
<think>*</think><answer>7.8</answer>

*Table F37.* **The absolute distance task QA preview.**

**Absolute Distance Task**

Please think about this question as if you were a human pondering deeply.
Engage in an internal dialogue using expressions such as 'let me think',
'wait', 'hmm', 'oh, I see', 'let's break it down', etc, or other natural
language thought expressions. It's encouraged to include self-reflection.
Video: <video>
Question: <text>
During RL: Please provide the thinking process within the <think> </think> tags. Please provide the numerical value (e.g.,
42 or 3.14) within the <answer> </answer> tags.

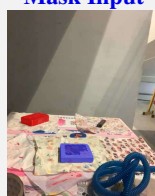

**Text Input**
What is the estimated distance between the red mask and the blue mask in the video (in meters)?

| **Bounding Box Input** | **Point Input** | **Mask Input** |
|---|---|---|
| `[{"x1":250, "y1":228,`
`"x2":307, "y2":314},`
`{"x1":402, "y1":229,`
`"x2":479, "y2":300}]` | `[[270, 278], [264, 442]]` | 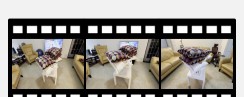 |

**SFT GT**
<think>The red and blue objects are two regular boxes of around 0.1m in size. Based on the position of the red and blue
objects in the image, the estimated absolute distance between them is approximately 0.31 meters. This estimation is made
based on the size and arrangement of the objects in the frame.</think> <answer>0.31 meters.</answer>
**RL GT**
<think>*</think><answer>0.31</answer>

*Table F38.* **The room size task QA preview.**

**Room Size Task**

Please think about this question as if you were a human pondering deeply.
Engage in an internal dialogue using expressions such as 'let me think',
'wait', 'hmm', 'oh, I see', 'let's break it down', etc, or other natural
language thought expressions. It's encouraged to include self-reflection.
Video: <video>
Question: <text>
During RL: Please provide the thinking process within the <think> </think> tags. Please provide the numerical value (e.g.,
42 or 3.14) within the <answer> </answer> tags.

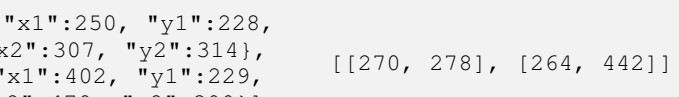

**Text Input**
Give an estimate of the room size based on the video (in square meters).

| **Bounding Box Input** | **Point Input** | **Mask Input** |
|---|---|---|
| Not applicable. | Not applicable. | Not applicable. |

**SFT GT**
<think>By observing the spatial layout in the image, I estimate the room's size. Based on the relative proportions and
perspective of the table and sofa within the room, and considering typical room size proportions with normal table and sofa
size, the area of the room is approximately 24.33 square meters. This estimation is derived from the visible space in the
image.</think> <answer>24.33 square meters</answer>
**RL GT**
<think>*</think><answer>24.33</answer>

*Table F39.* **The object counting task QA preview.**

**Object Counting Task**

Please think about this question as if you were a human pondering deeply.
Engage in an internal dialogue using expressions such as 'let me think',
'wait', 'hmm', 'oh, I see', 'let's break it down', etc, or other natural
language thought expressions. It's encouraged to include self-reflection.
Video: `<video>`
Question: `<text>`
During RL: Please provide the thinking process within the <think> </think> tags. Please provide the numerical value (e.g., 42 or 3.14) within the <answer> </answer> tags.

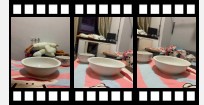

---

**Text Input**
How many bowls can be seen throughout the video?

| **Bounding Box Input** | **Point Input** | **Mask Input** |
|---|---|---|
| `[{"x1":81, "y1":357,`
`"x2":408, "y2":539}]` | `[[439, 243]]` | |

---

**SFT GT**
<think>Based on the objects in the image, two bowls are clearly visible in different positions. Their number is easily identifiable in the frame. Therefore, it can be estimated that there are 2 bowls visible in the video.</think> <answer>2 bowls</answer>
**RL GT**
<think>*</think><answer>2</answer>

*Table F40.* **The object size task QA preview.**

**Object Size Task**

Please think about this question as if you were a human pondering deeply.
Engage in an internal dialogue using expressions such as 'let me think',
'wait', 'hmm', 'oh, I see', 'let's break it down', etc, or other natural
language thought expressions. It's encouraged to include self-reflection.
Video: `<video>`
Question: `<text>`
During RL: Please provide the thinking process within the <think> </think> tags. Please provide the numerical value (e.g., 42 or 3.14) within the <answer> </answer> tags.

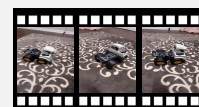

---

**Text Input**
How wide is the toy in the first frame (in centimeters)?

| **Bounding Box Input** | **Point Input** | **Mask Input** |
|---|---|---|
| `[{"x1":112, "y1":547,`
`"x2":717, "y2":1089}]` | `[[820, 437]]` | |

---

**SFT GT**
<think>By observing the size of the toy in the image and its relative proportion to surrounding objects, the width of the toy can be estimated. Based on the proportional reference in the frame, the toy's width is approximately 7.4 centimeters.</think> <answer>7.4 centimeters</answer>
**RL GT**
<think>*</think><answer>7.4</answer>

*Table F41.* **The route plan task QA preview for evaluation.**

**Route Plan/Navigation Task**

Please think about this question as if you were a human pondering deeply.
Engage in an internal dialogue using expressions such as 'let me think',
'wait', 'hmm', 'oh, I see', 'let's break it down', etc, or other natural
language thought expressions. It's encouraged to include self-reflection.
Video: `<video>`
Question: `<text>`
During RL: Please provide the thinking process within the <think> </think> tags. Please provide only the single option letter
(e.g., A, B, C, D, etc.) within the <answer> </answer> tags.

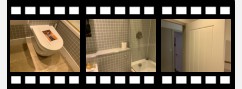

**Text Input**
What actions should you take to navigate from the toilet, facing the bathtub, to the door, given the following steps: 1. [please fill in] 2. Go forward until the door. You have reached the final destination?
A.Turn left B.Turn right C.Turn back

| **Bounding Box Input** | **Point Input** | **Mask Input** |
| --- | --- | --- |
| Not applicable. | Not applicable. | Not applicable. |

**SFT GT**
<think>Based on the layout in the image, assuming you start at the toilet, facing the bathtub, the first step is to turn right, then continue forward until you reach the door. This path is aligned with the relative positioning of the objects in the room.</think> <answer>Turn right</answer>
**RL GT**
<think>*</think><answer>B</answer>

*Table F42.* **The appearance order task QA preview.**

**Appearance Order Task**

Please think about this question as if you were a human pondering deeply.
Engage in an internal dialogue using expressions such as 'let me think',
'wait', 'hmm', 'oh, I see', 'let's break it down', etc, or other natural
language thought expressions. It's encouraged to include self-reflection.
Video: `<video>`
Question: `<text>`
During RL: Please provide the thinking process within the <think> </think> tags. Please provide only the single option letter
(e.g., A, B, C, D, etc.) within the <answer> </answer> tags.

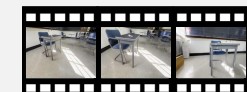

**Text Input**
Please provide the sequence in which these objects appear in the video: table, chair, blackboard.
Options: A.["table", "blackboard", "chair"] B. ["chair", "table", "blackboard"] C. ["blackboard", "chair", "table"] D. ["table", "chair", "blackboard"]

| **Bounding Box Input** | **Point Input** | **Mask Input** |
| --- | --- | --- |
| Not applicable. | Not applicable. | Not applicable. |

**SFT GT**
<think>Based on the objects' sequence in the image, the table appears first, followed by the chair, and finally the blackboard. This order is determined by the relative positions of the objects in the frame.</think> <answer>Table, chair, blackboard.</answer>
**RL GT**
<think>*</think><answer>D</answer>

*Table F43.* **The depth estimation task QA preview.**

**Depth Estimation Task**

Please think about this question as if you were a human pondering deeply. Engage in an internal dialogue using expressions such as 'let me think', 'wait', 'hmm', 'oh, I see', 'let's break it down', etc, or other natural language thought expressions. It's encouraged to include self-reflection.
Video: `<video>`
Question: `<text>`
During RL: Please provide the thinking process within the <think> </think> tags. Please provide the numerical value (e.g., 42 or 3.14) within the <answer> </answer> tags.

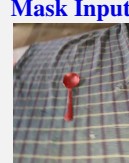

**Text Input**
Please provide the depth value of the object labeled by the red mask in the initial frame (in meters).

| **Bounding Box Input** | **Point Input** | **Mask Input** |
|---|---|---|
| `[{"x1":466, "y1":528, "x2":620, "y2":1043}]` | `[[733, 535]]` | 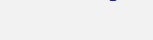 |

**SFT GT**
<think>Based on the position of the object labeled by the red mask in the image and considering the perspective and size of the object, the estimated depth of the object is 0.72 meters with the reference spoon's length of 0.17 meters.</think> <answer>0.72</answer>
**RL GT**
<think>*</think><answer>0.72</answer>

*Table F44.* **The view change inference task QA preview.**

**Area Estimation Task**

Please think about this question as if you were a human pondering deeply. Engage in an internal dialogue using expressions such as 'let me think', 'wait', 'hmm', 'oh, I see', 'let's break it down', etc, or other natural language thought expressions. It's encouraged to include self-reflection.
Video: `<video>`
Question: `<text>`
During RL: Please provide the thinking process within the <think> </think> tags. Please provide the numerical value (e.g., 42 or 3.14) within the <answer> </answer> tags.

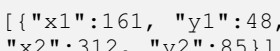

**Text Input**
What is the area of the roof shown in the video (in square meters)?

| **Bounding Box Input** | **Point Input** | **Mask Input** |
|---|---|---|
| `[{"x1":161, "y1":48, "x2":312, "y2":85}]` | Not applicable. | 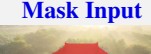 |

**SFT GT**
<think>Based on the perspective in the image and the relative size of the objects, the area of the roof can be estimated. By considering the proportions of the objects and the background perspective, the area of the roof is approximately 146.0 square meters.</think> <answer>The area of the roof in the video is 146.0 square meters.</answer>
**RL GT**
<think>*</think><answer>146.0</answer>

*Table F45.* **The object matching task QA preview.**

**Object Matching Task**

Please think about this question as if you were a human pondering deeply.
Engage in an internal dialogue using expressions such as 'let me think',
'wait', 'hmm', 'oh, I see', 'let's break it down', etc, or other natural
language thought expressions. It's encouraged to include self-reflection.
Video: `<video>`
Question: `<text>`
During RL: Please provide the thinking process within the `<think>` `</think>` tags. Please provide only the single option letter
(e.g., A, B, C, D, etc.) within the `<answer>` `</answer>` tags.

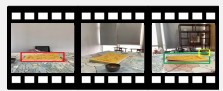

**Text Input**
Which bounding box (green, blue, or yellow) in the last frame is the same object as the red bounding box in the first frame?
Options: A.Yellow  B. Green

| Bounding Box Input | Point Input | Mask Input |
|---|---|---|
| `[{"x1":87, "y1":382,` `"x2":398, "y2":493}]` `[{"x1":122, "y1":515,` `"x2":237, "y2":617},` `{"x1":62, "y1":376,` `"x2":381, "y2":534}]` | Not applicable. | Not applicable. |

**SFT GT**
`<think>`By analyzing the object positions and color contrasts in the image, the green bounding box matches the object of
the red bounding box in the first frame. This conclusion is based on the relative positions and sizes of the objects in both
frames.`</think>` `<answer>`The green box.`</answer>`
**RL GT**
`<think>`*`</think>``<answer>`B`</answer>`

*Table F46.* **The obstacles location task QA preview.**

**Obstacles Location Task**

Please think about this question as if you were a human pondering deeply.
Engage in an internal dialogue using expressions such as 'let me think',
'wait', 'hmm', 'oh, I see', 'let's break it down', etc, or other natural
language thought expressions. It's encouraged to include self-reflection.
Video: `<video>`
Question: `<text>`
During RL: Please provide the thinking process within the `<think>` `</think>` tags. Please provide only the single option letter
(e.g., A, B, C, D, etc.) within the `<answer>` `</answer>` tags.

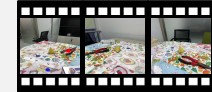

**Text Input**
Which object will cause obstacles when moving the pliers to the left of the box? (Red point denotes the pliers, Blue point
denotes the target position.)
Options: A.Pineapple  B. Charging plug  C. None

| Bounding Box Input | Point Input | Mask Input |
|---|---|---|
| Not applicable. | Not applicable. | Not applicable. |

**SFT GT**
`<think>`The red-handled pliers are on the right, and the target is on the left side of the box at the lower right. If the pliers are
moved to the box's left side, the path crosses the pineapple in the center of the table, making it the main obstacle.`</think>`
`<answer>`Pineapple.`</answer>`
**RL GT**
`<think>`*`</think>``<answer>`A`</answer>`

*Table F47.* **The manipulation planning task QA preview.**

**Manipulation Planning Task**

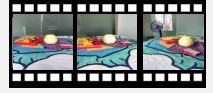

Please think about this question as if you were a human pondering deeply.
Engage in an internal dialogue using expressions such as 'let me think',
'wait', 'hmm', 'oh, I see', 'let's break it down', etc, or other natural
language thought expressions. It's encouraged to include self-reflection.
Video: `<video>`
Question: `<text>`
During RL: Please provide the thinking process within the <think> </think> tags. Please provide only the single option letter
(e.g., A, B, C, D, etc.) within the <answer> </answer> tags.

**Text Input**
How to move the box to the front of the apple from the viewpoint of the first frame? (Red point denotes the box, blue point
denotes the target position.)
Options: A.Moving backward 43.6cm  B. Moving left 10.2cm  C. Moving up 45.7cm  D. Moving backward 28.1cm

| **Bounding Box Input** | **Point Input** | **Mask Input** |
|---|---|---|
| Not applicable. | Not applicable. | Not applicable. |

**SFT GT**
<think>The red point denotes the current position of the box, and the blue point denotes the target. The task is to move
the box in front of the apple by shifting it along the red-to-blue direction about 28.1cm.</think> <answer>Move the book
backwards 28.1cm to put the box in front of the apple.</answer>
**RL GT**
<think>*</think><answer>D</answer>

*Table F48.* **The area estimation task QA preview.**

**View Change Inference Task**

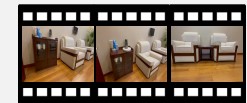

Please think about this question as if you were a human pondering deeply.
Engage in an internal dialogue using expressions such as 'let me think',
'wait', 'hmm', 'oh, I see', 'let's break it down', etc, or other natural
language thought expressions. It's encouraged to include self-reflection.
Video: `<video>`
Question: `<text>`
During RL: Please provide the thinking process within the <think> </think> tags. Please provide only the single option letter
(e.g., A, B, C, D, etc.) within the <answer> </answer> tags.

**Text Input**
What is the view change between the input frames?
Options: A.Back B. Down C. Right D. Left E. Front

| **Bounding Box Input** | **Point Input** | **Mask Input** |
|---|---|---|
| Not applicable. | Not applicable. | Not applicable. |

**SFT GT**
<think>By analyzing the angle change between the frames, it's clear that the view shifts downward. This conclusion is drawn
from comparing the position and angle of objects in the beginning frames.</think> <answer>Downward</answer>
**RL GT**
<think>*</think><answer>B</answer>