# OpenReview forum: "SpaceVista: All-Scale Visual Spatial Reasoning from mm to km"
_ICML.cc/2026/Conference — ICML 2026 regular_

### Official Review · Reviewer_wnMb · 2026-03-09

**Soundness:** 3
**Presentation:** 3
**Significance:** 3
**Originality:** 2
**Overall Recommendation:** 5
**Confidence:** 3

**Summary:**

The paper raises the question of spatial understanding using MLLMs across different scales, ranging from mm to 100m. The paper introduces the SpaceVista-1M dataset, which contains 1 million question–answer pairs from indoor and outdoor domains. Second, the article proposes an architecture with two types of encoders, along with scale-aware routing, and a training method consisting of SFT training and reinforcement learning. Third, the paper proposes a new benchmark SpaceVista-Bench. The obtained results demonstrate the effectiveness of the proposed model on various benchmarks, including those focused on spatial understanding tasks.

**Compliance With Llm Reviewing Policy:**

Affirmed.

**Final Justification:**

The authors' rebuttal addressed my questions, so I would like to raise my score. More specifically, the authors added technical details, positioning relative to previous baselines, the value of the model and the benchmark that could effectively cover different scene scales, as well as the correctness of the conducted ablations in terms of model size.

**Key Questions For Authors:**

- Please add more technical details about the training of the space-aware experts.
- Which types of questions are particularly challenging for the models in SpaceVista-Bench?
- Does the improvement in answers to Measurement-Related questions affect performance on Non-Measurement questions?
- Does training on all-scale data improve the quality of answers for individual scales, or would an expert model trained on a single scale perform better?

**Limitations:**

The authors discuss the limitations of the work.

**Strengths And Weaknesses:**

Strengths:

- Understanding metric distances across different scales with a single MLLM is an important problem, and the paper provides a strong motivation for it in the Introduction.
- The paper proposes a method that demonstrates strong empirical results both on the proposed new benchmark and on other benchmarks.
- Overall, the paper is well written and easy to follow, and the model architecture is supported by detailed ablation studies.

Weaknesses:
- Some technical details are partially omitted, in particular details about the training of LoRA-like Scale Experts, how the training data is split among the experts, and how semantic embeddings for the semantic recognition reward are computed.
- The idea of adding a self-supervised encoder is not entirely new; for example, it was used in Spatial-MLLM, which somewhat reduces the technical novelty of the work.
- The ablation studies are conducted on a 3B-based model, while comparisons with baselines are performed only with the 7B model.
- The paper does not discuss failure cases of the models on SpaceVista-Bench, nor which types of questions are particularly challenging.

---

> ### Author Rebuttal · Authors · 2026-03-30
>
> We thank Reviewer `wnMb` for this insightful feedback and integrate the following revision to the updated version (locations shown behind).
>
> * W1&Q1: Training Details `Line 378 in Sec.5, Line 1638 in Sec.C`
> * W4: Failure Case `Additional Sec.F.4`
> * Q2: Challenge Analysis `Line 415 in Sec.5`
> * Q3: Measurement-Related Impact `Line 2019 in Sec.D.5`
>
> > **W1&Q1 Missing Details:**
>
> To further clarify these details, we will make the following modifications to the manuscript:
>
> * Training Data (Sec. 5, Line 378): We will add: "The training data already includes four scale and scene labels from the construction process. These labels are used to divide the data, and each expert is trained on the data that matches its scale."
> * Semantic Reward (Sec. 5, Line 380): The semantic embeddings are computed using all-MiniLM-L6-v2 with the Sentence-Transformer Toolkit.
> * Training Details (Sec. C), we will provide supplementary training configurations for clarity: Built on PEFT and Transformers, the router has depth=4, hidden_size=1024, and softmax $\tau=0.5$. Experts are trained via DeepSpeed, bf16, and FlashAttention. Key HPs: grad_accum=1, $lr=10^{-6}$ (cosine), and weight decay=0.01.
>
> Note: The preview code is available on GitHub, but the link is prohibited for anonymity.
>
> > **W2 Spatial-MLLM Comparison:**
>
> There seems to be a slight misunderstanding. Spatial-MLLM actually utilizes the **supervised** encoder VGGT. In our initial version, we have cited Spatial-MLLM in line 155 and discussed it in Tab.3, Sec. C2, and Sec. D6, as:
>
> * Semantic info (Sec. C2, line 1652): while VGGT shows strong reconstruction capabilities, it **lacks rich semantic content**.
> * Sensitivity (Sec. D6): VGGT is **more sensitive to low-resolution videos**.
>
> While models such as Cambrian-1 use DINO for general video, our use of DINO v3 highlights a different, orthogonal point: pipelines built on supervised VGGT branches can perform well in indoor settings, whereas DINO v3 offers self-supervised geometric and depth priors that better support general, all-scale scenes.
>
> > **W3 Ablation on 3B:**
>
> In the GPT-4 report, they introduced the concept of **predictable scaling**. By training smaller models with just lower compute, they accurately forecast the final performance. Because training models multiple times on full data is expensive and impractical, testing on smaller models and reduced data has become **common practice**, as seen in SmolLM2 and LLaMA. The performance of 3B, 7B, and 32B is also analyzed in Sec.D.7.
>
> > **W4&Q2 Failure Case and Challenge Analysis:**
>
> **Qualitative Study**: Besides our discussion in Fig. 2 and Sec. D3, we provide [an arena with more failure cases here](https://mm2km.github.io/failure_cases). We find 1) Our SpaceVista is generally accurate in MR QA. 2) Current models are still weak in info integration across time.
>
> **Quantitative Study**: Challenges vary for different dimensions.
>
> * **Scale.** Tiny-scale scenarios are more challenging in Line 415, mainly because small absolute deviations can translate into large relative errors.
>
> * **OOD size.** In Sec. D3, OOD samples are more challenging because their atypical sizes require reasoning rather than memorization.
>
> * **Measurement-Related.** MR questions are generally more difficult than NMR questions. See Q3 below for details.
>
> * **Capability.** According to Sec. D4, Room Size and Route Planning tend to be difficult because they require global and compositional cues.
>
> We will add the above insights to Line 415 for clarity.
>
> > **Q3 Measurement-Related Impact:**
>
> To explore this impact, we conducted a statistical comparison between Measurement-Related (MR) and Non-Measurement-Related (NMR) cases. Under the same conditions, the results are:
>
> |Model|Avg.|MR|NMR|
> |-|-|-|-|
> |GPT-5|35.5|28.6|42.4|
> |Gemini-2.5 Pro|35.9|29.4|42.5|
> |InternVL2(8B/40B)|28.6/32.8|23.0/25.1|34.1/40.5|
> |VILA-1.5(8B/40B)|20.9/23.6|15.5/18.6|26.3/28.7|
> |LongVA-7B|19.9|13.4|26.5|
> |LLaVA-NeXT-Video(7B/72B)|14.7/18.1|10.1/13.6|19.3/22.7|
> |Qwen2.5-VL-7B|24.9|18.9|30.9|
> |SpaceVista-7B(Ours)|41.3|35.0|47.6|
>
> Therefore, we believe it is **difficult to** make a definitive claim about whether MR positively or negatively influences NMR. Nevertheless, MR tasks remain consistently **more challenging**. We will incorporate this insight into Sec.D.5.
>
> > **Q4 Performance on Single-scale**
>
> * Direct answer: For a single scale, the **expert model** trained exclusively on that scale achieves **better** performance, as shown in Fig. 4(a–c). This may be attributed to **potential knowledge conflicts** at Line 223. To mitigate this issue, we further design the scale router, which alleviates such conflicts, as demonstrated in Fig. 4(d).
> * Meaning of all-scale: The idea of the all-scale model is not about surpassing every specific-scale model. Actually, the scale is often **not known in advance**, but **emerges as the task unfolds**, like a robot doing 100m navigation and 1mm manipulation consequently. (Detail see Reviewer `7JPF Q2`)

---

> > ### Author Rebuttal · Reviewer_wnMb · 2026-04-02
> >
> > Thank you to the authors for their detailed response and for conducting the additional experiments, which have addressed my concerns regarding the paper.

---

> > > ### Author Response · Authors · 2026-04-02
> > >
> > > We sincerely thank Reviewer wnMb for the positive feedback and for recognizing our efforts to address the concerns raised. We are encouraged to know that our clarifications and revisions have **fully addressed** your questions.
> > >
> > > In light of this, we would be grateful if you would kindly reconsider your score.

---

### Official Review · Reviewer_7JPF · 2026-03-10

**Soundness:** 3
**Presentation:** 3
**Significance:** 3
**Originality:** 3
**Overall Recommendation:** 4
**Confidence:** 4

**Summary:**

SpaceVista is a comprehensive framework designed to advance "all-scale" visual spatial reasoning, spanning from millimeters to kilometers. It introduces the SpaceVista-Bench, SpaceVista-1M dataset containing 1 million spatial QAs across 19 tasks, and the SpaceVista-7B model, which uses scale-aware experts and a progressive training paradigm to resolve knowledge conflicts across different spatial magnitudes. By moving beyond traditional indoor-only 3D reasoning, the work establishes a new benchmark for generalizing spatial intelligence across diverse real-world scenarios.

**Compliance With Llm Reviewing Policy:**

Affirmed.

**Final Justification:**

I thank the authors for their thorough response, and I have no further questions. I would like to keep my original rating *Weak Accept*.

**Key Questions For Authors:**

As I mentioned in the weakness part:

1. Relying on estimated geometry from 2D images for data generation may introduce inherent noise and scale inconsistencies within the training set. Could authors provide an analysis or explanation on how to mitigate such errors in data generation?

2. I am not sure if “all-scale visual spatial reasoning” is meaningful in practical application, since downstream tasks of spatial reasoning are mainly embodied agents like robotics or self-driving cars, which usually operate within limited ranges of scales. Could the authors provide an analysis or explanation?

**Limitations:**

yes

**Strengths And Weaknesses:**

**Strength:**
1. It breaks the "single-scene" bottleneck by covering an unprecedented range of spatial scales from millimeters to kilometers in a unified framework.

2. The progressive data engine enables large-scale spatial task curation without relying on expensive and labor-intensive 3D scans.

**Weaknesses**
1. Relying on estimated geometry from 2D images for data generation may introduce inherent noise and scale inconsistencies within the training set.

2. I am not sure if “all-scale visual spatial reasoning” is useful in practical application, since downstream tasks of spatial reasoning are mainly embodied agents like robotics or self-driving cars, which usually operate within limited ranges of scales.

---

> ### Author Rebuttal · Authors · 2026-03-30
>
> We thank Reviewer `7JPF` for taking the time to review our paper and provide valuable feedback. We will integrate the following advice into the updated version (locations shown behind).
>
> * W1&Q1: Potential Errors `Line 212 in Sec.3`
> * W2&Q2: Application Discussion `Line 437 in Sec.6`
>
> > **W1&Q1 Potential Error**
>
> We have recognized this error and took steps to analyze it in the initial version as follows:
>
> * **Quality control.** Estimating geometry from 2D images is a well-established and reliable approach. Early methods, such as SpatialVLM proved effective on simple images, while recent video-based models, such as Video-Depth-Anything have further enhanced this direction. Additionally, we apply rigorous data filtering in Line 224 to remove incorrect examples and achieve an acceptable preference rate.
> * **Human preference.** As a standard practice for large-scale datasets, the result of manual inspection indicates a high human preference rate in Sec B.4.7, which is acceptable and comparable to recent prominent works inlcuding ShareGPT4Video and LVD-2M.
> * **Data effectiveness.** The ultimate test of data quality is its impact on model performance. As demonstrated in Tab. 2 and Tab. 5, models trained on our dataset show significant performance gains (improvements of 2-5% and 10%, respectively). This strongly indicates that minor errors do not diminish the dataset's overall contribution.
> * **High-quality benchmark with rigorous evaluation.** We acknowledge the presence of potential errors common in automated pipelines. To accurately assess their impact, we constructed a highly accurate, hand-recorded, manually annotated benchmark. The consistent performance improvements observed on our rigorous benchmark confirm that the potential errors in the training data are acceptable.
>
> To further emphasize this, we will highlight these quality control and evaluation measures in Sec. 3 of the revised paper.
>
>
> > **W2&Q2 All-Scale or Specific Scale?**
>
> Most previous models performed better only in indoor settings, but experienced performance degradation at other scales shown in Tab. 2. This motivated us to tackle the **challenge of balancing** the agent’s performance under broader conditions.
>
> We argue that all-scale models are **an option rather than a substitute** for scale-specific models, and their value lies in applications that require cross-scale capability, rather than in being universally better than scale-specific models.
>
> * **Application**: For tasks with a clear dominant scale, scale-specific models can be sufficient. However, many applications, like embodied intelligence, require seamless transitions across multiple orders of magnitude, from **large-scale navigation** (100m) to **fine-grained manipulation**  (1mm). In these settings, scale is not merely a modeling preference; it is intrinsic to the task.
>
> * **Environments**: In open and dynamic environments, the relevant scale is often **not known in advance**, but **emerges as the task unfolds**. This makes scale-specific solutions less natural for scenarios where perception must remain coherent across changing spatial granularities.
>
> From this perspective, all-scale models do not simply compete with scale-specific ones. Instead, they enable a fundamentally different capability: unified modeling across scales for applications whose structure is dynamic, coupled, and open-ended. Therefore, next-generation intelligence demands this kind of cross-scale **continuity and generalization**. This is exactly the space where our contribution matters.
>
> If you still have any concerns or questions, please feel free to let us know.

---

> > ### Author Rebuttal · Reviewer_7JPF · 2026-03-31
> >
> > I thank the authors for their thorough response, and I have no further questions.

---

> > > ### Author Response · Authors · 2026-04-01
> > >
> > > We sincerely thank Reviewer 7JPF for the positive feedback and for recognizing our efforts in addressing the concerns. It is encouraging to see that the clarifications and revisions have **fully addressed your questions**.
> > >
> > > We would be grateful if you could also kindly reconsider your score.

---

### Official Review · Reviewer_cEEG · 2026-03-13

**Soundness:** 3
**Presentation:** 3
**Significance:** 2
**Originality:** 2
**Overall Recommendation:** 4
**Confidence:** 3

**Summary:**

Existing spatial reasoning research is mostly confined to indoor scenes, relies on labor-intensive annotations, and lacks all-scale modeling. This paper introduces SpaceVista-1M, a large-scale dataset with 1M QA pairs across 38K video scenes spanning mm to km scales and 19 tasks.

To enable reliable evaluation, the authors construct SpaceVista-Bench, a high-accuracy benchmark with 3K manually verified QA pairs aligned with physical world measurements.
The proposed SpaceVista-7B model integrates non-semantic spatial features (e.g., depth, geometry) from self-supervised encoders, adopts LoRA-like scale experts to mitigate cross-scale conflicts, and uses progressive rewards mimicking human reasoning.

Extensive experiments across 5 benchmarks show that SpaceVista-7B outperforms existing open-source and specialized models, especially in all-scale generalization.

**Compliance With Llm Reviewing Policy:**

Affirmed.

**Final Justification:**

Thank the authors for their detailed response.
My concerns have been addressed.

**Key Questions For Authors:**

1. The manuscript mentions that DINOv3 outperforms VGGT in capturing spatial cues—could the authors elaborate on the specific weaknesses of VGGT in all-scale reasoning that DINOv3 effectively mitigates?

2. How to effectively tackle tiny tabletop and drone scenes, given the relatively scarce data available for these specific scenarios?

**Limitations:**

Limitations have not been discussed, and I see no obvious limitations.

**Strengths And Weaknesses:**

### Strengths
1. The work effectively tackles two critical gaps in current spatial reasoning research: the over-reliance on indoor-centric, scale-constrained scenarios and the lack of holistic all-scale training frameworks. The proposed SpaceVista-1M dataset fills the scale gap, and SpaceVista-7B model can generalize across diverse real-world applications.

2. The data construction has a meticulous, reproducible pipeline and strict quality assurance mechanisms, which benefits the visual spatial reasoning community.

3. The proposed SpaceVista-7B model demonstrates effectiveness for all-scale reasoning, equipped with DINOv3 features, LoRA-like scale experts with router and the progressive reward design aligns with human reasoning processes.

### Weaknesses
This paper is comprehensively detailed in its approach and execution, and I do not identify any obvious weaknesses in its core contributions or technical implementation.

However, given my limited familiarity with the broader landscape of spatial reasoning benchmarks, other reviewers with deeper expertise in this specific area may be able to provide further insights.

---

> ### Author Rebuttal · Authors · 2026-03-30
>
> We are glad to see that reviewer `cEEG` found our approach interesting and our results good enough to support our claims. We will revise our paper to improve the clarity (locations shown behind).
>
> * Q1: VGGT detailed explanation `line 439 in Sec.5`
> * Q2: Distribution Discussion `line 438 in Sec.6`
>
> > **Q1: Comparison of VGGT and DINO v3**
>
> We sincerely appreciate your insightful question. In fact, we **have conducted comprehensive experiments** in the appendix, providing detailed ablations and analyses of VGGT encoder and DINO v3 encoder.
>
> * **Lack semantic representation (Sec.C2)**: As discussed in the appendix (line 1652), while VGGT shows strong reconstruction performance, it relies on annotations that do not provide rich semantic information. In addition, its strong dependence on a large decoder for geometry recovery indicates that the encoder alone offers only raw representation ability.
> * **Sensitive to low-resolution (Sec.D6)**: VGGT is notably more sensitive to low-resolution videos, whereas DINO v3 exhibits greater overall robustness. Additionally, training on images rendered from 12 different viewpoints of VGGT's reconstruction generally yields **better performance** than using the VGGT encoder directly. This indicates that features extracted solely from the VGGT encoder lack sufficient semantic information, performing even worse than the semantics obtained from the rendered views of its own reconstruction.
>
> > **Q2: Scene Distribution**
>
> * **Tiny tabletop:** As shown in Fig.3(b), tiny tabletop data constitutes 26.8% of the overall dataset, indicating that it is **well-represented and not scarce**.
> * **Drone:** We indeed discuss this limitation of the drone in line 1063. During construction, acquiring high-quality drone data presents significant challenges. Despite our efforts to ensure a balanced dataset, the **inherent scarcity** of public raw drone videos, with our rigorous validation criteria, inevitably results in a limited number of drone scenes.
> * To transparently discuss this, we will add the **following statement** to the revised manuscript: "Due to the scarcity of public raw drone datasets, our exploration in drone scenarios remains preliminary." And we are still committed to including more relevant data for the development of our community, as the data becomes publicly available.
> * Therefore, we hope you can understand this dilemma, which **does not undermine** our core contribution of bridging diverse spatial scales.
>
> If you still have any concerns or questions, please feel free to discuss them with us.

---

> > ### Author Rebuttal · Reviewer_cEEG · 2026-04-03
> >
> > Thank the authors for their detailed response.
> > My question has been solved. I will keep my rating.

---

> > > ### Author Response · Authors · 2026-04-04
> > >
> > > We sincerely thank Reviewer wnMb for the positive feedback and for recognizing our efforts to address the concerns raised. We are encouraged to know that our clarifications and revisions have **fully addressed** your questions.

---

### Official Review · Reviewer_vCEv · 2026-03-13

**Soundness:** 2
**Presentation:** 4
**Significance:** 2
**Originality:** 3
**Overall Recommendation:** 4
**Confidence:** 4

**Summary:**

This work studies all-scale spatial reasoning for vision-language models from millimeter to kilometer levels, which remains under-explored in existing research. It presents a complete framework including the large-scale SpaceVista-1M dataset, the SpaceVista-Bench benchmark, and the SpaceVista-7B model with feature fusion, scale experts, and progressive reward learning. Experiments demonstrate competitive performance and clear advantages on the proposed benchmark.

**Compliance With Llm Reviewing Policy:**

Affirmed.

**Final Justification:**

My concerns have been adequately addressed.

**Key Questions For Authors:**

Given the length of the paper, I am unsure whether I have missed some key details. My questions are as follows:
1. The SFT training uses free-form responses, while RL training uses multiple-choice questions. I am unclear about the final evaluation format in the benchmark. For the multiple-choice setup, how are the options constructed? Could the option construction affect the final performance?
2. In Table 4, performance drops when using 6 experts. I don't find an explanation for this phenomenon in the paper.
3. As shown in Figure 4, drone scenes account for only 0.4% of the total dataset. This indicates a severe scale imbalance across different categories.
The authors are also encouraged to address the concerns mentioned in the weaknesses section.

**Limitations:**

The paper does not include a discussion on limitations or potential negative societal impacts. I suggest the authors at least clarify the limitations related to data construction and quality assurance.

**Strengths And Weaknesses:**

### **strengths:**
1. The topic is novel, focusing on all-scale spatial reasoning from millimeter to kilometer levels, which is under-explored in current vision-language research.
2. The work demonstrates substantial and thorough efforts, as reflected by the extensive appendix and comprehensive experimental validation.
3. SpaceVista-1M serves as a valuable large-scale dataset for cross-scale spatial reasoning, supporting future research in this direction.

### **weaknesses**
1. SpaceVista-1M is mainly constructed via an automatic pipeline. Only a subset of training data was manually inspected, with an accuracy of around 83%, raising concerns about overall data quality.
2. In tab.5, although SpaceVista-7B achieves the best overall performance on SpaceVista-Bench, it does not outperform strong commercial models in any single sub-category. This raises a fundamental question: is building an all-scale model meaningful, or is a model specialized for one specific scale more valuable in the application?
3. SpaceVista-7B does not show outstanding performance on other benchmarks such as MMSI-Bench (Table D20) and STI-Bench (Table D18), which fails to convincingly demonstrate that the model has truly earned spatial reasoning capabilities. In contrast, while it performs best on VSI-Bench (Table D21), the baseline comparisons in this benchmark do not include strong models such as Gemini 2.5 Pro.

---

> ### Author Rebuttal · Authors · 2026-03-30
>
> We thank Reviewer `vCEv` for the valuable feedback. The revisions to our manuscript will be (locations shown behind):
> * W1: Imprecise terms `line 239 in Sec.3`, `Sec.B.4.7`
> * W3: Extra models on VSI-Bench `Tab. D21 in Sec. D.4`
> * Q1: Option construction `line 1338 in Sec. B.3.7`
> * Q2: Missing explanation `line 429 in Sec. 5`
> * Q3: Distribution discussion `line 438 in Sec. 6`
>
> > **W1: Quality of Training Data:**
>
> We apologize for the imprecise use of “accuracy”. The reported metric is preference rate (or acceptance rate), following the human evaluation protocol like ShareGPT4Video. Therefore, ~83% means that 83% are preferred by human evaluators, rather than implying that the remaining 17% are factual errors. This term will be revised in Sec. 3 and Sec. B.4.7.
> * Subset-based manual inspection is a **common practice** for large-scale datasets. Our result indicates a rate of ~83%, which is consistent with popular automatically curated video datasets, e.g., 73% for ShareGPT4Video, 75% for LVD-2M, and 81% for OPENVID-1M.
> * Additionally, the **effectiveness of the training data** is well-demonstrated in Tab. 2 (+2-5% w/. training data) and Tab. 5 (+10% w/. training data). We include the preference rate purely for transparency.
>
> > **W2: Application and Comparison on commercial models:**
>
> * Although commercial models trained on more parameters and data do succeed in some subcategories and fail in others, SpaceVista-7B achieves the best overall score and shows no obvious weaknesses. We conjecture that the variation in peak performance across commercial models may be attributed to differences in their internal data distributions, revealing their **potential biases and limitations**.
> * We hold the firm belief that the application of our all-scale models mainly lies in dynamic, coupled, and open-ended tasks. Emerging applications such as embodied intelligence often require seamless transitions from **large-scale navigation** (\~100m) to **fine-grained manipulation** (\~1mm). In such settings, scale is not fixed in advance but evolves with the task, making unified modeling across scales a fundamentally different capability rather than a replacement for specific scales.
>
> > **W3: Detailed performance with other benchmarks:**
>
> * In Fig.2, benchmarks such as VSI-Bench and MMSI-Bench show a small **scale range**, which further introduces significant trade-offs between specialization and generalization in Sec.4 (Overview) and Fig.4(a–d). In addition, MMSI-Bench, which focuses on multi-image, is evaluated in a different way.
> * Despite the challenging conditions above, our results remain **broadly comparable** across benchmarks. On MMSI-Bench and STI-Bench, our model is surpassed only by commercial models and by open-source models that are 10× larger.
> * We apologize for missing Gemini 2.5 Pro in the appendix. The detailed results below are reproduced and added to Tab. D21, which is also mentioned as the average score in Tab. 2.
>
> |Model|Avg.|App. Order|Abs. Dis.|Count|Rel. Dis.|Size Est.|Room Est.|Route Plan|Rel. Dir.|
> |-|-|-|-|-|-|-|-|-|-|
> |GPT-5|44.2|46.9|28.3|51.8|44.7|59.2|39.4|33.0|48.4|
> |Gemini-2.5 Pro|45.0|41.1|31.2|54.5|48.3|59.7|42.9|36.8|45.8|
> |...||||||||||
> |SpaceVista-7B|48.6|56.3|36.0|62.9|44.2|58.1|42.0|38.9|49.7|
>
>
> > **Q1: Option construction**
>
> We would like to clarify that our RL and evaluation encompass both multiple-choice and regression, as noted in Line 96 and Tab. 1.
>
> * **Regression**: Option is not applicable here, and the final score is based on the VSI-Bench protocol (MRA:.5:.95:.05).
>
> * **Multi-Choice**: 1) Object-related questions: Distractors are randomly sampled according to co-occurrence frequency to ensure sufficient challenge and confusion. 2) Fixed-option questions (e.g., directions): Distractors are sampled from predefined patterns. 3) Relatively open questions: The plausible yet incorrect distractors are generated by the powerful LLM. After construction, we then randomize option lengths and answer distributions to prevent shortcuts.
>
> > **Q2: Missing explanation**
>
> * We will add the description in line 429: “However, more routers are not always better; increasing the number of experts places greater pressure on the router and leads to load imbalance.” This insight will be detailed in the expert allocation mentioned in Sec. D.2.
>
> > **Q3: Drone Scene**
>
> * We indeed discuss this limitation in line 1063. Despite our efforts to ensure the balance, the scarcity of public drone video, coupled with our rigorous validation criteria, inevitably led to limited drone scenes. We will add the discussion to line 256: *"Due to the scarcity... exploration in drone scenarios remains preliminary."* But we are still committed to including more relevant drone data for a better community, as the data becomes publicly available. Therefore, we hope you understand this dilemma, which does not undermine our core contribution.
>
> Your feedback is really important to us, so please feel free to share any further thoughts.

---

> > ### Author Rebuttal · Reviewer_vCEv · 2026-04-04
> >
> > W1: The argument regarding the preference rate sounds reasonable. Since SpaceVista-1M is mainly constructed through an automatic pipeline, and data is a very important foundation of this work, I remain cautious about the data quality.
> >
> > W3: How can the current results show that SpaceVista-7B has truly gained spatial reasoning capabilities?

---

> > > ### Author Response · Authors · 2026-04-04
> > >
> > > Thank you very much for participating in the follow-up discussion. We further provide the following clarification.
> > >
> > > > W1: Automatic pipeline and data quality.
> > >
> > > Thanks for agreeing that our analysis of the training data is **reasonable**. To further address this concern, we summarize 5 major pieces of evidence for data quality:
> > >
> > > 1. **Automatic pipelines.** Using automatic pipelines to construct training data is common practice. Examples include ShareGPT4Video, LVD-2M, and OpenVid-1M for general video training, and SpatialVLM for spatial image training.
> > > 2. **Human preference rate.** As the acknowledgment above also noted, our preference rate is reasonable and comparable to prior work.
> > > 3. **Downstream performance gains.** As shown in Table 2 and Table 5, models trained on our dataset achieve clear improvements (2–10%, respectively), indicating that the dataset is effective and of sufficient quality.
> > > 4. **Qualitative visualization.** We showcase a wide range high quality visualization examples for training from Page 43-55. Additionally, failure cases are also added in `W3` below for insights.
> > > 5. **Quality control.** In the Sec B.4.7, we provide a detailed discussion of potential errors and quality control. Reviewer `7JPF` also indicated in `W1/Q1` that this concern had been fully addressed.
> > >
> > > Taken together, we believe these analyses constitute a comprehensive evaluation of training data quality under **standard community practices**. Sincerely hope the reviewer can understand our dedication, and we will add this detailed summary in Sec.3 and Sec.B.4.7.
> > >
> > >
> > > > W3: True gain of spatial reasoning capabilities
> > >
> > > **Before everything:** Due to the scale differences, we discussed the trade-offs between specialization and generalization in Sec. 4 (Overview) and Fig. 4(a–d). On VSI-Bench/MMSI-Bench, our model previously achieved up to a **5.5%/3.8% performance gain using specific-scale data** shown below. However, to develop an all-scale model, we chose to report the results obtained after all-scale balance. We therefore argue that the value of an all-scale model lies in **balanced cross-scale capability**, rather than **universally superior** to every scale-specific model.
> > >
> > > |Qwen2.5VL-7B|VSI-Bench|MMSI-Bench|SpaceVista-Bench|
> > > |-|-|-|-|
> > > |vanilla w/. indoor data|**54.1**|**36.1**|22.8|
> > > |vanilla w/. all-scale data|42.0|27.3|29.5|
> > > |Ours w/. all-scale data|48.6|32.3|**39.2**|
> > >
> > > We believe there is strong evidence that the improvement of **SpaceVista-7B** is a **true gain** for generalization.
> > >
> > > 1. Our all-scale benchmarks. As described in Sec. B.2.7, we construct a highly accurate benchmark through **manual collection, annotation, and retrieval**. Since this benchmark is not derived from the automated pipeline, the observed improvement proves generalization.
> > >
> > > 2. Previous benchmarks. Although prior benchmarks differ in scale, distribution, and evaluation protocols, our model achieves **substantial improvements over prior models of comparable size**. Moreover, the results remain broadly comparable across benchmarks with even commerial model.
> > >
> > > 3. Qualitative analysis. Besides examples in Figure 2, following Reviewer `wnMb`’s suggestion, we provide an [arena with more failure cases](https://mm2km.github.io/failure_cases) with model comparison in `wnMb W4&Q2`.Reviewer `wnMb` also expressed satisfaction with this clarification.
> > >
> > > 4. Additional analysis. In Sec.D.3, Fig. D25 and [preview website](https://mm2km.github.io/guinnessworldrecords.html), we evaluate our model on an **out-of-distribution benchmark of unusual object sizes** constructed from Guinness World Records, where it performs better than popular baseline models. In addition, as shown in Sec. D.4, the model demonstrates **reasonable overall performance on General Video Bench**.
> > >
> > > Therefore, it is undeniable that **general balanced cross-scale capability** is achieved as genuine gains in all-scale spatial reasoning.
> > >
> > > > Summary
> > >
> > > We would like to further emphasize that, to the best of our knowledge, this work represents **the first attempt to broaden the scope of all-scale spatial intelligence**. Our contribution to the community is not limited to introducing spatial reasoning data and models; more importantly, we offer **valuable initial exploration** of all-scale reasoning as an emerging research direction. Moreover, we provide **extensive analysis** to illustrate the challenging nature of the task and offer **deeper insights** into the unique application of all-scale reasoning.
> > >
> > > We sincerely hope these points help address your concerns. If our statement helps, we would be grateful for kindly reconsidering your score.

---

### Decision · Program_Chairs · 2026-04-30

**Decision:**

Accept (regular)

**Comment:**

This paper was reviewed by four experts in the field, with recommendations of three Weak Accepts and one Accept. Based on the reviewers’ feedback, the decision is to recommend acceptance of the paper. The reviewers also raised several valuable concerns that should be addressed in the final camera-ready version, particularly the need for more detailed experimental comparisons and ablation studies, as noted by Reviewers vCEv and wnMb, as well as clearer presentation and exposition, as highlighted by Reviewers 7JPF and wnMb. The authors are encouraged to address these issues to the best of their ability.